# GRAF1 integrates PINK1-Parkin signaling and actin dynamics to mediate cardiac mitochondrial homeostasis

Qiang Zhu [1], Matthew E. Combs[1], Juan Liu[2], Xue Bai[1], Wenbo B. Wang[3], Laura E. Herring [4], Jiandong Liu [1,5], Jason W. Locasale[2], Dawn E. Bowles[6], Ryan T. Gross[6], Michelle Mendiola Pla [6], Christopher P. Mack[1,5] & Joan M. Taylor[1,5] ✉

The serine/threonine kinase, PINK1, and the E3 ubiquitin ligase, Parkin, are known to facilitate LC3-dependent autophagosomal encasement and lysosomal clearance of dysfunctional mitochondria, and defects in this process contribute to a variety of cardiometabolic and neurological diseases. Although recent evidence indicates that dynamic actin remodeling plays an important role in PINK1/Parkin-mediated mitochondrial autophagy (mitophagy), the underlying signaling mechanisms remain unknown. Here, we identify the RhoGAP GRAF1 (*Arhgap26*) as a PINK1 substrate that regulates mitophagy. GRAF1 promotes the release of damaged mitochondria from F-actin anchors, regulates mitochondrial-associated Arp2/3-mediated actin remodeling and facilitates Parkin-LC3 interactions to enhance mitochondria capture by autophagosomes. Graf1 phosphorylation on PINK1-dependent sites is dysregulated in human heart failure, and cardiomyocyte-restricted Graf1 depletion in mice blunts mitochondrial clearance and attenuates compensatory metabolic adaptations to stress. Overall, we identify GRAF1 as an enzyme that coordinates cytoskeletal and metabolic remodeling to promote cardioprotection.

Mitochondria are essential for diverse cellular processes. In addition to generating ATP for energy, they produce reactive oxygen species (ROS) as a byproduct of their ATP-yielding electron transport activity and provide metabolites for anabolism and substrates for signaling reactions. Thus, even slight mitochondrial dysfunction can disrupt cellular redox balance, impair calcium homeostasis, and readily lead to cell death[1,2]. Consequently, several efficient quality control mechanisms have evolved to ensure mitochondrial homeostasis including mitochondrial fission, fusion, biogenesis, and selective autophagic clearance (i.e. mitophagy). Highly metabolic, long-lived cells like cardiomyocytes and neurons are particularly reliant on mitophagy as they have limited capacity for cell division-dependent dilution of damaged mitochondria or for altered mitochondrial dynamics[3]. Indeed, dysregulation of mitophagy has been linked to the development of heart failure and several neurological disorders including Parkinson's and Alzheimer's disease (PD, AD)[4]. Unfortunately, effective strategies to promote the clearance of damaged mitochondria are currently lacking.

The mitophagic signaling pathways that control the rapid recognition, ubiquitin-tagging and encasement of dysfunctional

[1]Department of Pathology, University of North Carolina, Chapel Hill, NC 27599, USA. [2]Department of Pharmacology and Cancer Biology, Duke University School of Medicine, Durham, NC 27710, USA. [3]Department of Biostatistics, University of North Carolina, Chapel Hill, NC 27599, USA. [4]UNC Proteomics Core Facility, Department of Pharmacology, University of North Carolina, Chapel Hill, NC 27599, USA. [5]McAllister Heart Institute University of North Carolina, Chapel Hill, NC 27599, USA. [6]Division of Surgical Sciences, Duke University Medical Center, Durham, NC 27710, USA. ✉e-mail: joan_m_taylor@med.unc.edu

mitochondria into protective double-membrane structures have been well characterized. A major mechanism involves the serine/threonine kinase PINK1 and the E3 ubiquitin ligase, Parkin. In healthy mitochondria, PINK1 is transported to the inner membrane (IMM) and is degraded by mitochondrial proteases. This translocation event is inhibited in depolarized damaged mitochondria leading to PINK1 accumulation on the outer membrane (OMM)[5,6]. PINK1-dependent phosphorylation of OMM ubiquitin (pUb) and mitofusin-2 (Mfn2), a dual regulator of fusion and mitophagy leads to the recruitment of Parkin to damaged mitochondria[4,7,8]. Subsequent phosphorylation of Parkin by PINK1 combined with Parkin's association with pUb induces a conformational change that enables E2 conjugating enzyme binding and full activation. Upon full activation, Parkin ubiquitinates several OMM substrates[9] that subsequently recruit ubiquitin-binding adapter proteins such as p62, NBR1, NDP52, Tax1BP1, and optineurin (OPTN) that bind ubiquitin and the autophagosome-associated protein, microtubule-associated protein light chain 3 (LC3), providing a critical link between the cargo to be degraded and the autophagosome[4-13]. While certain forms of mitophagy (such as those that involve Rab9-mediated endosomal-mediated clearance pathways) occur in a Parkin-independent fashion[14], it is clear from studies performed in multiple tissues and cell types that the PINK1-Parkin pathway renders mitophagy more efficient. Accordingly, mutations in PINK1 and Parkin contribute to the pathogenesis of neurodegenerative and other related metabolic diseases in patients[15]. Parkin-dependent mitophagy has also been shown to be required for metabolic maturation of postnatal cardiomyocytes[16], for cardioprotection provided by ischemic preconditioning[17] and for limiting cell death following myocardial infarction[18,19].

What is less known is how mitochondrial clearance processes are spatially controlled. During cell growth and differentiation, coordinated microtubule and actin dynamics position functional mitochondrial to specific intracellular locales that couple local energy production with metabolic needs. Following damage, mitochondria relocate to subcellular regions that facilitate their clearance which, in many cell types, involves retrograde trafficking to peri-nuclear lysosomes[20,21]. While recent studies indicate that actin remodeling can facilitate Parkin-dependent mitochondrial clearance[22-27], the underlying players that coordinate localized cytoskeletal remodeling to facilitate the trafficking and clearance of damaged mitochondria remain unclear.

The GRAF family of RhoA GTPase activating proteins (GAP) are poised to co-regulate actin- and lipid-dynamics by virtue of their shared multi-domain structure[28-32]. We previously reported that *Arhgap26* which encodes for GRAF1 (GTPase regulator associated with Focal adhesion kinase) is strongly expressed in highly metabolic tissues (i.e. heart, brain, and skeletal muscle) and is important for limiting RhoA-dependent F-actin formation and for maintenance of striated muscle tissues[28-33]. We now demonstrate that GRAF1 is rapidly recruited to damaged mitochondria where, upon phosphorylation by PINK1 (and/or PINK1-dependent kinases), it facilitates mitophagosome formation by controlling localized actin remodeling, mitochondrial trafficking, and Parkin-LC3 interactions. This pathway is conserved between various cell types, and data from patient samples and preclinical mouse models indicate that GRAF1 dysregulation contributes to (and may be a viable therapeutic target for) heart failure and perhaps other cardiometabolic or neurological diseases that result from failed mitochondrial clearance.

## Results

### GRAF1 regulates mitochondrial homeostasis in the heart
We previously demonstrated that GRAF1 depletion significantly exacerbated cardiac and skeletal muscle degeneration in the *mdx* model of Duchenne's muscular dystrophy (DMD)[32]. Given that mitochondrial dysfunction is a well-known pathological feature of DMD and stress-induced cardiomyopathies and that myofiber necrosis occurs because of failed clearance of damaged mitochondria in such models[34,35], we explored the possibility that GRAF1 deficiency alone might impact mitochondrial function. In support of this possibility, ultrastructural imaging of wild type (WT) mouse hearts revealed uniform ovoid mitochondria arranged as parallel arrays alongside myofibrils while hearts isolated from a GRAF1-germline deficient mouse line, GRAF1^gt/gt (which harbors the gene trapping vector VICTR48 within the first intron of Graf1[33]) had a significantly higher number of mitochondria that were small, more round and more randomly dispersed (Fig. 1a–c; Fig. S1a–c) suggesting compromised cardiac mitochondrial fitness. Accordingly, siRNA-mediated depletion of GRAF1 from neonatal rat ventricular cardiomyocytes (NRVCMs; Fig. S1d, e) led to mitochondrial dysfunction as indicated by increased mitochondrial membrane depolarization (reduced red: green ratio of JC-1 staining; Fig. 1d, e), and reduced respiratory capacity (assessed by Seahorse stress test; Fig. 1f, g).

To confirm a cardiomyocyte-autonomous role for GRAF1 in maintaining mitochondrial homeostasis, we developed a mouse line using 'knock-out-first' targeting vectors from Eucomm to conditionally target the GRAF1 allele. Southern blot analysis confirmed successful targeting in embryonic stem cells (ES) and germline transmission chimeras (Tm1a), which were subsequently crossed with Flp recombinase mice to remove a neomycin resistance cassette used for ES cell selection. We next established a cardiomyocyte-specific GRAF1 knockout mouse line (GRAF1^CKO) by crossing GRAF1 Tm1c (floxed) mice with the αMHC-MerCreMer (MCM) line (*JAX* line 005657) which led to a significant reduction in cardiac GRAF1 following tamoxifen treatment. As shown in Fig. 1h–j and Table S1, cardiac physiology was maintained in adult GRAF1^CKO mice following acute GRAF1 depletion. However, treatment with isoproterenol (ISO), an α-adrenergic agonist that increases cardiac energy demand[36], provoked dysfunction in GRAF1^CKO mice, indicating that GRAF1 deficiency confers susceptibility to cardiac stress. Notably, dysfunction was not accounted for by changes in compensatory hypertrophy as no significant differences were observed in cardiomyocyte cross-sectional areas (Fig. S1f, g).

### GRAF1 facilitates mitochondrial quality control
Given the accumulation of small, dysfunctional mitochondria in GRAF1-deficient hearts and the lack of adaptation of GRAF1^CKO mice to an ISO challenge, we reasoned that GRAF1 might play an important role in regulating mitochondrial fission and/or mitochondrial quality control. In support of the latter, GRAF1 depletion attenuated levels of activated LC3 as assessed by turnover of autophagosome-localized, cleaved, and lipidated form (LC3-II) in the presence and absence of the lysosome inhibitor, chloroquine (Fig. S1h, i). Activated LC3 is a protein marker reliably associated with completed autophagosomes[37] and our findings are consistent with a prior study which reported that GRAF1 promotes LC3II production and autophagosome number in Hela cells[38]. Importantly, LC3II/LC3I ratios were also significantly lower in saline and ISO treated GRAF1^CKO hearts relative to similarly treated controls (Fig. 1k, l). Moreover, we quantified the presence of mitochondria-associated autophagic vacuoles in hearts from ISO-treated wild-type (wt) and GRAF1^CKO mice by TEM, and as shown in representative Fig. 1m, we observed significantly fewer mitophagosomes per unit area in GRAF1^CKO hearts. Additionally, we observed increased mitochondria-associated glycogen accumulation (Fig. 1m, Fig. S1j), consistent with defects in autophagy and/or inefficient utilization of glycogen as a substrate. Taken together, these findings indicate that GRAF1 facilitates cardiac autophagy under both physiological and high energy demand situations.

LC3-mediated, autophagosome-dependent clearance processes can be non-selective (which generally occurs in response to energy starvation and randomly engulfs various organelles) or selective which typically occurs in response to changing environmental conditions and

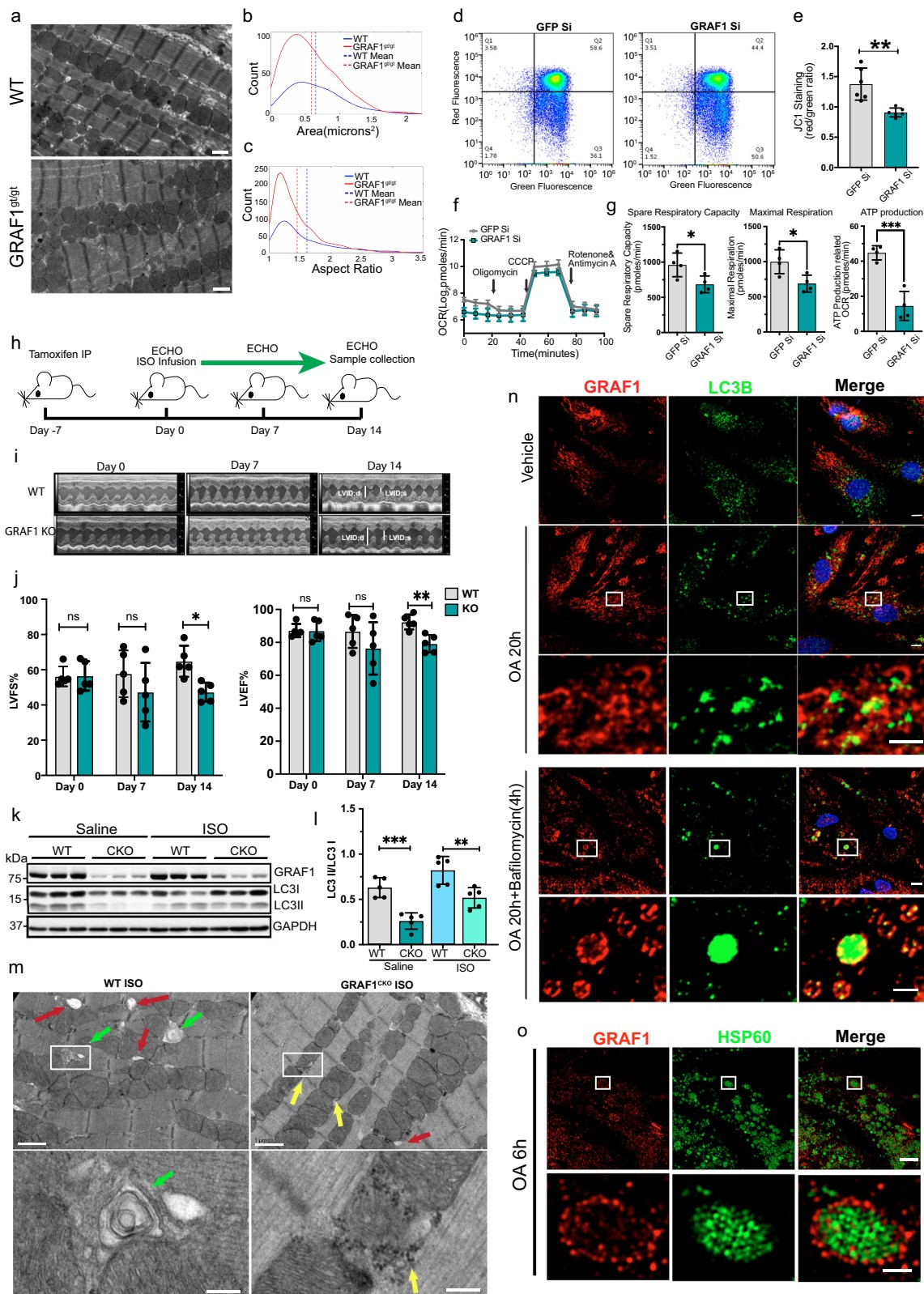

results in the elimination of specific damaged organelles. To distinguish between these processes, we next examined levels of p62 (also known as Sequestosome 1, or SQSTM1) a substrate degraded by general autophagy. As shown in Fig. S1h, i, p62 did not accumulate in GRAF1-deficient NRVCMs. Instead, GRAF1-deficient NRVCM exhibited higher levels of cleaved Caspase 3, indicative of increased presence of dysfunctional mitochondria.

To further explore a role for GRAF1 in mitochondrial-selective autophagy, we next used a well-characterized model that involves subjecting cells to a treatment with oligomycin-A and antimycin-A (OA) to induce respiratory-failure mediated mitochondrial damage. As shown in Fig. S1k, l, OA treatment led to a significant accumulation of inner mitochondrial membrane and mitochondrial matrix proteins in GRAF1-deficient NRVCMs, suggesting that GRAF1 is necessary for

**Fig. 1 | GRAF1 is essential for cardiac mitochondrial homeostasis.**
**a**–**c** Representative TEM and histogram analysis of mitochondrial size and aspect ratio in GRAF1gt/gt hearts relative to genetic control (WT). **b** Nonparametric kernel-smoothing distributions of mitochondrial cross-sectional areas (CSAs) from TEMs of hearts from wild-type (blue solid line) and GRAF1gt/gt (red solid line) mice. Mean mitochondrial CSA for wild-type (blue dashed line, $n = 330$) = $0.6516\,\mu m^2$, mean CSA for GRAF1gt/gt (red dashed line, $n = 714$) = $0.5876\,\mu m^2$, p-value (unpaired two-tailed t-test) = 0.0145. **c** Nonparametric kernel-smoothing distributions of mitochondrial aspect ratios (ARs) for TEMs of hearts from wild-type (blue solid line) and GRAF1gt/gt (red solid line) mice. Mean mitochondrial AR from wild-type (blue dashed line, $n = 330$) = 1.610, mean AR for GRAF1gt/gt (red dashed line, $n = 714$) = 1.452, p-value (unpaired two-tailed t-test) < 0.0001. **d**, **e** Quantification of mitochondrial membrane potential(red/green ratio) in NRVCMs by JC-1 staining; $n = 6$, p-value = 0.0064. **f**, **g** Quantification of mitochondria respiratory functions in NRVCMs by seahorse XF assay; $n = 4$, P-value(spare capacity) = 0.036, P-value(maximal respiration) = 0.024, P-value(ATP production OCR) = 0.0006. **h** Schematic of adrenergic challenge in WT and GRAF1CKO mice; $n = 5$ mice/group. **i** Representative left ventricular M-mode echocardiographic tracings. **j** Quantified Left Ventricle Fractional Shortening (LVFS) and Left Ventricle Ejection Fraction (LVEF).
**k**, **l** Representative Western blot and densitometric quantification of GRAF1 and LC3 II/LC3 I ratio in hearts from saline or ISO-treated WT and GRAF1CKO mice ($n = 5$/group). **m** Representative TEM of hearts from ISO-treated WT and GRAF1CKO mice. Green arrows show mitochondria in autophagosomes, red arrows highlight the attachment of small autophagosomes to mitochondria, yellow arrows indicate glycogen accumulation in proximity to mitochondria. Scale bar: $1\,\mu m$ or $0.2\,\mu m$ (magnified images). **n** Representative confocal image of endogenous GRAF1 and LC3B in NRVCMs treated with vehicle, OA and OA + bafilomycin. Scale bars: $5\,\mu m$ or $2\,\mu m$ (magnified images). **o** Representative superresolution-structured illumination microscopy (SR-SIM) of NRVCMs treated with OA and stained for GRAF1 and HSP60. Scale bar: $5\,\mu m$ (top), $1\,\mu m$ (bottom). Data are presented as mean ± SD; ns, not significant; *$P < 0.05$; **$P < 0.01$, ***$P < 0.001$ by two-tailed student's t-test (e,g,l) or Two-way ANOVA plus Sidak's multiple-comparisons test (**j**).

maintaining homeostatic control of mitochondrial biogenesis and clearance. Mitofusin 1, which is known to be degraded by the ubiquitin-proteosome system following mitochondrial damage, was degraded to a similar extent in control and GRAF1-deficient cells, indicating that mitochondria were equally damaged by this treatment (Fig. S1k). PGC1α, a transcriptional co-activator and central mediator of mitochondrial biogenesis was significantly reduced (not increased) in GRAF1 deficient NRVCMs (Fig. S1m). Taken together, these data indicate that the increase in mitochondria in GRAF1 deficient cardiomyocytes is likely due to failed mitochondrial clearance. In strong support of a role for GRAF1 in mediating cardiomyocyte mitophagy, using a combination of confocal and structured illumination microscopy in NRVCM, we showed that OA promoted the recruitment of endogenous GRAF1 to circularized structures that co-labeled with Parkin (Fig. S1n, o), the outer mitochondrial membrane marker, Tomm20 (Fig. S1p, q), and LC3 (Fig. 1n; a phenotype particularly obvious when cells were treated with bafilomycin A to inhibit lysosome activity), and surround the mitochondrial matrix as marked by HSP60 (Fig. 1o).

## GRAF1 promotes Parkin-mediated recruitment of autophagosomes to damaged mitochondria

To explore a direct role for GRAF1 in mediating mitophagy, we next turned to a well-validated Hela/YFP-Parkin cell line in which stable expression of the E3 ubiquitin ligase Parkin enables marked mitochondrial clearance within 24 hr following OA or CCCP-induced mitochondrial dysfunction[39]. Importantly, GRAF1 depletion by siRNA treatment (or by CRISPR-mediated knock-out) significantly mitigated OA-induced mitochondrial turnover in Hela/YFP-Parkin cells as assessed either by stabilization of mitochondrial inner membrane and matrix protein levels by Western blotting or by mitochondrial staining, suggesting that GRAF1 plays a critical role in Parkin-mediated mitophagy (Fig. 2a–c, Fig. S2a–g). As further confirmation that this model involves autophagosome-dependent clearance, we showed that, like depletion of GRAF1, depletion of the core autophagosomal elongation protein, autophagy related gene 12 (ATG12[40]) fully attenuated OA-mediated degradation of inner membrane and matrix proteins (but not mitofusin 1) in YFP/Parkin Hela cells(Fig. S2a, b). Moreover, Atg12 depletion led to a marked increase in GRAF1 levels indicating that endogenous GRAF1 may be degraded in an autophagy-dependent fashion (Fig. S2a, b). Providing additional confirmation of GRAF1's role in Parkin-mediated mitophagy, while GRAF1 deficiency suppressed mitochondrial clearance in Hela cells expressing YFP-Parkin when subjected to OA treatment, GRAF1 deficiency did not impact the degradation of mitochondria in the parent, Parkin-deficient sHela cells (Fig. S2c and d). This observation reinforces the notion that GRAF1 plays a crucial role in the context of Parkin-dependent mitophagy triggered by OA treatment. Moreover, while OA treatment promoted significant degradation of mitochondrial inner membrane and matrix

proteins in YFP/Parkin- Hela cells, it led to a slight but significant increase in levels of P62 (Fig. S2e), a protein typically targeted for degradation within autolysosomes, serving as an indicator of autophagic degradation activity. Notably, P62 also functions as an adapter protein for the clearance of mitochondria. This finding suggests that under the conditions of OA-induced mitophagy, the general autophagy machinery undergoes a shift from the conventional autophagy pathway to a selective clearance pathway, specifically aimed at degrading damaged mitochondria.

To begin to understand how GRAF1 might facilitate mitophagy, we first undertook an affinity purification-mass spectrometry approach to identify mitophagy-dependent GRAF1 binding proteins. Surprisingly, Parkin itself was identified as a GRAF1 binding partner in OA-treated cells as assessed by LC-MS/MS (16.8% peptide coverage). In support of this possibility, using confocal imaging we found that, like in cardiomyocytes, OA treatment promoted the recruitment of endogenous GRAF1 to the Tomm20-labeled OMM of Parkin-associated mitochondria (Fig. 2d, Fig. S2h). Moreover, reciprocal co-IPs (Fig. 2e, f; Fig. S2i) confirmed a physical association between GRAF1 and Parkin. Using a standard deletion analysis, we proceeded to map the GRAF1 interaction site on Parkin to two strings of highly conserved residues (aa101-110 and aa116-123) within a linker region located between the N-terminal ubiquitin-like domain (UBL) and the RING-between-RING (RBR) super-domain that is only fully exposed upon Parkin activation (Fig. 2f)[41,42]. Notably, reduced GRAF1 association was observed with both the ubiquitination-defective Parkin variant (C431S) and the linker variant (hereafter referred to as Δ18; Fig. S2j). Since Δ18 is equally modified by ubiquitination, these data indicate that it is the exposure of this activation-induced linker, not Parkin ubiquitination that is critical for GRAF1 binding. To our knowledge, GRAF1 is the only protein identified to date to interact within this activation-dependent domain.

In keeping with the idea that GRAF1 acts downstream of Parkin to facilitate mitophagy, we found that Parkin, GRAF1 and LC3 form a complex in cells upon Parkin activation by PINK1 (Fig. S2k) and that GRAF1 depletion significantly reduced LC3's interactions with Parkin, suggesting that GRAF1 may function to target autophagosomes to damaged mitochondria (Fig. 2g). To further explore this possibility, we employed a validated mitophagy reporter cell line that co-expresses mCherry-Parkin and GFP-LC3. As shown in Fig. 2h and i, GFP-LC3 puncta representing autophagosomes and Tomm20 labeled mitochondria robustly co-localized within 6 h of OA treatment in control cells, indicative of association of damaged mitochondria with autophagosomes into so-named "mitophagosomes". However, mitochondria-autophagosome associations were significantly reduced in similarly treated GRAF1 deficient cells. Collectively, these findings indicate that GRAF1 promotes mitophagy by driving mitophagosome production.

We also noted that peri-nuclear clustering of damaged (Parkin-labeled) mitochondria was reduced in GRAF1 deficient cells relative

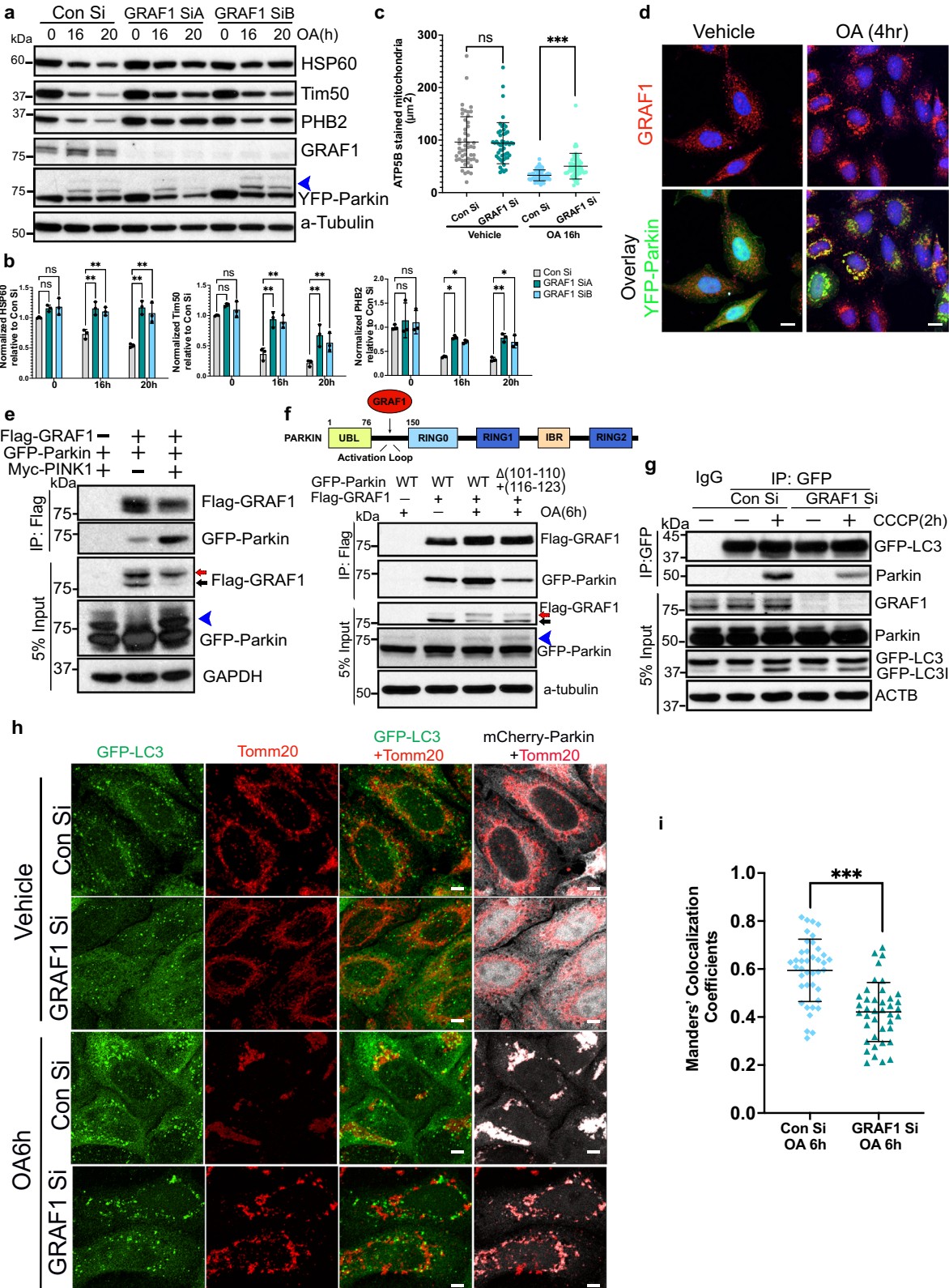

to controls 6 hr following OA treatment (Fig. 2h). While some mitochondrial clustering does occur in GRAF1 deficient cells after 16 hr OA treatment as assessed by ATP5B staining (Fig. S2f) or TEM (Fig. S2l), the extent of clustering was still diminished relative to similarly treated control cells. Thus, GRAF1's presence makes peri-nuclear clustering more efficient. Moreover, consistent with the idea that peri-nuclear mitochondrial clustering facilitates delivery

of damaged mitochondria to lysosomes, the association between mitochondria and LAMP1-labeled lysosomes was markedly reduced in OA-treated GRAF1 deficient Hela/Parkin cells relative to similarly treated controls (Fig. S2m). Collectively, these findings indicate that GRAF1 acts downstream of Parkin to promote autophagosome capture and to facilitate trafficking events that enhance mitochondrial clearance.

**Fig. 2 | GRAF1 promotes Parkin-dependent mitophagy in Hela cells by facilitating autophagosome capture. a, b** siRNA transfected Hela/YFP-Parkin cells were treated with OA for indicated times and levels of mitochondrial proteins were detected and quantified by Western blot/densitometry (**b**) $n = 3$ independent experiments. **c** Quantification of mitochondria in Hela/YFP-Parkin cells as assessed by ATP5B staining ($n = 50$ cells from 3 independent experiments; see Fig. S2f for representative images). **d** Representative confocal image of endogenous GRAF1 and YFP-Parkin in Hela/YFP-Parkin cells following OA treatment, scale bar: 2.5 μm. **e** Co-IP of GFP-Parkin with Flag-GRAF1 in COS7 cells co-transfected with Myc-PINK1 or empty vector. **f** Top: Schematic of GRAF1 binding site within Parkin activation linker. Bottom: Co-IP of GFP-Parkin WT and Δ18 variant (deletion of aa 101-110 and 116-123) with Flag-GRAF1 in transfected COS7 cells treated with OA or vehicle. Red arrow(**e, f**) denotes GRAF1 band shift. Blue arrowhead (**a–e, f**) indicates ubiquitinated GFP-Parkin. **g** Co-IP of Parkin with GFP-LC3 in GRAF1-depleted Hela cells stably expressing Parkin and GFP-LC3. **h, i** Representative confocal microscopy images of Tomm20, GFP-LC3 and mCherry-Parkin in Hela cells stably expressing GFP-LC3 and mCherry-Parkin. Z stack images were acquired and deconvolution images are presented as maximum intensity z-projection Scale bar: 5 μm. Fraction of GFP-LC3 puncta that overlaps with Tomm20 labeled mitochondria in OA treated cells was quantified by Manders' colocalization coefficient in 40 cells from 3 independent experiments (**i**). Data are presented as mean ± SD. ns not significant; *$P < 0.05$; **$P < 0.01$; ***$P < 0.001$ by two-tailed student's $t$-test (**c, i**) or by Two-way ANOVA with Dunnett's multiple comparison test (**b**).

## GRAF1 is phosphorylated in a PINK1-dependent fashion

In further support of a direct role for GRAF1 in mitophagy, we noticed that co-expression of Parkin and PINK1 or treatment with mitochondrial poisons induced a mobility shift in GRAF1 that is indicative of a post-translational modification (Fig. 2e, f; see red arrowheads on input lanes). Subsequent cell fractionation studies revealed that the modified GRAF1 form localized almost exclusively to mitochondria in CCCP-treated cells (Fig. 3a, see Fig. S3a for quantification). Further studies in Hela/YFP-Parkin cells revealed that an OA-dependent mobility shift was apparent at 1 h and peaked by 6 h following treatment (Fig. 3b), a time course consistent with clustering of depolarized mitochondria[20]. Accordingly, this modification was induced to a much greater extent in conditions that promote Parkin dependent mitophagy (i.e. OA treatment) compared to those that promote Parkin-independent macroautophagy (i.e. serum and glucose deprivation; Fig. S3b). A requirement for active Parkin was confirmed by comparing GRAF1 mobility in lysates from OA-treated sHela cells (that are Parkin-deficient) with the YFP-Parkin stable expressing HeLa cell line (Fig. S3c). Since this mobility shift was even more apparent when a phosphate-trapping electrophoresis method was applied (Phos-tag) and was largely reversed by treatment with calf-intestinal phosphatase, we reasoned these modifications were most likely due to phosphorylation (Fig. 3b).

We initially applied a proteomic approach to identify mitophagy-dependent GRAF1 modifications. Unfortunately, while we did detect several phosphorylated GRAF1 peptides, no phosphopeptides were recovered that exhibited significant PINK1/Parkin−dependent phosphorylation. Upon closer evaluation of the 44% peptide coverage observed, we noticed lack of coverage of the serine/proline (SP)-rich region of GRAF1. As further in silico approaches indicated lack of reasonable protease cleavage sites within this region, we proceeded with a standard deletion approach that focused on this area. As shown in Fig. 3c and d, deletion of amino acids 658-703 nearly completely mitigated OA-dependent GRAF1 phosphorylation in Hela/YFP-Parkin cells. We next individually mutated the 20 serine or threonine residues in this domain to a phosphor-deficient alanine and explored the mobility of these variants in un-treated (Fig. S3d) and OA-treated conditions (Fig. 3e). Using this deconstructive approach, we identified three major sites of phosphorylation, S668, T670 and S671 each of which were evolutionarily invariant in GRAF1 (Fig. 3e, f). These amino acids were also conserved in other GRAF family members (albeit in different combinations) indicating the likely importance of this regulatory region. For example, T670 and S671 are conserved in human GRAF2, S668 and S671 in human GRAF3 and T670 was conserved in the related human protein Oligophrenin. We next developed a specific antibody that recognizes phosphorylation of GRAF1 on S668, T670, and/or S671 and confirmed that OA treatment induced GRAF1 phosphorylaiton of these sites (Fig. S3e).

We also used this tool to confirm that phosphorylated GRAF1 selectively accumulates on damaged mitochondria. To this end, we developed a sophisticated APEX2-based enzyme-catalyzed proximity biotinylation system[43]. This method involves the fusion of a peroxidase enzyme to a bait protein (in this case an OMM targeting domain) that efficiently biotinylates proteins within a 10 nm radius following addition of biotin-phenol (BP) and hydrogen peroxide ($H_2O_2$). Biotinylated proteins can then be purified (and quantified) following precipitation with streptavidin-labeled beads. As shown in Fig. S3f, immunoblots of streptavidin precipitates from vehicle-treated APEX2-OMM/Hela-Parkin cells stimulated with $H_2O_2$ and BP for 30 s, revealed a small but detectible level of GRAF1. But, consistent with our cell fractionation studies, significantly more GRAF1 was apparent in streptavidin precipitates from APEX2-OMM/Hela-Parkin cells treated with OA for 2 hr prior to 30 s stimulation with $H_2O_2$ and BP. Moreover, immunoblotting with GRAF1 p668/670/671 Ab and Parkin revealed mitochondrial-exclusive biotinylation of phosphorylated GRAF1 and active (polyubiquitinated) Parkin in OA/$H_2O_2$/BP-treated cells. These data confirm that phosphorylated GRAF1 accumulates on mitochondria in a depolarization-dependent fashion.

Given the accumulation of phosphorylated GRAF1 on damaged mitochondria, we next explored the exciting possibility that GRAF1 might be one of a select few bona fide PINK1 substrates. In support of this likelihood, siRNA-mediated depletion of PINK1 reduced OA-stimulated phosphorylation of endogenous GRAF1 in Hela/YFP-Parkin as assessed by both a change in mobility on SDS-PAGE and Phos-tag gels and by intensity of detection with our phosphorylation-specific Ab (Fig. 3g). Importantly, the capacity of active PINK1 to directly phosphorylate full length GRAF1 was confirmed using in vitro kinase assays with purified PINK1 and GRAF1 fusion proteins (Fig. 3h). Notably, mutation of either S668 or S671 to alanine significantly reduced PINK1-mediated phosphorylation of GRAF1. Collectively, these studies provide strong support for GRAF1 as a substrate for PINK1 itself and/or additional PINK1/Parkin-dependent kinases.

## PINK1-dependent phosphorylation of GRAF1 promotes mitochondrial WAVE2 complex formation and mitochondrial clustering

As noted above, one of the first defects observed in GRAF1 deficient cells was a lack of perinuclear clustering following mitochondrial depolarization. Of major importance, WT GRAF1, but not a 668/670/671 triple alanine GRAF1 variant (hereafter referred to as GRAF1^AAA), restored OA-induced perinuclear mitochondrial clustering and degradation in GRAF1-deficient Hela/YFP-Parkin cells. This finding indicated an important functional consequence of PINK1-dependent phosphorylation (Fig. 3i–k). We next used LC-MS/MS to identify proteins that differentially IP'd with WT GRAF1 versus GRAF1^AAA in Hela/YFP-Parkin cells following OA treatment (ProteomeXchange ID PXD043212). Given the impact of GRAF1 phosphorylation on mitochondrial clustering, we focused further efforts on proteins that might be expected to impact mitochondrial trafficking. One such protein, Abelson-interactor adaptor-2 (ABI2) was significantly enriched in WT versus GRAF1^AAA IPs. This was intriguing because ABI2 is known to play a role in the positioning and activation of the Wiskott−Aldrich syndrome protein family member, WAVE2 a major regulator of Arp2/3-dependent actin nucleation that was previously implicated in

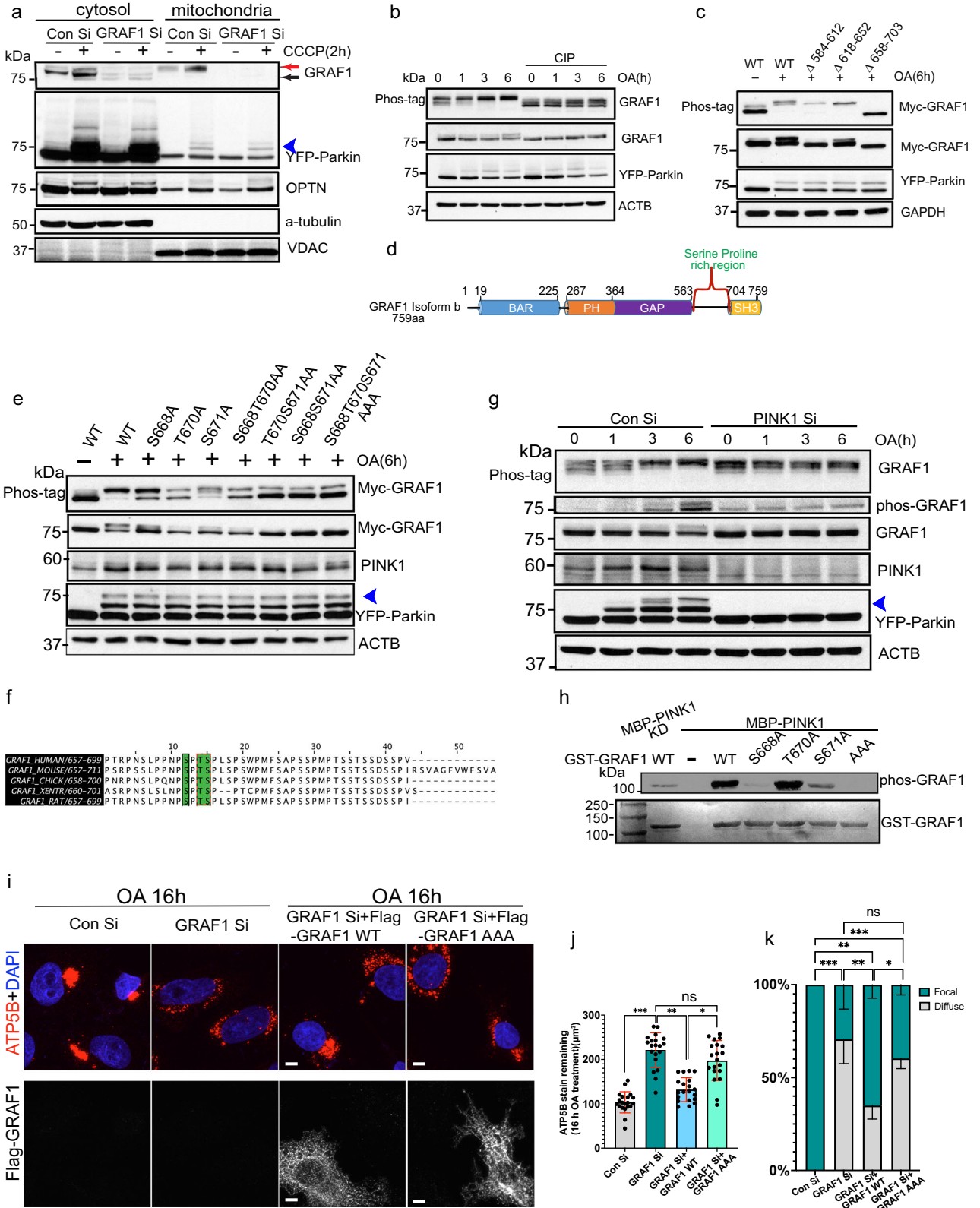

promoting starvation-induced autophagy in plants[21]. Confirmatory studies revealed that ABI2 co-IPd with GRAF1 in an OA-dependent fashion and revealed that this interaction was dependent on GRAF1 phosphorylation (note reduced binding of the AAA variant; Fig. 4a). Moreover, in strong support of a functional consequence of this interaction, GRAF1 depletion markedly reduced ABI2:WAVE2 complex formation (Fig. 4b see Fig. S3g for quantification).

Using GST-fusion precipitation, we further confirmed a direct association between ABI2 and GRAF1 and mapped the interaction site to the GRAF1 SH3 domain (Fig. S3h). While ABI2 strongly interacted with the isolated SH3 domain, it failed to interact with a construct that also contained the upstream serine/proline (SP) rich domain that harbors PINK1-dependent phosphorylation sites (Fig. S3h; Fig. 4c). Since these phosphosites are located within a canonical PXXP SH3

**Fig. 3 | PINK1-dependent phosphorylation of GRAF1 promotes retrograde mitochondrial trafficking and clustering. a** siRNA transfected Hela/YFP-Parkin cells were treated with CCCP or vehicle. Target proteins in cytosolic and mito-chondrial enriched fractions were analyzed by Western blot(see Fig. S3a for quantification). Red arrow denotes GRAF1 band shift. **b** Cell lysates from OA- and vehichle-treated Hela/YFP-Parkin cells were treated with calf intestinal alkaline phosphatase (CIP) or vehicle prior to Phos-tag gel and Western blot analysis. **c** Hela/YFP-Parkin cells expressing GRAF1 WT and deletion constructs were treated with OA and analyzed by Phos-tag gel and Western blot. **d** Schematic showing GRAF1b domain structure. **e** Hela/YFP-Parkin expressing GRAF1 WT and phosphor-deficient variants were treated with OA and analyzed by Phos-tag gel and Western blot. **f** Species alignment of mitophagy-associated GRAF1 phospho-sites. **g** siRNA transfected Hela/YFP-Parkin cells were treated with OA and GRAF1 phosphorylation was examined by Phos-tag gel and pS668/T670/S671 GRAF1 Ab (pGRAF1). Blue arrowhead (**a–g**) indicates ubiquitinated YFP-Parkin. **h** Top: purified GST-GRAF1 WT and indicated variants were incubated with WT or kinase dead (KD) MBP-PINK1. GRAF1 phosphorylation was examined by pGRAF1 Ab. AAA: S668A/T670A/S671A. Bottom: Ponceau S staining of GST-GRAF1 input. **l, j, k** 48 h post transfection of siRNA targeting GRAF1, Hela/YFP-Parkin cells were transduced with lentivirus expressing GRAF1 siRNA resistant Flag-GRAF1 WT or AAA variant. 24 h later cells were treated with OA. Z stack images were acquired and deconvolution images are presented as maximum intensity z-projection (**i**). Mitochondria were quantified by ATP5B staining. n = 20 cells/condition from 3 independent experiments (**j**). Mito-chondria clustering phenotypes were quantified, the statistical significance was calculated by focal perinuclear clustering, n = 20 cells/condition from 3 independent experiments (**k**). Unless otherwise indicated all Western blots are representative of at least 3 independent experiments. Data are presented as mean ± SD. ns, not significant, *P < 0.05; **P < 0.01; ***P < 0.001 by one-way ANOVA with post-hoc Tukey test (**j, k**).

binding motif, we reasoned that phosphorylation of these sites might relieve an autoinhibitory interaction between the GRAF1 SP and SH3 domains. In strong support of this idea, mutation of S668/T670/S671 to phosphorylation mimetic DDD or EEE variants rendered the GRAF1 SP-SH3 fusion protein capable of ABI2 binding (Fig. 4c). Moreover, co-IP experiments revealed that the full length phosphomimetic variant (GRAF1DDD) strongly interacted with ABI2 even in the absence of OA treatment (Fig. 4d), further confirming the function of PINK1-dependent phosphorylations.

### GRAF1 facilitates actin remodeling to promote PINK1/Parkin-dependent mitophagy
There is a growing realization of the importance of actin cytoskeletal remodeling in macro- and selective autophagy[21,44]. Generally, RhoA dependent F-actin cables limit mitophagy, while Arp2/3-dependent branched actin polymers facilitate the process by promoting autophagosome formation and lysosomal transport. However, it is not currently clear how these processes are controlled in a spatial and temporal fashion following mitochondrial depolarization. Given that GRAF1 limits RhoA-dependent F-actin formation[28–33], and is dynamically recruited to depolarized mitochondria where it associates with an Arp2/3 regulator in a PINK1/Parkin-dependent fashion, we postulated that GRAF1 might function to interconnect actin remodeling with mitochondrial depolarization to facilitate mitophagy. In strong support of this possibility, GRAF1 depletion significantly reduced OA-dependent recruitment of the WAVE2 complex (WAVE2, ABI2 and CYFIP1) to mitochondria as assessed by subcellular fractionation (Fig. 4e, f). We next used super resolution microscopy (SR-SIM) and confocal imaging to monitor mitochondrial-associated actin following OA treatment. In control cells, OA treatment led to the dissolution of linear F-actin filaments and the formation of small actin bundles and this remodeling was temporally associated with mitochondrial clustering. In contrast, OA-mediated F-actin dissolution was attenuated in GRAF1 deficient cells and mitochondria failed to cluster and remained associated with peripheral linear actin bundles (Fig. 4g, h, Fig. S3i, see Fig. S3j for quantification). Treatment with the Arp2/3 inhibitor, CK666 partially phenocopied the impairment of F-actin remodeling and mitochondrial clustering (Fig. S3k), and the impairment of mitochondrial clearance (as assessed by HSP60 levels; Fig. S3l, m) observed in GRAF1 deficient cells. Treatment with the F-actin polymerization drug, latrunculin B (LatB), which limits stress fiber formation, significantly (although not completely) rescued mitochondrial clustering defects in GRAF1-deficient cells (Fig. 4i and j, Fig. S3n).

Collectively, these data support a model in which following mitochondrial depolarization, GRAF1 facilitates localized F-actin depolymerization which both releases mitochondria from F-actin cables and generates G-actin monomers for nascent polymerization. In turn, GRAF1 phosphorylation is induced in a PINK1-Parkin dependent fashion on damaged mitochondria which leads to ABI2 recruitment,

WAVE2 complex formation and Arp2/3-dependent branched actin formation that is necessary for mitochondrial clearance. Thus, GRAF1 is distinctive from other mitophagy receptors because it facilitates mitophagy by serving not only as a Parkin to autophagosome scaffold, but also as an enzyme that promotes local mitochondria-associated actin remodeling (see Fig. 4k).

### GRAF1CKO hearts exhibit impaired metabolic flexibility following isoproterenol challenge
Given that cardiomyocytes exhibit a unique cytoskeletal and mito-chondrial organization, we sought to confirm that GRAF1 also promotes mitochondrial clearance in a PINK1/Parkin/LC3 and actin remodeling-dependent fashion in these cells. In support of this possibility, OA treatment induced rapid PINK1-dependent phosphorylation of GRAF1 on 668/670/671 in isolated cardiomyocytes (Fig. 5a–d). Furthermore, OA treatment promoted the recruitment of endogenous GRAF1 to mitochondria that were associated with branched actin filaments (Fig. 5e). Importantly, OA−induced actin remodeling and mitochondrial circularization and clustering were all significantly reduced in GRAF1-deficient cardiomyocytes (Fig. 5f, g).

Mounting evidence in the field indicates that mitophagy and mitochondrial biogenesis are intrinsically linked (i.e. mitophagy promotes mitochondria biogenesis via PINK/Parkin-mediated degradation of a powerful PGC-1α transcriptional repressor, PARIS)[45]. Thus, we next sought to test the extent to which GRAF1 was necessary for a mitophagy-dependent homeostatic circuit in intact hearts. To this end, we subjected WT and GRAF1gt/gt mice to hyperoxia- a stress that loosely mimics the changes in oxygen exposure that occurs at birth and has been shown to promote mitophagy-dependent mitochondrial biogenesis[46]. As shown in Fig. 5h, hyperoxia induced mitochondrial biogenesis in WT hearts as assessed by increased expression of PGC1α target genes that regulate mitochondrial transcription (PPARγ), the TCA cycle (citrate synthase), and OXPHOS (COXIV). In contrast, hyperoxia treatment did not significantly alter expression of these genes in GRAF1gt/gt hearts. These data further substantiate a role for GRAF1 in promoting mitochondrial quality control in the myocardium.

Consistent with prior studies that ISO treatment also induces cardiac mitophagy[36], acute treatment of mice with ISO also led to pronounced phosphorylation of GRAF1 (Fig. 5i, j). To further confirm that GRAF1 plays a crucial role in adult cardiomyocyte mitophagy, we performed additional experiments using the powerful mt-Keima mouse model. Mt-Keima is a pH sensitive florescent protein that is targeted to the mitochondrial matrix by the N-terminal targeting signal of COX8A and has been validated as a rigorous and reproducible reporter to track PINK1/Parkin-dependent mitophagy in vivo[47]. Using this model system, we show that hearts from ISO-treated mt-Keima/GRAF1CKO mice have significantly lower levels of mitophagy than hearts from similarly treated mt-Keima/GRAF1+/+ mice (Fig. 5k, l). Thus, when taken together with evidence that mitophagy-dependent mitochondrial biogenesis

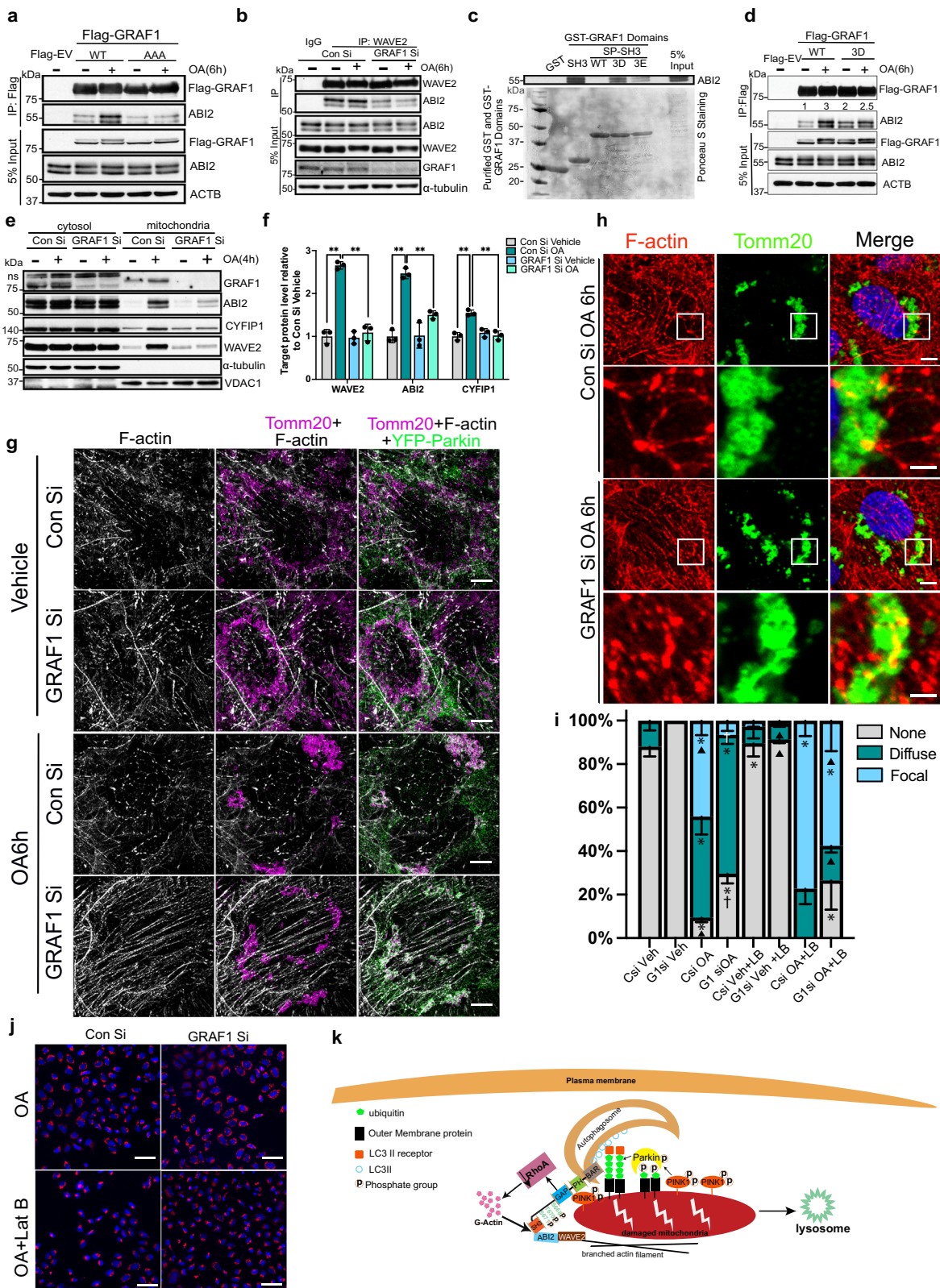

promotes a compensatory transition to glycolysis in response to a hemodynamic challenge, we posited that the cardiac dysfunction observed in ISO-treated GRAF1$^{CKO}$ mice might be due, at least in part, to variations in cardiac substrate metabolism. To test this possibility, we next applied unbiased metabolomics approaches using liquid chromatography coupled to high-resolution mass spectrometry (LC-HRMS)[48]. The raw metabolomic data used in this study are available in the

University of North Carolina Digital Repository under the accession code cr56nb66p. Metabolites from ISO treated WT and GRAF1$^{CKO}$ hearts stratified into two distinct clusters and subsequent KEGG pathway analysis identified 9 metabolic pathways that differed significantly between genotypes (Fig. 5m, n, Fig. S3o, Source Data File). Most of the pathways elevated in WT hearts were associated with carbohydrate metabolism and energy production, whereas most of the pathways

**Fig. 4 | PINK1 phosphorylation promotes GRAF1 SH3-domain dependent recruitment of the WAVE2 complex to damaged mitochondria and facilitates mitochondrial-associated actin remodeling. a** Co-IP of endogenous ABI2 with Flag-GRAF1 WT or phosphor-deficient AAA variant in GRAF1 KO Hela/YFP-Parkin cells. EV: empty vector. **b** Co-IP of endogenous ABI2 and endogenous WAVE2 in siRNA treated cells assessed by Western blot. See Fig. S3g for densitometric quantification, $n = 3$ independent experiments. **c** In vitro interaction of purified GST-GRAF1 domains with endogenous ABI2 in Hela/YFP-Parkin lysates. Bottom: Ponceau S staining of GST-fusion protein input. SP-SH3: serine proline (SP) region and SH3 domain. 3D and 3E: DDD and EEE phosphomimetic variants. **d** Co-IP of endogenous ABI2 with Flag-GRAF1 WT or DDD variant in GRAF1 KO Hela/YFP-Parkin cells. **e, f** WAVE2 complex proteins in cytosolic and mitochondrial enriched fractions of siRNA transfected Hela/YFP-Parkin cells were analyzed by Western blot (**e**) and quantified in (**f**) $n = 3$ indepdent experiments. ns denotes non-specific band. **g, h** Representative SR-SIM (**g**) or confocal microscopy (**h**) of Tomm20, Phalloidin and YFP-Parkin in siRNA transfected Hela/YFP-Parkin cells. Note continued

presence of F-actin bundles and anchored mitochondria in GRAF1-deficient cells following OA treatment, see Fig. S3i for j for quantification of branched actin formation in control and GRAF1-deficient cells following 6 hr OA treatment. **i** Quantification of mitochondria clusters in siRNA transfected Hela/YFP-Parkin cells with indicated treatments. Latrunculin B (LB), $n = 4$ independent experiments per condition. * Denotes $P < 0.05$; OA vs. DMSO for the same siRNA. † Denotes $P < 0.05$ siGRAF1 vs. siControl for the same treatment (±OA). Black traingle denotes $P < 0.05$ in LB vs. no LB for the same siRNA/±OA treatment. See Fig. S3n for mitochondrial clustering phenotypes. **j** Representative confocal image of ATP5B in siRNA transfected Hela/YFP-Parkin cells treated with OA in prescence or absence of Latrunculin B(Lat B). Scale bars: 50 μm. **k** Schematic showing GRAF1 phosphorylation mediates mitochondria-assoiced actin remodeling. Unless otherwise indicated all Western blots are representative of at least 3 independent experiments. Scale bars: 5 μm (**g,h**) and 2 μm(magnification in h). Data are represented as mean ± SD, ns, not significant, *$P < 0.05$, **$P < 0.01$ by two-tailed student's $t$-test (**i**) or by One-way ANOVA with post-hoc Tukey test (**f**).

elevated in GRAF1$^{CKO}$ hearts involved amino acid metabolism/catabolism (Fig. S3o). These findings indicate that GRAF1$^{CKO}$ hearts exhibited impaired ability to efficiently utilize carbohydrate as an alternative substrate for energy production, a finding consistent with our ultrastructural data that revealed notable accumulation of glycogen around mitochondria in GRAF1$^{CKO}$ hearts (see Fig. 1m). In support of these findings, similar (albeit more modest) differences were observed in serum metabolites isolated from the ISO-treated mice (e.g. ISO-treated WT mouse serum exhibited significantly higher levels of metabolites derived from fructose/mannose metabolism, the pentose phosphate pathway and glycolysis/gluconeogenesis), whereas none of these metabolites were significantly different between saline-treated WT and GRAF1$^{CKO}$ mice. Indeed, only 5 of the 224 metabolites detected in serum samples differed between saline treated animals, consistent with the notion that GRAF1-dependent processes are employed under conditions of high energy demand. The high levels of glycolytic intermediates observed in serum from WT animals is consistent with prior studies showing that cardiac stress promotes a change in fuel utilization in the myocardium from predominantly fatty acid consumption to glycolysis[49]. Instead, GRAF1$^{CKO}$ hearts had altered amino acid metabolism and produced lower levels of ATP, indicating a lack of metabolic flexibility—a process recently linked to mitochondrial quality control mechanisms including mitophagy[50].

Finally, in strong support of a critical role for GRAF1-dependent mitophagy in the pathological progression of human disease, we found that GRAF1 phosphorylation at PINK1/Parkin-dependent sites was dysregulated in cardiac samples from patients diagnosed with either hypertrophic or dilated cardiomyopathy (HCM or DCM) relative to age and sex matched controls (Fig. 5o, Fig. S3p, see Table S2 for individualized patient data). When taken together with the significant reduction in mitochondrial content (as assessed by Hsp60 levels) and total GRAF1 protein, these findings are in keeping with disruptions in mitochondrial homeostasis in HF. Collectively, these findings indicate that dysregulation of GRAF1 expression or activity may exacerbate heart failure and possibly other disorders that result from defects in PINK1/Parkin-mediated mitochondrial quality control.

## Discussion

Recent findings suggest that mitochondrial dysfunction is one of the overriding pathophysiological hallmarks of numerous cardiometabolic, skeletal-, and neuro-degenerative diseases[15,35,51,52]. Unfortunately, effective strategies to promote the clearance of damaged mitochondria are currently lacking. This is due, at least in part, to the fact that several proteins involved in this process have redundant functions and relatively few of the essential proteins identified to date have targetable enzymatic moieties.

Herein, we identified GRAF1 as a mitophagy regulator that plays a fundamental role in facilitating PINK1/Parkin-dependent clearance of

damaged mitochondria. Our data support a model in which mitochondrial membrane depolarization triggers partial GRAF1 translocation to the OMM of damaged mitochondria where it is phosphorylated by PINK1 and/or PINK1-dependent kinases. Phosphorylated GRAF1 then forms a physical interaction with active Parkin and promotes capture and encasement of damaged mitochondria by autophagosomes. Our temporal approaches suggest that GRAF1 does not impact the initial recruitment of Parkin or the autophagy receptor OPTN[20,52], but acts downstream as an enzyme that promotes localized actin remodeling to facilitate mitochondrial trafficking and clearance. The future application of high powered, real time imaging to track the fates of individual mitochondria and localized actin fibers during mitochondrial clearance will be necessary to provide further support for this conclusion and to shed light on additional unanswered questions in the field regarding the precise mechanisms that control the spatial and temporal clearance of damaged mitochondria in various cell types.

### Actin dynamics and mitochondrial homeostasis

Actin has been long known to regulate mitochondrial fission, fusion and trafficking, and more recent studies indicate that actin plays a wider role in mitochondrial clearance[21,22,44]. Cytoplasmic actin exists as monomeric units (G-actin) or in polymerized states that can be branched or linear (F-actin). F-actin filaments are formed by RhoA-dependent activation of mDia and the formin family of actin binding proteins while branched actin is formed by Rac1-dependent processes and the coordinated activity of the Actin-related protein 2/Actin-related protein 3 (Arp2/3) complex and various nucleation-promoting factors (NPFs), including WAVEs, WASP, WASH, and others. Recent studies have shown that during general autophagy, local assembly of Arp2/3-dependent branched actin filaments is necessary to support formation of nascent autophagosomes and for the expansion, transport, and fusion of the autophagsomes with lysosomes[23–25,53].

During mitophagy, Arp2/3- and myosin (MyoV1)- mediated actin remodeling has been further implicated in forming protective shells around dysfunctional mitochondria and in dispersing clumps of damaged mitochondria to facilitate subsequent autophagosome capture and clearance[26,53]. The prevailing model is that rapid formation of protective shells or cages can shield the cell from dangerous mitochondrial byproducts (i.e. ROS) and/or contents (i.e. cytochrome C, mitochondrial DNA) that are released by dysfunctional mitochondria and, at the same time, prevent re-fusion of the damaged mitochondria with their healthy neighbors until autophagosomes can be generated and recruited to clear these organelles. However, a number of important questions remain with respect to the formation of these structures such as: What is the mechanism by which Arp2/3 is activated following mitochondrial depolarization? Which NPFs are involved in mitochondrial-localized Arp2/3 activation? How are the actin shells removed or dissambled to allow encapsulation by the autophagosome?

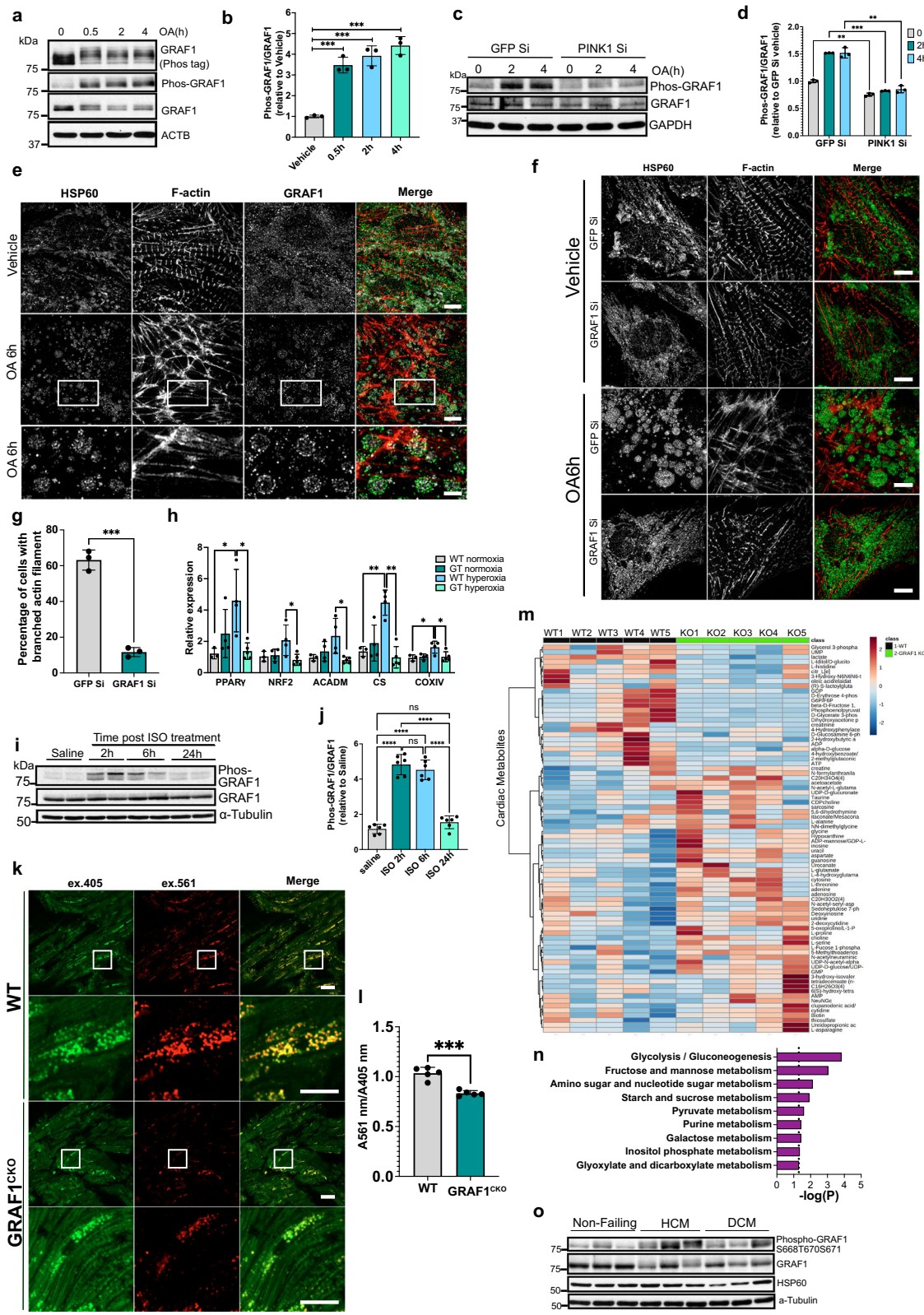

Mitochondrial clearance also requires precise spatiotemporal control of actin de-polymerization yet, again, the mechanisms that control these events are incompletely understood. Mitochondrial poisoning promotes rapid F-actin depolymerization, and the importance of this event is underscored by recent studies indicating that F-actin impedes mitophagy and exacerbates Parkinson's-like phenotypes in animal models. Indeed, Moskal et. al. identified Rho-kinase

(ROCK) inhibitors in a discovery-based screen for small molecule activators of mitophagy and further showed that ROCK inhibitors provided protection from paraquat-mediated neurodegeneration in flies. Likewise, Sarcar et. al. showed that neurodegeneration in transgenic α-synuclein expressing flies was associated with excess F-actin stabilization, and the degenerative phenotype was attenuated by co-expression of the actin severing protein, gelsolin[54].

**Fig. 5 | GRAF1 is required for stress-induced metabolic adaptation in adult hearts and is dysregulated in human heart disease. a, b** Endogenous GRAF1 phosphorylation in NRVCMs treated with OA and examined by Phos-tag gel and Western using GRAF1 phos-S668/T670/S671 Ab, and densitometric quantification (**b**) $n = 3$ independent experiments. **c, d** siRNA transfected NRVCMs were treated with OA and GRAF1 phosphorylation was examined using pS668/T670/S671 GRAF1 Ab. Densitometric quantification (**d**) $n = 3$ independent experiments. **e** Representative SR-SIM of NRVCMs treated with vehicle and OA and stained for GRAF1, HSP60 and Phalloidin. Scale bar: 5 μm (top, middle), 2 μm (bottom). Note OA-dependent recruitment of GRAF1 to circular structures surrounding mito-chondria are associated with branched actin. **f** Representative SR-SIM of HSP60 and Phalloidin stained siRNA transfected NRVCMs treated with OA or vehicle. Scale bar: 5 μm. **g** Quantification of branched actin formation in control and GRAF1-deficient NRVCMs following 6 hr OA treatment. ($n = 40$ cells from 3 independent experiments). **h** WT and GRAF1[gt/gt] mice were subjected to hyperoxia (100% O2) for 48 hr to induce mitophagy followed by 96 hr of normoxia. Transcript levels of target genes in hearts (WT normoxia mice($n = 3$), GRAF1[gt/gt] normoxia mice($n = 4$), WT

hyperoxia mice($n = 4$) and GRAF1[gt/gt] hyperoxia mice($n = 5$)) were assessed by real time PCR. **i, j** Endogenous GRAF1 phosphorylation in mouse hearts following ISO treatment (S.C, 200 mg/kg). $n = 6$ mice/group and densitometric quantification (**j**). **k, l** Representative confocal microscopy images of mt-Keima in the hearts from ISO treated WT and GRAF1[CKO] mice expressing mt-Keima.(S.C, single dosage of 200 mg/ kg for 3 days) $n = 5$ mice /group. Ratiometric fluorescence analysis of mt-Keima probe(l). Scale bar: 40 μm and 20 μm(magnified images). **m** Supervised hierarchical clustering heatmap showing top 80 changed metabolites between ISO treated WT and GRAF1[CKO] hearts. **n** Histogram of metabolic pathways enrichment using sta-tistically differed metabolites in Fig. S3o. Hypergeometric Test ($p < 0.05$) was used to determine the pathway enrichment. **o** Western blot analysis of GRAF1 phos-phorylation and mitochondria mass in human heart samples, see Fig. S3p for densitometric analysis. $n = 6$/group. HCM: hypertrophic cardiomyopathy. DCM: dilated cardiomyopathy. Data are represented as mean ± SD. ns, not significant; *$P < 0.05$, **$P < 0.01$, ***$P < 0.001$, ****$P < 0.0001$ by two-tailed student's $t$-test (**d, g, l**) or by One-way ANOVA with post-hoc Tukey test (**b, h, j**).

Our studies provide clues regarding the interconnection between mitochondrial membrane depolarization and actin remodeling. Firstly, consistent with prior studies from our lab and others demonstrating that GRAF1 is necessary for limiting RhoA mediated F-actin formation[29], we show that GRAF1 aids in promoting mitochondrial poison-induced F-actin de-polymerization which both releases mitochondria from F-actin cables to facilitate mitochondrial trafficking and generates G-actin monomers for nascent polymerization. In turn, GRAF1 is recruited to the OMM of damaged mitochondria, is phosphorylated by PINK1 and/or PINK1-dependent kinases, which facilitates WAVE2 complex formation and Arp2/3-dependent branched actin formation that is necessary for mitochondrial clearance. This model is supported by the similar kinetics of PINK1-dependent phosphorylation and the exclusive accumulation of fully phosphorylated GRAF1 on mitochondria. We provide strong evi-dence that PINK1-dependent phosphorylation of GRAF1 relieves auto-inhibition between the GRAF1 serine-proline rich and SH3 domains to facilitate ABI2/WAVE2/Arp2/3 recruitment and mitochondrial cluster-ing. Our findings that WT GRAF1 but not the triple phosphor-deficient GRAF1 variant can rescue mitochondrial clustering in GRAF1 deficient cells lends strong support to this model. However, it is formally possible that other GRAF1 SH3 binding partners and/or RhoA-dependent mechanisms are involved. Because GRAF1 also facilitates autophago-some recruitment, we posit that GRAF1 has the capacity to fine-tune actin polymerization/depolymerization in a spatial and temporal fash-ion to ensure the quick and efficient capture and clearance of damaged mitochondria. Recent studies in Atg7 knockout mice revealed defects in both autophagosome formation and actin assembly[21], further sup-porting an actin-mitophagy interconnection.

While our studies in Hela cells support a role for GRAF1 in pro-moting retrograde trafficking of damaged mitochondria, very recent studies suggest that some cells including neurons, brown adipocytes, and cardiomyocytes can also eliminate damaged mitochondria into the extracellular space by autophagy-dependent anterograde traf-ficking to the plasma membrane[21,55–57]. While these events likely also require coordinated membrane and cytoskeletal remodeling events, future studies are necessary to determine the relative impact of this clearance mechanism in the heart and the extent to which GRAF1 might be involved. Likewise, while our data strongly support the contention that GRAF1 selectively promotes PINK1-Parkin dependent mitophagy in settings in which mitochondrial membrane depolariza-tion or dysfunction is induced, it is formally possible that GRAF1 plays additional roles in general macroautophagy in conditions that favor this response, as suggested by a prior report[38].

### GRAF1 recruitment to damaged mitochondria
Under basal conditions in healthy muscle cells, GRAF1 physically interacts with FAK in actin-associated adhesion complexes[28] and we

now show for the first time that upon mitochondrial damage, GRAF1 redistributes to the OMM of Parkin and LC3-labeled mitochondria. While further studies are necessary to determine the precise mechanisms by which GRAF1 is recruited to damaged mitochondria, we propose a model in which the GRAF1 BAR-PH domain may mediate binding to depolarization-exposed phospholipids on the outer mito-chondrial membrane. Notably, a PH domain ligand, cardiolipin (CL) is rapidly transferred to the outer leaflet of the OMM by phospholipid scramblase 3, and its externalization is known to promote mitophagy through mechanisms that involve direct interactions with LC3[11]. Moreover, depolarization-mediated accumulation of additional phos-pholipids such as phosphatidyl serine, phosphatidyl ethanolamine and phosphatidic acid induce curvature of damaged mitochondrial membranes[11], which could drive BAR domain-mediated interactions. The lipid-bending BAR domain of GRAF1 may serve to further facilitate inward membrane curvature-dependent mitochondrial encasement of LC3B-containing autophagosomes[58,59]. Future studies are warranted to determine the extent to which these modifications might contribute to GRAF1 recruitment to and/or retention on damaged mitochondria.

### In vivo consequences and putative clinical relevance of GRAF1-dependent mitophagy
We provide strong evidence that GRAF1 is necessary for mediating PINK1/Parkin-dependent mitochondrial homeostasis in the heart fol-lowing an adrenergic challenge and that GRAF1 is dysregulated in human heart failure. These data are in keeping with our prior studies showing that co-depletion of GRAF1 and dystrophin[32] unmasked car-diac and skeletal muscle phenotypes that were associated with meta-bolic dysfunction. Under each of these conditions, GRAF1-deficient hearts exhibited widespread fibrosis, focal necrosis, inflammation, significantly decreased ejection fraction and fractional shortening, and ultrastructural mitochondrial defects[31,32]. Consistently, overall meta-bolic profiles from hearts and serum from ISO-treated GRAF1[CKO] mice are suggestive of impaired mitophagy-dependent transition to glyco-lysis and compensatory increase in amino acid metabolism. In this regard it is interesting to note that the phenotype of cardiac GRAF1-deficient mice mirrored cardiac ATG5-deficient mice in that there was no significant difference in cardiac function until mice were challenged[36]. Taken together, these studies suggests that autophagy/ mitophagy is not essential for basal cardiac function, but is necessary for situations in which there is a mismatch in energy supply and demand. This finding is in keeping with recent studies indicating that mitochondrial flexibility is necessary to maintain energy stability and muscle function in conditions with high demand to fuel ratios[60].

There are a few notable limitations of our in vivo studies. Firstly, it is formally possible that additional RhoA-dependent effects could contribute to cardiac dysfunction in our model. However, it is

important to note that depletion of GRAF1 leads to a modest and transient activation of RhoA[31–33] and studies from others indicate that moderate (~2·5 fold) overexpression of constitutively active RhoA does not negatively impact cardiac function and may even be cardioprotective under stressed conditions[61]. Secondly, we did not include female mice in this study. Based on our prior published studies that revealed no statistical differences in cardiac function in male and female GRAF1[gt/gt] mice[32], and studies from others showing lack of sexual dimorphism in isoproterenol-induced cardiac outcomes including heart rate, hypertrophy, dysfunction, and fibrosis, we would not expect a difference between males and female mice in this particular stress model[62], however, future studies will be necessary to confirm this thesis.

Based on our data showing that GRAF1 plays a unique role in promoting PINK1/Parkin-dependent mitophagy, we surmise that it could regulate the pathological progression of additional cardiometabolic and/or neurological diseases associated with dysfunctions in mitochondrial clearance. It will be of future interest to determine if, like Parkin, GRAF1 is also necessary for the metabolic maturation that is known to occur during early postnatal cardiac development[16] and for limiting ischemic stress-mediated injury in the adult myocardium[17–19]. With respect to neurodegenerative disease, it is notable that nearly all PD-associated mutations impact proteins that participate in mitochondrial quality control[52,63], and GRAF1 was shown by others to be neuroprotective in a human Parkinson's cell culture model (MPP + -treated dopaminergic neuroblastoma SK-N-SH cells)[64]. While related patient data are relatively scarce, 24 patients to date have been reported with GRAF1 autoimmunity, and of these cases, 83% of patients had cerebellar ataxia (a feature common in PD and related diseases), approximately 30% exhibited cerebellar atrophy and 1 patient had been diagnosed with early stage PD[65]. Thus, our studies provide a strong rationale for a more extensive exploration of the association between *Arhgap26* polymorphisms and susceptibility to heart failure, PD, and perhaps other diseases that involve metabolic dysfunction[66].

# Methods

## Inclusions and ethics statement

The research described adheres with all relevant ethical regulations. All experiments involving mice were approved by the UNC-Chapel Hill Institutional Animal Care and Use Committee under protocol 22-203. Mice were housed at temperatures of 20-23 °C with 40-60% humidity. Human samples are from subjects consented and collected for future research by the Duke Human Heart Repository (Pro00005621). A "Request for Waiver or Alteration of Consent and HIPAA Authorization" was submitted to the Duke IRB for the present study and a separate request for "Exemption from IRB Review" was submitted to the University of North Carolina IRB to allow these studies to be performed at UNC (21-2644). De-identified samples were obtained and data are presented without PHI. The following data were collected and presented per Duke's IRB protocol: average age of subject group at time of tissue collection, gender, medical history information (excluding rare diseases but including common concomitant medications), and common co-morbidities.

## Experimental model and subject details.

GRAF1 Knockout-first allele (with conditional potential; GRAF1[tm1a]) mice were generated at UNC animal model core facility using Arhgap26[tm1a] HEPD0821_3_D02 embryonic stem (ES) and HEPD0821_3_D03 ES cell clones (EuMMCR). All breeding and experimental mice were maintained on C57BL/6 J genetic background. GRAF1[tm1a] and flippase (Flp)/+ mice were crossed to generate GRAF1[tm1c/+](GRAF1[flox/+]) mice. GRAF1[flox/+] mice were crossed with hemizygous αMHC-MerCreMer(MCM) mice to generate αMHC-MCM/+: GRAF1[flox/+] that subsequently were crossed with hemizygous αMHC-MCM mice to generate αMHC-MCM/MCM: GRAF1[flox/+] mice.

Next, male αMHC-MCM/MCM:GRAF1[flox/+] mice were crossed with female GRAF1[flox/+] mice to generate hemizygous αMHC-MCM mice as littermate control and αMHC-MCM/+: GRAF1[flox/flox] as tamoxifen inducible cardiac-restricted knockout mice. Mt-Keima mice were a kind gift from Dr. Toren Finkel(University of Pittsburgh). Male αMHC-MCM/MCM:GRAF1[flox/+] mice were crossed with female mt-Keima: GRAF1[flox/+] mice to generate αMHC-MCM mice expressing mt-Keima as littermate controls, and αMHC-MCM/+: GRAF1[flox/flox] mice expressing mt-Keima for the experimental group. GRAF1 gene trap mice (GRAF1[GT/GT]) were previously generated using ES cells obtained from the Texas A&M Institute for Genomic Medicine (College Station, TX)[33]. Heterozygous GRAF1 gene trap mice (GRAF1[gt/+]) were crossed to generate WT and GRAF1[gt/gt] for experiment. Male mice (2–3 month old) and littermate controls were used in animal studies. GRAF1[flox/+] genotype primers: forward Primer 5′-GAGTAGCTGAGTGCTGTGCAAATGTATC-3′, reverse primer 5′-TCAGAATCCAACCTTTTTTAGGGGAG-3′. αMHC-MerCreMer genotype primers: forward primer 5′-ATACCGGAGATCATGCAAGC-3′, reverse primer 5′- AGGTGGACCTGATCATGGAG-3′. All mice were housed in pathogen-free facilities under a 12-h light/dark cycle with unrestricted access to food and water. Animals were treated in accordance with the approved protocol of the University of North Carolina (Chapel Hill, NC) Institutional Animal Care and Use Committee, which is in compliance with the standards outlined in the guide for the Care and Use of Laboratory Animals.

## Cell culture and transfection.

sHela (basal) cell, Hela cells stably expressing YFP-Parkin(Hela/YFP-Parkin) or mCherry-Parkin/GFP-LC3(Hela/mCherry-Parkin/GFP-LC3) were kind gifts from Dr. Richard Youle at the National Institutes of Health (NIH). Hela cells stably expressing Parkin/GFP-LC3(Hela/Parkin/GFP-LC3) were a kind gift from Dr. Beth Levine at University of Texas Southwestern Medical Center. COS7 cells and Hela cells were cultured in Dulbecco's modified Eagle's medium DMEM (Gibco) supplemented with 10% (v/v) FBS (Sigma) and 100 units penicillin/100 μg streptomycin (final concentration).

The plasmids containing the designed cDNA variants were generated by site-directed mutagenesis PCR (QuickChange, Agilent) or Gibson Cloning (New England BioLabs) following manufacturers' instructions. Plasmid was transfected into Hela or COS7 cells using Lipofectamine™ 2000 Transfection Reagent for 20 h followed by 2.5 μM oligomycin A and 250 nM antimycin A for 6 h to induce mitophagy. Hela cells were transfected with target siRNA for 72 h by Lipofectamine™ RNAiMAX Transfection Reagent. Primary neonatal rat ventricle cardiomyocytes (NRVCM) were transfected with target siRNA using Hiperfect transfection reagent (Qiagen) for 48 h followed by another transfection for 48 h. Hela cells were treated with vehicle (DMSO) or 10 μM oligomycin A(Sigma) and 5 μM antimycin A (Sigma) for short time points (up to 6 h) or 2.5 μM oligomycin A and 250 nM antimycin A for 16 or 20 h in fresh growth medium. NRVCM were treated with vehicle (DMSO) or 5 μM oligomycin A and 5 μM antimycin A for different periods of time as indicated.

## NRVCM isolation and culture.

NRVCMs were isolated from 2–3 day old Wistar rats following protocol of Neonatal Cardiomyocyte Isolation System (Worthington Biochemical Corporation, NJ). Briefly, 5-15 hearts were rinsed wtih 10 ml sterile calcium- and magnesium-free Hank's balanced salt solution (CMF HBSS) and minced to <1 mm³ pieces in Petri dish, then were digested with 10 ml CMF HBSS containing 50-100 μg/ml trypsin at 4 °C for 16–20 h. Next day, minced heart tissues in CMF HBSS were transferred to a 50 ml centrifuge tube using wide-mouth 10 ml serological pipet and supplemented with 1 ml 2 mg/ml Soybean Trypsin Inhibitor. After warming heart tissues and buffer to 30-37 °C in water bath, 300U/ml collagenase II in 5 ml Leibovitz L-15 medium was slowly added. Heart tissues were then gently rotated at 37 °C for 45 min. Next, with standard 10 ml plastic serological pipet,

minced heart tissues were gently triturated about 10 times and filtered through 70 μm cell strainer into a fresh 50 ml centrifuge tube. Filtered cells were undisturbed for 0.5–1 h at room temperature. Subsequently, cells were centrifuged at 100xg for 5 min and resuspended in 20 ml medium199(15%FBS, 1% penicillin/streptomycin) and cultured in 150 mm culture dish for 1 h in cell incubator, then cells in culture medium were transferred to a fresh 150 mm culture dish for another 1-h incubation. Next, NRVCMs in the culture medium were plated in fibronectin (30 μg/ml in PBS)-coated 100 mm dishes with Medium199(15%FBS, 1% penicillin/streptomycin and 100 μM BrdU). 24 h later, NRVCMs were washed with PBS twice and cultured in serum-free Medium199(1% penicillin/streptomycin) for 1–2 days followed by siRNA transfection or virus infection.

**Lentivirus packaging and transduction.** Lentivirus packaging was conducted in 293 T cells that were maintained in DMEM(Corning) growth media containing 10% FBS, 1% penicillin/streptomycin, 1x glutamine and 1x non-essential amino acids (Gibco). One day before transfection, 293 T cells were plated at a density of $12 \times 10^6$ cells/15 cm dish that had been coated by Poly-L-Lysine solution(10ug/ml) for 1 h at room temperature. The next day, 27 μg lentiviral transfer plasmid encoding gene of interest (target gene was cloned into pLenti CMV GFP, Addgene #17448), 13.85 μg psPAX2 (Addgene #12260) and 13.85 μg pMD2.G (Addgene #12259) were co-transfected into 293 T cells using lipofectamine 2000 transfection reagent. The cell culture medium was changed on the next morning. Supernatant was collected 48 h and 72 h post-transfection and filtered through 0.45 μm pore size cellulose acetate filter (Thermo Fisher) and incubated with PEG4000 solution (8% final concentration) overnight at 4 °C. The following day, mixtures were centrifuged at 1500xg for 30 min at 4 °C. After discarding the supernatant, the pellet containing the virus was resuspended in 100 μl sterile Phosphate Buffered Saline (PBS) per every 8 ml supernatant. Lentiviruses were used freshly or aliquoted and frozen at −80 °C. To transduce lentivirus in cells, an appropriate amount of lentivirus was added to cell culture medium in the presence of polybrene (final concentration 8 μg/mL).

**Small interfering RNA(siRNA) transfection in Hela cells and NRVCMs.** Hela cells were plated at $1.2 \times 10^5/2$ ml/well in 6 well plate. The next day, Hela cells were transfected with indicated siRNA (final concentration 10 nM) using Lipofectamine® RNAiMAX(Thermo Fisher) following the manufacturer's instructions for 72 h. NRVCMs were plated at $8 \times 10^6/10$ ml/100 mm dish. At 1–2 days after culturing in serum-free Medium199, NRVCMs were transfected with indicated siRNA (final concentration 50 nM) using HiPerfect transfection reagent (Qiagen) according to the manufacturer's instructions. At 48 h post the first transfection, NRVCMs were transfected again for another 48 h. NRVCMs were collected from 100 mm dish by trypsinization and subcultured in 6 well plate or 4 well chamber slide for various treatments. All the stealth siRNAs were obtained from Thermo Fisher.

Stealth siRNA transfected in NRVCMs,
GRAF1 stealth siRNA:
5′-CGGAAGUUUGCAGAUUCCUUAAAUG-3′,
5′-CAUUUAAGGAAUCUGCAAACUUCCG-3′;
GFP stealth siRNA:
5′- GGUGCGCUCCUGGACGUAGCC[dT][dT]-3′
5′-GGCUACGUCCAGGAGCGCACC[dT][dT]-3′
Stealth siRNA transfected in Hela cells, siGRAF1-A:
5′- UCUUCACUUUCUAUCACCAUGGUUA-3′,
5′- UAACCAUGGUGAUAGAAAGUGAAGA-3′;
siGRAF1-B:
5′- CAUUUCUAUGAAGUAUCCCUGGAAU-3′,
5′- AUUCCAGGGAUACUUCAUAGAAAUG-3′.
Rat PINK1 stealth siRNA (Catalog# RSS311777), Human PINK1 stealth siRNA (Catalog# HSS127945), Human Atg12 stealth siRNA

(Catalog# HSS113534) and stealth negative control siRNA (Catalog# 12935110) were directly purchased from Thermo Fisher.

**Assessment of ΔΨm.** JC-1(Thermo Fisher) forms J-aggregates (red) in healthy cells with high ΔΨm and exists as monomers (green) in cytosol or in cells containing unhealthy mitochondria. Loss of ΔΨ is visualized by a shift from red to green fluorescence. Carbonyl cyanide 3-chlorophenylhydrazone(CCCP)(Sigma) is a potent mitochondrion oxidative phosphorylation uncoupler, so CCCP treated cells were used as control to monitor a fluorescence shift.

To assess ΔΨm by flow cytometry, at 96 h post indicated siRNA transfection, NRVCMs were trypsinized from 100 mm dishes and passed through 70 μm mesh size cell strainer (Falcon) to remove cell aggregates. Next, NRVCMs were aliquoted at $\sim 3 \times 10^5$/ml in Eppendorf tubes and incubated with JC-1(2 μM) in the presence or absence of CCCP (50 μM) for 30 min at 37 °C and 5% CO2. NRVCMS then were quickly washed 2 times with PBS and analyzed by flow cytometer in FL1 channel (green, fluorescent monomeric signal) and FL2 channel (red fluorescent aggregated signal). A total of $1 \times 10^5$ NRVCMs of each experimental condition were recorded and analyzed. Signal from CCCP treated NRVCMs was used as background control for data analysis. The intensity ratio of FL2(red) to FL1(green) was used to evaluate mitochondria membrane potential in cells. All sample acquisition was performed with Attune NxT (version 3.1; Thermo Fisher) and further analyzed by FlowJo software version 10.6.1.

**Mito stress test by seahorse XF assay.** Cell Mito Stress Test was performed on a Seahorse XFe24 analyzer (Agilent) following manufacturer's instruction. NRVCMs were plated in a sterile XFe24 cell culture plate at $5 \times 10^4$/well (5 wells for individual experiment condition) and were serially stimulated with the following compounds at the final concentration noted: 5 μM oligomycin A; 5 μM CCCP; 2 μM rotenone/antimycin A. Respiratory parameters including basal oxygen consumption rate (OCR), OCR related to ATP production, maximum respiratory capacity, spare respiratory capacity were calculated based on oxygen levels detected in cell culture medium.

**Inducible CRISPR/CAS9 mediated knockout.** sgRNA(5′-gtttcgaaaggaacagatcg-3′) targeting human GRAF1 or non-target sgRNA was cloned into LentiCRISPRv2-BSD-DD-Cas9 vector expressing inducible SpCas9. To establish DD-Cas9 stable expression cells, Hela/YFP-Parkin cells were transduced with lentivirus expressing DD-Cas9/sgRNA-GRAF1 or DD-Cas9/non-target sgRNA followed by Blasticidin(6 μg/ml) selection. Next, Hela/YFP-Parkin cells were treated with 1 μM shield-1(MedChemExpress) for 5 days to induce GRAF1 knockout prior to OA treatment.

**Real time PCR.** Total RNA was extracted from cultured NRVCMs using the RNeasy mini kit (Qiagen). Total 500 ng RNA was used for reverse transcription using iScript cDNA synthesis kit (Bio-Rad) in 20 μl reaction mixtures following the manufacturer's protocols. Gene expression was analyzed by semi-quantitative PCR (qPCR) using iTaq Universal SYBR Green Supermix 2X(Bio-Rad). The relative gene expression levels were calculated using delta-delta Ct method, also known as the $2^{-\Delta\Delta Ct}$ method. Primer pairs used for each target were listed in SOURCE FILE.

**Cytosolic and mitochondria-enriched subcellular fractionation.** Hela/YFP-Parkin cells cultured in 100 mm dishes were washed with PBS and gently scrapped and collected in 1 ml cold mitochondria isolation buffer (210 mM mannitol, 70 mM sucrose, 5 mM Tris-HCl pH7.4, 1 mM EDTA) supplemented 1x HALT phosphatase and protease inhibitor cocktail (Thermo Fisher). Cells were centrifuged at 450xg for 10 min at 4 °C to collect cell pellets that were subsequently resuspended in 500 μl mitochondria isolation buffer. Next, cells were homogenized

using a Teflon pestle operated at 1600 rpm. Cell debris was removed by centrifugation at 1400xg for 10 min. Supernatants were transferred to a clean Eppendorf tube followed by centrifuge at 12000xg at 4 °C for 15 min. Supernatant (cellular cytosolic fraction) was carefully transferred to a fresh Eppendorf tube and the pellet (mitochondria enriched fraction) was gently resuspended and washed in 1 ml mitochondria isolation buffer. Both cytosolic and the mitochondrial fractions were centrifuged again at 12000xg at 4 °C for 15 min. Cytosolic supernatant was then transferred to a clean Eppendorf. The pellets of mitochondria fractions were lysed in 100 μl RIPA buffer (150 mM NaCl, 25 mM Tris-HCl pH7.4, 1% NP40, 0.5% sodium deoxycholate and 0.1% SDS).

**Immunocytochemistry and light microscopy.** Precision NO 1.5H coverslips (Thorlabs) were coated with 5 μg/ml Fibronectin(R&D) overnight at 4 °C, appropriate number of HeLa cells or NRVCMs were plated on the coverslips as indicated in the figure legends. After treatment, cells were rinsed in PBS and fixed with PBS-buffered 4% paraformaldehyde for 15 min at room temperature. To stain mitochondria matrix proteins HSP60 and ATP5B, cells were permeabilized with PBS containing 0.1% Triton X-100 at room temperature for 10 min. To stain all the other targets, cells were permeabilized with PBS containing 0.1% saponin for 15 min at room temperature. Subsequently, cells were blocked in PBS containing 10% goat serum for 1 hr at room temperature. Next, slides were incubated with primary antibody for 1 hr at room temperature followed by 1 hr incubation of fluorophore conjugated secondary antibody and mounted with ProLong Diamond Antifade Mountant (Thermo Fisher) overnight at room temperature. Primary antibodies were mouse monoclonal anti-ATP5B (MAB3494, Millipore, 1:500), rabbit anti-HSP60 (12165 S, Cell Signaling Technology, 1:500), mouse anti-TOMM20 (sc-17764, Santa Cruz, 1:250), mouse anti-LC3B(sc-398822, Santa Cruz, 1:100), mouse anti-HSP60(sc-13115, Santa Cruz, 1:250), chicken anti-GFP(A10262, Thermo Fisher, 1:500), rabbit anti-GRAF1(homemade antibody, 1:250). Secondary antibodies were highly cross-adsorbed goat anti-mouse Alexa Fluor 488(A-11001, Thermo Fisher, 1:500), goat anti-rabbit Alexa Fluor 488(A11008, Thermo Fisher, 1:500), donkey anti-rabbit Alexa Fluor 555(A31570, Thermo Fisher, 1:500), and donkey anti-mouse Alexa Fluor 647(A31571, Thermo Fisher, 1:500), goat anti-chicken Alexa Fluor 488(A11039, Thermo Fisher, 1:500). F-actin was stained by Phalloidin Alexa Fluor 555(A34055, Thermo Fisher, 1:100). For confocal microscopy, images were acquired using 20x/0.80 Plan Apo or 63 × /1.4 Plan Apo Oil objective on an LSM 700 confocal laser-scanning microscope (Zeiss, Germany). Images used to compare each other were acquired with the same setting. Single plane or Z stack images were acquired. Z-series were deconvoluted and displayed as maximum z-projections, and gamma, brightness, and contrast were adjusted (identically for compared image sets) using Fiji (version 1.53q). Mander's colocalization coefficient was obtained by using the Fiji Coloc 2 plugin. To differentiate GFP-LC3 puncta from the GFP background signal, 4–5 regions near the GFP-LC3 puncta were selected in the representative slice. The mean background value was calculated for these regions, and an identical threshold based on mean background was applied to all images prior to analysis. Individual cells were manually chosen as "regions of interest" (ROIs) within the deconvolved image through hand-drawing. Colocalization analysis was performed using the Fiji Coloc 2 plugin, which calculated the Mander's colocalization coefficient. The entire Z-stack was analyzed to assess the extent of colocalization between GFP-LC3 puncta and Tomm20.

For super-resolution structured illumination microscopy (SR-SIM), 3D SIM images were acquired by N-SIM Ti inverted microscope (Nikon) equipped with a 100x oil immersion objective (1.49 NA, Nikon CFI Apochromat TIRF 100x) and EMCCD camera (Andor DU-897).

Three angles of the excitation grid with five phases each were acquired for each channel. SIM processing was performed using the SIM module of the NIS view software package. Subsequently, 16-bit grayscale tiff files were exported to Fiji for processing into rendered colored images.

**Bacterial protein expression, purification, and pulldown assays.** GRAF1b isoform full length or various GRAF1 domains were cloned into pGEX-6P-1 vector and expressed as glutathione-s-transferase (GST) fusion proteins in BL21 E. coli. The expressed GST-fusion proteins were purified with Glutathione Sepharose 4B (Cytiva). For GST pull-down assays, Hela/YFP-Parkin cells were collected in lysis buffer (20 mM Tris HCl, 150 mM NaCl, 10% glycerol, 0.5 % Triton X100 pH7.4 with 1x HALT phosphatase & protease inhibitor cocktail). Cell lysates were incubated on ice for 15 min and centrifuged at 20,000xg for 15 min. The supernatants were incubated with GST-fusion protein-loaded beads for 1 hr end-over-end rotation at 4 °C. Beads were washed 5 times with cell lysis buffer and stored 4 °C until use or up to 7 days.

**In Vitro phosphorylation of GRAF1.** GST-GRAF1 WT and non-phosphorylatable mutants (S668A, T670A, S671A, and AAA) were expressed in BL21 E. coli and purified as described above. Wild-type or kinase-dead PINK1 (MRC-PPU reagents) was incubated with purified GST-GRAF1WT and mutants, separately (-1 μg each) in kinase reaction buffer (10 mM MOPS, pH 7.2, 5 mM glycerol 2-phosphate, 10 mM $MgCl_2$, 2 mM EGTA, 0.8 mM EDTA, 1 μM DTT, and 1 μM ATP). After incubation for 15 min at 30 °C, the reaction mixtures were boiled in Laemmli buffer + β-mercaptoethanol for 5 min and resolved by 7% SDS-PAGE. GRAF1 phosphorylation was examined by western blot using homemade rabbit anti-GRAF1 phos-S668T670S671 antibody. Total protein was measured by Ponceau S staining.

**Co-Immunoprecipitations and western blot analyses.** To perform co-immunoprecipitation of target proteins from COS7 cell lysates, COS7 cells were transfected with the specified plasmids and expressed the target proteins for a duration of 20 h. Subsequently, transfected COS7 cells were treated with a combination of Oligomycin A (2.5 μM) and Antimycin A (250 nM) for a period of 6 h to induce mitophagy. Cell lysates were prepared in lysis buffer (50 mM HEPES-HCl pH7.4, 150 mM NaCl, 1 mM EDTA, 10% glycerol, 0.8% Chaps and 1x HALT phosphatase & protease inhibitor cocktail). For co-immunoprecipitation of target proteins from Hela cells, cell lysates were prepared in lysis buffer (20 mM Tris HCl, 150 mM NaCl, 10% glycerol, 0.5 % Triton X100 pH7.4 with 1x HALT phosphatase & protease inhibitor cocktail). Cell lysates were incubated on ice for 15 min and centrifuged at 20,000 rcf for 15 min. Protein concentration was measured by Pierce BCA protein assay kit (Thermo Fisher). Protein lysates (500 - 800 μg) were incubated with mouse anti-Flag M2 (F1804,Sigma) or rabbit anti-GFP (A11122, Thermo Fisher) bound with Dynabeads® Protein G (Thermo Fisher) with end-over-end rotation for 3 h at 4 °C. Magnetic beads were washed 4 times with lysis buffer and were transferred to a clean tube for one more wash. Co-IPd protein complexes were eluted by boiling in Laemmli buffer plus β-mercaptoethanol for 5 min.

For western blot analysis, Hela cells or NRVCMs in 6 well plates with indicated treatment were lysed in Triton X-100 lysis buffer (25 mM Tris-HCL pH7.4, 150 mM NaCl, 10% glycerol, 1% Triton X-100 with 1x HALT phosphatase & protease inhibitor cocktail). Lysate concentration was measured by Pierce BCA protein assay kit. Approximately 25 - 50 μg protein per sample was separated on SDS-PAGE and then wet transferred in Tris-glycine transfer buffer to 0.2 μm pore size nitrocellulose membranes (Bio-Rad). HRP-conjugated secondary antibodies and ECL chemiluminescent substrate (GE Healthcare) were used to detect target proteins in blots.

The following primary antibodies were used in western blot: mouse anti-SQSTM1/p62 (ab56416, Abcam, 1:3000), rabbit anti-LC3A/B (12741 S, Cell Signaling Technology, 1:1000), mouse anti-Flag M5(F4042, Sigma, 1:4000), rabbit anti-Myc(2278 S, Cell Signaling Technology, 1:1000), mouse anti-Parkin(4211 S, Cell Signaling Technology, 1:1000), rabbit anti-PINK1(6946 S, Cell Signaling Technology, 1:500), rabbit anti-GFP(A11122, Thermo Fisher, 1:1000), mouse anti-β-Actin(3700 S, Cell Signaling Technology, 1:1000), rabbit anti-GAPDH(5174 S, Cell Signaling Technology, 1:1000), mouse anti-α-Tubulin(T6074, Sigma, 1:3000), rabbit anti-HSP60(12165 S, Cell Signaling Technology, 1:1000), rabbit anti-VDAC(4661 S, Cell Signaling Technology, 1:1000), rabbit anti-ATG7(8558 S, Cell Signaling Technology,1:500), mouse anti-TIM50(sc-393678, Santa Cruz,1:1000), rabbit anti-PHB2(PA5-14133, Thermo Fisher, 1:500), rabbit anti-ATP-B(ab14730, Abcam,1:500), rabbit anti-WAVE-2(3659 S, Cell Signaling Technology, 1:1000), rabbit anti-ABI2(14890-1-AP, Proteintech, 1:500), rabbit anti-CYFIP1(ab156016, Abcam, 1:500). Rabbit anti-GRAF1 and rabbit anti-GRAF1 phospho-S668T670S671 polyclonal antibodies were homemade antibodies in our lab.

**Affinity purification mass spectrometry analysis.** Plasmids containing Flag-Empty Vector (control), Flag-GRAF1 WT (wildtype), or the S668T670S671 non-phosphorylatable variant (mutant) were transiently expressed in GRAF1 KO Hela/YFP-Parkin cells for 20–24 h followed by treating with OA (2.5 μM Oligomycin A and 250 nM Antimycin A) for 6 h. Flag-GRAF1 was IPd by M2 Flag antibody as the method described above. Next, protein samples were subjected to on-bead trypsin digestion, as previously described[67]. After the last wash buffer step, 50 μl of 50 mM ammonium bicarbonate (pH 8) containing 1 μg trypsin (Promega) was added to beads overnight at 37 °C with shaking. The next day, 500 ng of trypsin was added then incubated for an additional 3 h at 37 °C with shaking. Supernatants from pelleted beads were transferred, then beads were washed twice with 50 ul LC/MS grade water. These rinses were combined with original supernatants, then acidified with 2% formic acid. Peptides were desalted with peptide desalting spin columns (Thermo) and dried via vacuum centrifugation. Peptide samples were stored at -80 °C until further analysis.

**LC-MS/MS analysis.** The peptide samples were analyzed in duplicate by LC-MS/MS using an Easy nLC 1200 system coupled to a QExactive HF mass spectrometer (Thermo Scientific). Samples were injected onto an Easy Spray PepMap C18 column (75 μm id × 25 cm, 2 μm particle size) (Thermo Scientific) and separated over a 120 min period. The gradient for separation consisted of a stepwise gradient from 5 to 48% mobile phase B, where mobile phase A was 0.1% formic acid in water and mobile phase B consisted of 0.1% formic acid in 80% acetonitrile. The flow rate was set to 250 nl/min. The QExactive HF was operated in data-dependent mode where the 15 most intense precursors were selected for subsequent fragmentation. Resolution for the precursor scan (m/z 350–1700) was set to 60,000 with a target value of $3 \times 10^6$ ions. MS/MS scan resolution was set to 15,000 with a target value of $1 \times 10^5$ ions, 75-ms maximum injection time. The normalized collision energy was set to 27% for higher-energy collisional dissociation (HCD). Dynamic exclusion was set to 30 s, peptide match was set to preferred, and precursors with unknown charge or a charge state of 1 and $\geq 7$ were excluded.

**Data analysis.** Raw data (see ProteomeXchange database under accession code PXD043212) were processed using the MaxQuant software suite (version 1.6.12.0) for identification and label-free quantitation (LFQ)[68,69]. Data were searched against a Uniprot Human reviewed database (containing 20,396 sequences, downloaded March 2021) using the integrated Andromeda search engine. A maximum of two missed tryptic cleavages were allowed. The variable

modification specified was phosphorylation of serine/threonine/tyrosine, oxidation of methionine, protein N-terminal acetylation. LFQ was enabled. Results were filtered to 1% FDR at the unique peptide level and grouped into proteins within MaxQuant[70]. Match between runs was enabled.

Protein and phosphopeptide datasets were both imported into Perseus (version 1.6.14.0) for[71] further processing. Only proteins with >1 unique+razor peptide were used for LFQ analysis. Proteins with >50% missing values were removed, and missing values were replaced from normal distribution. Log2 fold change ratios (wildtype vs control and mutant vs control) were calculated using the averaged log2 LFQ intensities of the technical replicates for each sample group. Student's t-test was performed for the pairwise comparisons, and p-values were calculated. Proteins were considered GRAF1 interactors if the log2 fold change ratio compared to control was >1.

**Protein de-phosphorylation and Phos-tag gels.** To dephosphorylate protein in cell lysates, cells were collected and lysed in Triton X-100 lysis buffer as above supplemented with 10 mM MgCl$_2$. Calf intestinal phosphatase (CIP or Lambda PP; New England BioLabs) was added to half of the cell lysate and the other half was used as an untreated control. Both samples were incubated for 30 min at 37 °C and analyzed by SDS–PAGE and immunoblotting. To analyze GRAF1 phosphorylation by Phos-tag gels, lysates were prepared in Triton X-100 lysis buffer with 1x HALT phosphatase & protease inhibitor cocktail and run on 8% SDS-PAGE gels containing 25 μM Phos-tag (Wako Chemicals) and 50 μM MnCl$_2$. Regular SDS-PAGE gels were run simultaneously as control. Just after the electrophoresis, the Phos-tag gels were soaked in a general transfer buffer containing 10 mM EDTA for 15 min/time with gentle agitation for two times to remove excess Mn$^{2+}$ in the gel before wet transfer. Next, the gels were soaked in a general transfer buffer without EDTA for another 10 min with gentle agitation. All other steps in this analysis were identical to general SDS-PAGE and immunoblotting protocols. Fiji was used for quantification of bands in both normal SDS-PAGE and Phos-tag immunoblotting.

**Anti-Phospho-GRAF1 antibody generation.** Based on GRAF1b isoform amino acids, the following peptides were synthesized at >90 purity by Thermo Fisher.

Phospho-S668T670S671-KLH:LPPNP[S]P[T][S]PLSPS[C]-KLH
Phospho-S668T670S671:LPPNP[S]P[T][S]PLSPS
S668T670S671: LPPNPSPTSPLSPS

[S],[T],[S] indicate phosphorylation modification, [C]-KLH indicates addition a Cysteine at C terminus of synthetic peptide and conjugation of carrier protein keyhole limpet hemocyanin (KLH). Phospho-peptides conjugated to KLH were submitted to Cocalico Biologicals, Inc. to produce polyclonal antibody in a standard 91-day protocol in rabbits. Anti phospho-peptide antibody purification employed two peptide columns, a non-phospho-peptide column to subtract antibodies that recognize total protein and a phospho-peptide column for affinity purification of phospho-specific antibody[72,73].

**Immobilization of peptide to resin.** Generally, follow instruction of AminoLink Plus Coupling Resin (Thermo Fisher) to generate peptide conjugated resin. 2 mg peptide (the non-phospho-peptide for the subtraction or the phosphor-peptide for affinity purification) was dissolved in 3 ml coupling buffer (0.1 M sodium citrate, 0.05 M sodium carbonate, pH 10). 4 ml AminoLink Plus Coupling Resin slurry (2 ml resin) was equilibrated by adding 10 ml (5 resin-bed volume) pH10 coupling buffer to capped tube. 3 ml dissolved peptide solution was added to the column. The top cap was placed on the column and the reaction slurry was mixed by gentle end-over-end rotation for 4 hr. Next, both top and bottom caps were removed to let solution in the column to drain. The resin was washed with 10 ml pH7.2 coupling

buffer (0.1 M sodium phosphate, 0.15 M NaCl, pH 7.2). In a fume hood, 2 ml pH7.2 coupling buffer and 40 μL sodium cyanoborohydride solution (5 M NaCNBH3 in 1 M NaOH) was added. Resin in column was mixed by end-over-end rotation overnight at 4 °C. The next day, resin was washed with 5 ml quenching buffer (1 M Tris HCl, pH7.4). In a fume hood, 2 ml pH7.2 coupling buffer and 40 sodium μL cyanoborohydride solution was added to the column.The columnn was gently mixed for 30 min by end-over-end rotation. Next, the column was washed with a 10 ml wash solution (1 M NaCl) followed by washing with 10 ml PBS containing 0.05% sodium azide. The column was stored at 4 °C.

**Affinity Purification of Phospho-Antibody.** 4 ml crude antiserum was spun at 2000g for 5 min at 4 °C to remove large debris, then the antiserum was diluted 1:1 with binding buffer (PBS, pH7.4). The non-phospho-peptide column was equilibrated by adding 10 ml PBS and subsequently adding the diluted antiserum to the column. The column was capped and parafilm wrapped and rotated end-over-end overnight at 4 °C. The next day, antiserum flow-through was collected from the column, column was washed with 5 ml PBS and flow-through was collected. The flow-through was the mixture of antibodies that did not bind to non-phospho-peptide column and contained antibody specific to the phospho-peptide.

Next, the phospho-peptide column was equilibrated with 10 ml PBS, then flow-through fractions collected from the non-phospho-peptide column were added to the phospho-peptide column. The column was caped and parafilm wrapped, gently inverted and the column was rotated overnight at 4 °C. The next day, incubation was continued at room temperature for 1 h then flow-through was collected. Next, the column was washed with 10 ml (5 resin-bed volume) PBS followed by 20 ml of stringent washing buffer (1 M NaCl, 1% TritonX-100 in PBS, pH7.4). The column was washed again with 10 ml PBS prior to elution of anti-phospho-peptide antibody. 150 μl of 2 M Tris HCl pH 8.0. was added to 15 × 1.5 m collection tubes. 15 ml Elution buffer (0.1 M glycine pH 2.6) was then added to the column and 1 ml fractions were collected in each tube and immediately mixed by inversion to neutralize the sample. 5 μl of each fraction was spotted on pH paper to evaluate neutralization. The antibody solution was transferred to Amicon Ultra-15 10 K centrifugal filter device and spun at 4000 g for ~20 min at 4 °C to concentrate purified antibody. Next, concentrated antibody solution was dialyzed in 300 ml PBS containing 50%(vol/vol) glycerol overnight at 4 °C. Dialysis buffer was exchanged at least one time during dialysis. Lastly, ~0.05 volume of 20 mg/ml BSA was added to the antibody solution to obtain 1 mg/ml BSA total, and purified antibody solution was stored at −20 °C.

**Transmission electron microscopy.** Mouse cardiac muscle was immersed in 2% paraformaldehyde/2.5% glutaraldehyde/0.15 M sodium phosphate, pH 7.4 and stored for several days at 4 °C. After post-fixation in 1% osmium tetroxide/0.15 M sodium phosphate the samples were washed in deionized water, then dehydrated through a graded series of ethanol (30%, 50%, 75%, 90%, 100%, 100%, 100%) and propylene oxide. The cardiac muscles were infiltrated with a 1:1 mixture of propylene oxide and Polybed 812 epoxy resin for 3 h and infiltrated overnight in 100% resin (Polysciences, Inc.,Warrington, PA). Samples were transferred to embedding molds and polymerized at 60 C overnight. Using a diamond knife, 1 μm semi-thin sections were cut, mounted on slides, and stained with 1% toluidine blue to examine by light microscopy and isolate the region of interest. Ultrathin sections (70-80 nm) were cut with a diamond knife and mounted on 200 mesh copper grids followed by staining with 4% aqueous uranyl acetate for 12 min and Reynold's lead citrate for 8 min (Reynolds, 1963). Samples were observed using a JEOL JEM-1230 transmission electron microscope operating at 80 kV (JEOL USA, Inc., Peabody, MA) and images were acquired with a Gatan Orius SC1000 CCD Digital Camera and Gatan Microscopy Suite 3.0 software (Gatan, Inc., Pleasanton, CA).

**Image analysis (FIJI).** To quantify mitochondria cross-sectional areas and aspect ratios in the TEM of hearts, the freehand selection tool and ROI manager were used to manually define each mitochondrion as a region of interest in FIJI. Set Measurements was used to select area and shape descriptors, and Measure was used to generate tables of cross-sectional areas and aspect ratios for each mitochondrion in each TEM. The total image areas as well as areas for exclusion from analysis (i.e. the intermyofibrillar and nuclear spaces) were measured by the same process for calculation of number of mitochondria per μm2. Mitochondrial cross-sectional areas and aspect ratios were plotted in MATLAB (version 9.2.0.959691; R2017a) as histograms and/or nonparametric kernel-smoothing distributions using hisfit with 20 bins. To quantify mitochondria levels by immunostaining, the ATP5B-stained mitochondria were highlighted as the region of interest (ROI) using the same threshold value in all samples in Fiji. Subsequently, mitochondrial volume was calculated by multiplying the cell height as determined by the z-slice distance measured using Fiji. To quantify percentage of cells with branched actin filaments, images of cells dual labeled with phalloidin and mitochondria marker were acquired and manually scored in a blinded fashion for the presence of branched or linear actin filaments. To quantify mitochondrial clustering, the extent of mitochondrial clustering was evaluated manually in a blinded fashion based on image examples depicting different levels of clustering, including none, diffuse, and focal patterns. To determine colocalization, the regions of interest (ROI) within the acquired images was defined using the Fiji Coloc 2 plugin, a straight line within the ROI was drawn, and the intensity values of target channel (fluorophore) along the selected line was measured and graphical representation of the intensity profile and colocalization result was created.

**Metabolite extraction, profiling and metabolomic data analysis.** Metabolite extraction was performed as described in a previous study[74]. Following flash freezing ~ 30 mg tissue sections were homogenized into powder in liquid nitrogen by mortar & pestle. 3 - 10 mg tissue powder was thoroughly mixed with 200 μl ice cold 80% methanol (HPLC grade). An additional 300 μL ice cold 80% methanol was added in each tube and vortexed rigorously for 1 min. After incubation on ice for 10 min, the tissue extract was centrifuged at 20,000 g at 4 °C for 15 min. The supernatant, normalized to sample weight, was transferred and aliquoted into two clean Eppendorf tubes. Next, the tubes were speed vacuum dried at room temperature. The dry pellets were stored at −80 °C for liquid chromatography with high-resolution mass spectrometry (LC-MS) analysis.

Ultimate 3000 UHPLC (Dionex) coupled to Q Exactive Plus-Mass spectrometer (QE-MS, Thermo Scientific) was used for metabolite profiling. The dry pellets were reconstituted into 30 μL sample solvent (water:methanol:acetonitrile, 2:1:1, v/v) and 3 μl was further analyzed by LC-MS. A hydrophilic interaction chromatography method (HILIC) employing an Xbridge amide column (100 × 2.1 mm i.d., 3.5 μm; Waters) was used for polar metabolite separation. Detailed LC method was described previously[75] except that mobile phase A was replaced with water containing 5 mM ammonium acetate (pH6.8). The QE-MS was equipped with a HESI probe with related parameters set as below: heater temperature, 120 °C; sheath gas, 30; auxiliary gas, 10; sweep gas, 3; spray voltage, 3.0 kV for the positive mode and 2.5 kV for the negative mode; capillary temperature, 320 °C; S-lens, 55; A scan range (m/z) of 70 to 900 was used in positive mode from 1.31 to 12.5 min. For negative mode, a scan range of 70 to 900 was used from 1.31 to 6.6 min and then 100 to 1000 from 6.61 to 12.5 min; resolution: 70000; automated gain control (AGC), $3 \times 10^6$ ions. Customized mass calibration was performed before data acquisition. LC-MS peak extraction and integration were performed using commercially available software Sieve 2.2 (Thermo Scientific). The integrated peak area was used to represent the relative abundance of each metabolite in different

samples. To avoid infinite values when calculate fold changes, a value 1000 was used for metabolites not detected in some samples but that have measurable signals in one or more other samples. The annotated metabolites from Kyoto Encyclopedia of Genes and Genomes (KEGG) Compound database (www.genome.jp/kegg) and Human Metabolome Database (HMDB) were used for analysis. The data matrix of metabolites extracted from mouse hearts was normalized by sum and then was auto-scaled (mean-centered and divided by the standard deviation of each variable). The data matrix of metabolites extracted from mouse serum was normalized by median and then was auto-scaled (mean-centered and divided by the standard deviation of each variable). Metabolite pathway analysis, hierarchical clustering and heatmap and statistical analysis was conducted using Metaboanalyst 5.0 (http://www.metaboanalyst.ca/MetaboAnalyst/).

**Adrenergic challenge and transthoracic echocardiography.** αMHC-MerCreMer/+ and αMHC-MerCreMer/+: GRAF1[flox/flox] male mice (2–3-month-old) received a single intraperitoneal injection (IP) injection of 40 mg/kg tamoxifen (Sigma) dissolved in sterile peanut oil. For the chronic β-adrenergic stimulation model, mice were treated at 7 days post tamoxifen injection with Isoproterenol (ISO)(Sigma) at a dosage of 15 mg/kg/day for 14 days using subcutaneously implanted Osmotic pumps (model 2002, Alzet). Cardiac physiology was evaluated by echocardiography prior to ISO infusion as well as at 7 days and 14 days post ISO infusion. Transthoracic echocardiography (ECHO) was performed using a VisualSonics Vevo 2100 system equipped with MS400 transducer (Visual Sonics). 2-D guided M-mode images of the left ventricle were obtained from long axis view at the mid-ventricular level of the papillary muscles in conscious, gently restrained mice. ECHO measurements from 5-6 consecutive cycles were averaged using Visual Sonics software (VevoLAB, Version-5.5.0). For acute β-adrenergic stimulation model, mice were treated at 7 days post tamoxifen injection with a single subcutaneous injection of ISO (200 mg/kg/day) for 3 consecutive days. Heart tissues were collected 4–6 h post the last ISO injection. To measure cardiac mitophagy, we employed confocal microscopy, following a previously described methodology[76].

**GRAF1 phosphorylation in human myocardial tissue.** Human myocardial tissue samples from hypertrophic cardiomyopathy (HCM), dilated cardiomyopathy (DCM) and non-failing (NF) control 33–57-year-old male patients (6 samples per group) were used in this study. These samples were procured from the Duke Human Heart Repository (DHHR) which is a Duke University Hospital Institutional Review Board approved tissue repository (IRB No. Pro00005621). Samples were obtained using informed consent or a waiver of consent for discarded tissues. Left ventricular (LV) cardiac tissue from HCM and DCM patients were obtained at the time of cardiac transplantation. LV cardiac tissue from NF subjects were obtained from deceased donors whose hearts were not used for transplantation. At the time of collection, the LV tissue was sharply dissected then flash frozen in a cryovial using liquid nitrogen. The tissue was stored at −80 °C until it was utilized. Patient data at the time of tissue procurement was retrospectively collected by DHHR personnel from an electronic medical record database or from organ donor records and de-identified information is presented in the Supplementary Information (Table S2). Heart tissue was homogenized in Triton X-100 lysis buffer. GRAF1 phosphorylation in the lysate was analyzed by Western blotting as described above.

**Statistical calculations.** Differences were analyzed by two-tailed unpaired Student's $t$-test for experiments with two groups and one-way or two-way ANOVA with post-hoc Tukey's or Sidak test for multiple comparisons for experiments ≥3 groups. A value of $P < 0.05$ was considered statistically significant. No statistical analysis was used to predetermine sample sizes. Data are represented as mean ± SD or mean ± SEM from at least three independent experiments or at least three biological samples. In all cases similar results were found from experiment to experiment and all technically sound experiments were used in data analysis. All statistical data were calculated and graphed using GraphPad Prism 10.0. Where not provided in figure legend due to space limitation, exact $p$-values are included in accompanying source data file.

### Reporting summary
Further information on research design is available in the Nature Portfolio Reporting Summary linked to this article.

## Data availability
All data are available in the main text, supplementary materials or relevant repositories. Proteomic data used in this study have been deposited in the ProteomeXchange database under accession code PXD043212 and raw metabolomic data used in this study have been deposited in the University of North Carolina Digital Repository under accession code cr56nb66p. We will continue our plan to share materials and manage intellectual property, in adherence to the NIH Grant Policy on Sharing of Unique Research Resources including the Sharing of Biomedical Research Resources Principles and Guidelines for Recipients of NIH Grants and Contracts. Plasmids, cell lines and other materials generated in this study are available upon request to the lead contact. Mouse lines generated by our laboratory will be freely distributed upon assurances detailed below. Some mouse lines produced in the proposed project, however, will be generated by breeding lines previously generated by other laboratories. Thus, following the characterization and peer-reviewed publication of the transgenic mouse strain generated we will either direct requests for mice to the appropriate investigators, or upon receipt of consent from the appropriate investigators we will distribute them to investigators at AAALAC (Association for Assessment and Accreditation of Laboratory Animal Care International) accredited academic institutions wanting mice for non-commercial research. The recipient investigators would provide written assurance and evidence that the animals will be used solely in accord with their local IACAC review; that animals will not be further distributed by the recipient without consent of our program; that animals will not be used for commercial purposes. Requests for mice from for-profit corporations to use the mice commercially will be negotiated by our institution's technology transfer office. All licensing shall be subject to distribution pursuant to my institution's policies and procedures on royalty income. The technology transfer office will report any disclosure submitted to them to the appropriate Federal Agency. Should any intellectual property arise which requires a patent, we would ensure that the technology remains widely available to the research community in accordance with the NIH Principles and Guidelines document. Further information and requests for resources and reagents should be directed to and will be fulfilled by the lead contact; Joan Taylor. Source data are provided with this paper (https://doi.org/10.6084/m9.figshare.24535183).

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

## Acknowledgements

We thank Krsna Rangarajan for assistance with mouse surgery, Brian A. Mack for tissue mounting, sectioning, and staining, figure creation and design and general technical support. We thank Li Qian for reviewing the manuscript and providing helpful suggestions, and Richard J. Youle for sHela cells, Hela/YFP-Parkin cells and Hela/mCherry-Parkin/GFP-LC3, Beth Levine for Hela/Parkin/GFP-LC3 cells, Toren Finkel for mt-Keima mice. pEGFP-parkin WT and C431S plasmids were gifts from Edward A. Fon. pCMVTNT PINK1 N-myc plasmid was a gift from M. Cookson. EGFP-LC3 plasmid was a gift from Karla Kirkegaard. LentiCRISPRv2-BSD-DD-Cas9 plasmid was a gift from Cyrus Vaziri. This research is based in part upon work conducted at the Microscopy Services Laboratory, Department of Pathology and Laboratory Medicine and work conducted at UNC Proteomics Core Facility, which are supported in part by a National Cancer Institute Cancer Center Core Support Grant to the University of North Carolina Lineberger Comprehensive Cancer Center (P30 CA016086). This work was also supported by NIH including NHLBI/R01HL130367 and 1RO1HL165786 to JMT, NIH T32HL007101 to M.M.P., 1F31HL145983-01 to M.E.C., and American Heart Association 16PRE30630002 to Q.Z.

## Author contributions

Experiment conceptualization, data analysis and manuscript preparation: Q.Z. Mathematical analyses of mitochondrial dimensions, CSA: M.C. metabolomic analyses: J.L. Mouse surgeries (ISO minipump implantation): X.B. Approach to assess mitochondrial clustering: W.B.W. Guidance on metabolomic assessment and manuscript preparation: J.D.L. and J.W.L. LC-MS/MS and proteomics analysis: L.E.H. D.E.B., R.T.G., and M.M.P. identified, collected and prepared indicated human heart samples for studies. Conceptual development of studies and manuscript preparation: C.P.M. Conceptual development of studies, data analysis and manuscript preparation: J.M.T.

## Competing interests

The authors declare no competing interests.
