## [Peer Review File · Nature Communications]

GRAF1 integrates PINK1-Parkin signaling and actin dynamics to mediate cardiac mitochondrial homeostasisREVIEWER COMMENTS

Reviewer #1 (Remarks to the Author):

This paper written by Zhu and colleagues describe actin remodeling and mitochondrial quality control and clearance, and the role of the RhoGAP, GRAF1 (Arhgap26) in those processes. First, the authors obtained evidence that GRAF1 plays a role in maintaining mitochondrial quality control in cardiomyocytes (NRVCM cells). They further characterize this process in HeLa cells and found that a) GRAF1 forms a complex with Parkin and PINK1, b) GRAF1 is a PINK1 substrate, c) phosphorylation of GRAF1 may contribute to mitochondrial positional control/clustering in response to OA-induced mitochondrial damage, and d) phosphorylated GRAF1 recruits the WAVE2 complex to dysfunctional mitochondria. Lastly, there is evidence presented that GRAF1 is required for stress-induced metabolic adaptation in adult hearts and is dysregulated in human heart disease

Overall, the manuscript addresses an interesting and important question: mechanisms underlying mitochondrial quality control in heart and heart disease, and the role of the actin cytoskeleton in those processes. In addition, a number of rigorous, technically sophisticated approaches were used. However, there are a number of major concerns regarding the use of HeLa as a model for cardiomyocytes and the physiological significance of the mechanistic links between GRAF1 and mitophagy.

1. HeLa cells may not be a good model to study GRAF1 function in heart. The data shown in Figures 10 and 5C indicate that treatment with OA for 6 hrs results in recruitment of GRAF1 to punctate structures that surround clusters of mitochondria in NRVCM cells. It is not clear what those structures are. However, based on staining with Hsp60, a mitochondrial marker protein, the punctate-GRAF1-containing structure are not mitochondria. On the other hand, in Fig. 2D, data is presented the GRAF1 co-localizes with Parkin, which presumably localizes to mitochondria, in cells HeLa after 4 hrs of OA treatment. This suggests that there are fundamental differences in GRAF1 behavior in heart and HeLa cells.

2. All of the studies led to the identification of direct interactions between GRAF1 and Parkin, phosphorylation of GRAF1 by Pink1, identification of sites for GRAF1 phosphorylation in response to OA treatment and GRAF1 binding partners were carried out in HeLa (or COS1) cells. These studies were thorough and for the most part well-controlled. However, they were carried out with ectopic, overexpression of Parkin-GFP, which raises concerns about the physiological relevance of the finding obtained.

3. The authors do show that GRAF1 is phosphorylated in response to OA treatment in NRVCM cells and effects on actin and of Arp2/3 complex at those punctate structures that surround but are not mitochondria in those cells. However, there is no evidence that Pink1 is the kinase under these conditions. In addition, the observed effects on mitochondrial clustering are subtle and the links between mitochondrial clustering and autophagy are not well documented in heart.

4. Although ectopically expressed Parkin is recruited to mitochondria in HeLa cells, it is not clear how

autophagy contributes to regulation of mitochondrial levels with OA treatment. Specifically, the authors provide evidence that OA treatment results in a decrease in the levels of mitochondrial marker proteins (Hsp60 and Tim50) in HeLa cells. The claim is that this decrease is due to mitophagy. However, although deletion of Atg7 results in a modest increase in Hsp60 and Tim50 levels in OA-treated cells compared to WT cells (Fig. 2A), deletion of ATG7 does not restore the level of either protein to that observed in untreated cells. This raises the possibilities that a) there are autophagy-dependent and independent mechanisms for OA-induced loss of mitochondria and b) autophagy plays a minor role in that process. Moreover, since loss of mitochondria appears to occur by autophagy-dependent and independent processes, it is not clear that the increase in mitochondrial marker proteins observed in response to reduced GRAF1 levels is due to effects on mitochondrial autophagy.

Additional technical concerns.

1. In Figures 2H and 2I, Manders' coefficients are used to quantify the level of co-localization of Tomm20 and GFP-LC3. First, in OA-treated cells, LC3 expression seems enhanced compared to the OA-treated GRAF1 knockdown cells. Therefore, the high background of LC3 signal in OA-treated cells can skew the Manders' coefficient. In figure 2I, the figure legend did not indicate whether this quantification is for vehicle-treated or OA-treated cells, which should be specified (lines 1084-1085). In material and method (M&M), the co-localization quantification is summarized in one sentence in Line 728, which is not sufficient for readers to replicate the results. Moreover, Manders' coefficients measures the co-occurrences or signal overlaps and not signal correlation, which has its own pitfall when quantifying co-localization (1,2). It would be more convincing if authors can provide co-localization coefficients using different methods (e.g. Pearson's correlation coefficient or Spearman's coefficient).
2. In Figures 2C, 3J and 4I, mitochondrial clustering is quantified by measuring ATP5B signal. However, the quantification method is not described in either the M&M or figure legend, which is confusing. In addition, it seems that this assay has only been done once with 20 cells/condition/cell line.
3. In Figure 4I, the authors examine the mitochondrial clustering phenotype in Lat-B treated cells. No micrographs were included in the main figure or in the supplemental figures. There are also no images to illustrate mitochondrial clustering phenotypes (none, diffuse and focal). In addition, there are unexplained symbols in figure 4I (▲ and †.)
4. In lines 333-335, the authors described that GRAF1 deficient cells fail to reorganize F-actin and cluster mitochondria in OA-treated cells. The mitochondrial clustering phenotype is evident, but, in figure 4H, the differences in actin and mitochondria morphology between GRAF1 deficient cells and WT cells are very subtle (there are still actin filaments on the mitochondria in GRAF1 deficient cells), which cannot support the argument in lines 333-335.
5. In Figure S3I, there is description of the method used to measure the percentage of cells with branched actin filaments.

Reviewer #2 (Remarks to the Author):

The authors found that GRAF1 was the direct substrate of PINK1 that controls mitophagy. Furthermore, they found that GRAF1 could promote mitochondria release from F-actin anchors, facilitated mitochondria capture by autophagosomes by enhancing Parkin-LC3 interactions, and mitochondria-associated Arp2/3-dependent actin remodeling. This work is of great meticulous and comprehensive. However, I have some questions as follows:

1. The authors showed GRAF1 might be a possible target in heart disease and Parkinson's disease, but did not verify in any real disease model.
2. The ISO treatment did not result in a decrease in ejection fraction, that is, it was not studied in a disease model. TAC can be adopted as a classic model of heart failure.
3. The experiments in vitro was of cardiomyocytes, and the part of the experiment did not cover Parkinson's disease at all. It is recommended not to mention Parkinson's disease in the text.
4. The mechanism was explored in great detail, however, whether a key target or small molecule compound can be identified to have a therapeutic effect on related diseases

Reviewer #4 (Remarks to the Author):

The authors study the mechanisms of clearance of dysfunctional mitochondria, in particular how the clearance processes are spatially controlled by cytoskeletal remodeling. The authors postulate that the RhoGAP, GRAF1, is a PINK1 substrate that controls mitophagy. Cardiomyocyte-restricted depletion of GRAF1 resulted in accumulation of dysfunctional mitochondria and heart failure.

The paper is very well written, elegantly combines a range of different techniques and the methods are very detailed. Altogether, this manuscript is nicely put together and I enjoyed reading it. I have the following suggestions:

- 1) Many experiments are performed in cell lines (Fig 2-4). Can you reproduce more of your key findings in the relevant cell type, i.e. adult cardiomyocytes?
- 2) Can you visualise mitochondrial remnant material in the autophagosomes to complement your assessment of the ratio of LC3I/II by Western blotting?
- 3) Figure H-J: The n-numbers in the in vivo experiment using ISO are small. Experiments should be performed in male and female mice.
- 4) Can you perform a further computational validation how likely it is that GRAF1 and Parkin interact via the proposed mechanism?

- 5) With regards to identifying mitophagy-dependent GRAF1 modifications, did you consider alternative proteases other than trypsin?
- 6) The validation in human heart tissue is important. How comparable are the patients that were used as non-failing controls vs HCM and DCM patients (age, sex, etc)? Could medication affect your read outs, i.e. beta blockers?
- 7) Also, the NF hearts may have been stored in a transplant solution? This could affect your protein phosphorylation levels.
- 8) Figure 5K: It seems GRAF1 is reduced in HF patients? Did you also assess mitochondrial protein content in failing versus non-failing hearts?
- 9) Figure 5I. The metabolite profiles look very variable between ISO-treated WT and GRAF1 conditional KO hearts? Did you control the fasting state of the mice? How do the changes relate to mitochondrial content/ morphology in WT and conditional KO hearts? How similar are their metabolite profiles at baseline without ISO treatment?
- 10) It would strengthen the manuscript if also immunostainings could be performed on histological sections from human non-failing and failing hearts.

We thank the reviewers for their positive comments and suggestions for the improvement of our manuscript. We have comprehensively addressed all concerns using a combination of new experiments, additional quantification and additional discussion points.

Specific responses follow:

Reviewer #1 Comments

This paper written by Zhu and colleagues describe actin remodeling and mitochondrial quality control and clearance, and the role of the RhoGAP, GRAF1 (Arhgap26) in those processes. First, the authors obtained evidence that GRAF1 plays a role in maintaining mitochondrial quality control in cardiomyocytes (NRVCM cells). They further characterize this process in HeLa cells and found that a) GRAF1 forms a complex with Parkin and PINK1, b) GRAF1 is a PINK1 substrate, c) phosphorylation of GRAF1 may contribute to mitochondrial positional control/clustering in response to OA-induced mitochondrial damage, and d) phosphorylated GRAF1 recruits the WAVE2 complex to dysfunctional mitochondria. Lastly, there is evidence presented that GRAF1 is required for stress induced metabolic adaptation in adult hearts and is dysregulated in human heart disease

Overall, the manuscript addresses an interesting and important question: mechanisms underlying mitochondrial quality control in heart and heart disease, and the role of the actin cytoskeleton in those processes. In addition, a number of rigorous, technically sophisticated approaches were used. However, there are a number of major concerns regarding the use of HeLa as a model for cardiomyocytes and the physiological significance of the mechanistic links between GRAF1 and mitophagy.

1. HeLa cells may not be a good model to study GRAF1 function in heart. The data shown in Figures 1O and 5C indicate that treatment with OA for 6 hrs results in recruitment of GRAF1 to punctate structures that surround clusters of mitochondria in NRVCM cells. It is not clear what those structures are. However, based on staining with Hsp60, a mitochondrial marker protein, the punctate-GRAF1-containing structures are not mitochondria. On the other hand, in Fig. 2D, data is presented the GRAF1 co-localizes with Parkin, which presumably localizes to mitochondria, in cells HeLa after 4 hrs of OA treatment. This suggests that there are fundamental differences in GRAF1 behavior in heart and HeLa cells.

Based on our original findings and new data provided in the revised manuscript, we believe that GRAF1 is recruited to the outer mitochondrial membrane (OMM) of Parkin-labeled damaged mitochondria in both cardiomyocytes and HeLa cells. It has been previously reported that damaged mitochondria in various cells (including HeLa cells) form cup-like structures that appear circular or ring-like in high-powered confocal or super-resolution images when stained with an outer mitochondrial membrane marker and/or Parkin (Narendra D., et al., J Cell Biol. 2008 Dec 1;183(5):795-803. Chen Y, Dorn GW 2nd, Science. 2013 Apr 26;340(6131):471-5. Kruppa AJ., et al., Dev Cell. 2018 Feb 26;44(4):484-499). We agree that it was somewhat difficult to appreciate these structures in the low power images previously provided to support co-localization of GRAF1 and Parkin in OA-treated HeLa cells (Fig 2D). We have performed additional experiments and now

provide new data using higher power confocal imaging to confirm that endogenous GRAF1, Parkin, and the outer mitochondrial marker, TOMM20 co-label similar circular structures in both OA-treated HeLa cells and OA-treated cardiomyocytes (see new Figs S1N and O, Fig S2C). These findings are in keeping with our prior super-resolution images showing that endogenous GRAF1 is recruited to circular structures that surround the mitochondrial matrix as marked by HSP60 (Fig 1O and Fig 5E). Thus, we believe that HeLa cells are a valid model for studying the underlying mechanisms by which GRAF1 promotes mitophagy. Indeed as described in further detail below, our new findings provide further support that GRAF1 promotes PINK1/Parkin-dependent mitophagy in both HeLa cells and cardiomyocytes in vitro and in vivo.

2. All of the studies led to the identification of direct interactions between GRAF1 and Parkin, phosphorylation of GRAF1 by Pink1, identification of sites for GRAF1 phosphorylation in response to OA treatment and GRAF1 binding partners were carried out in HeLa (or COS1) cells. These studies were thorough and for the most part well-controlled. However, they were carried out with ectopic, overexpression of Parkin-GFP, which raises concerns about the physiological relevance of the finding obtained.

To confirm and extend the physiological relevance of our findings, we conducted additional experiments in primary neonatal cardiomyocytes (under endogenous expression conditions). Our results show that OA treatment of NRVCMs promotes GRAF1 phosphorylation on AA 668/670/671 (the exact sites identified in HeLa and Cos7 cells; Figures 5A and B). Likewise, these sites are also phosphorylated in mouse hearts following the induction of mitophagy with isoproterenol treatment and are phosphorylated in human heart samples from patients with HCM and DCM (Figures 5I,J,O and Fig S3P). We also performed additional experiments wherein we depleted endogenous PINK1 in NRVCMs using validated siRNAs and now show that PINK1 depletion significantly and robustly mitigated OA-dependent phosphorylation of GRAF1 on AA 668/670/671 (Fig 5C and D). Moreover, as mentioned above, we provide strong new evidence that following OA treatment, endogenous GRAF1 and endogenous Parkin co-localize on mitochondria in NRVCMs (Figure S1N). When taken together with additional findings that siRNA-mediated GRAF1 knockdown (Figure S1K and L) and CRISPR-mediated GRAF1 knockout (Figure S2B) attenuated OA-stimulated mitochondrial clearance in NRVCMs and that mitophagy was significantly reduced in ISO treated GRAF1^{CKO} hearts (Fig 1K,L,M and Fig S1J, new Fig 5K and L), these studies provide strong support for our conclusion that GRAF1 promotes PINK1/Parkin-dependent mitophagy in both HeLa cells and cardiomyocytes in vitro and in vivo.

3. The authors do show that GRAF1 is phosphorylated in response to OA treatment in NRVCM cells and effects on actin and of Arp2/3 complex at those punctate structures that surround but are not mitochondria in those cells. However, there is no evidence that Pink1 is the kinase under these conditions. In addition, the observed effects on mitochondrial clustering are subtle and the links between mitochondrial clustering and autophagy are not well documented in heart.

As noted above, we have performed additional experiments to confirm that GRAF1 phosphorylation in NRVCMs is PINK1-dependent (Fig 5C and D). We have also shown that

following OA treatment, GRAF1 localizes to the OMM in both HeLa-Parkin and NRVCs (Fig S2C and Fig S10), that GRAF1 is required for OA-mediated actin remodeling in both cell types (Fig 4G and H, Fig 5F and G), that GRAF1 is required for mitophagosome formation in both cell types and that GRAF1 is required for mitochondrial clearance in both cell types. Nonetheless, as noted by the reviewer, there are some notable differences between these cell types with respect to the spatial-temporal dynamics of actin remodeling and mitochondrial clearance. Notably, in HeLa cells, OA-treatment leads to wide-spread F-actin dissolution and rapid translocation of damaged mitochondria to peri-nuclear clusters where they are subsequently encapsulated by autophagosomes and cleared by lysosomes. In cardiomyocytes, the actin cytoskeleton is remodeled following OA-treatment, but the cytoskeletal structure is better maintained which limits mitochondrial mobility. Thus, damaged mitochondria form clusters and associate with autophagosomes and lysosomes, but do not undergo robust retrograde trafficking. Importantly, as noted above, we have shown that GRAF1 is necessary for the association of damaged mitochondria with both autophagosomes and lysosomes in both HeLa cells and in NRVCs, indicating a direct role for GRAF1 in mediating mitophagy in both cell types.

To further confirm that GRAF1 plays a crucial role in adult cardiomyocyte mitophagy, we performed additional experiments using the powerful mt-Keima mouse model. Mt-Keima is a pH sensitive fluorescent protein that is targeted to the mitochondrial matrix by the N-terminal targeting signal of COX8A and has been validated as a rigorous and reproducible reporter to track PINK1/Parkin-dependent mitophagy in vivo (Sun N., et al., Mol Cell. 2015 Nov 19;60(4):685-96. Liu YT., et al., Autophagy. 2021 Nov;17(11):3753-3762). Using this model system, we now show that hearts from ISO-treated mt-Keima/GRAF1^{CKO} mice have significantly lower levels of mitophagy than hearts from similarly treated mt-Keima/GRAF1^{+/+} mice (Figure 5K and L). These new results provide strong support for our hypothesis that GRAF1 facilitates cardiomyocyte mitophagy and, when coupled with the significant functional and metabolic changes in cardiac-restricted GRAF1 knockout (GRAF1^{CKO}) mice following ISO treatment, further highlight the physiological relevance of our study.

4. Although ectopically expressed Parkin is recruited to mitochondria in HeLa cells, it is not clear how autophagy contributes to regulation of mitochondrial levels with OA treatment. Specifically, the authors provide evidence that OA treatment results in a decrease in the levels of mitochondrial marker proteins (Hsp60 and Tim50) in HeLa cells. The claim is that this decrease is due to mitophagy. However, although deletion of Atg7 results in a modest increase in Hsp60 and Tim50 levels in OA-treated cells compared to WT cells (Fig. 2A), deletion of ATG7 does not restore the level of either protein to that observed in untreated cells. This raises the possibilities that a) there are autophagy-dependent and independent mechanisms for OA-induced loss of mitochondria and b) autophagy plays a minor role in that process. Moreover, since loss of mitochondria appears to occur by autophagy-dependent and independent processes, it is not clear that the increase in mitochondrial marker proteins observed in response to reduced GRAF1 levels is due to effects on mitochondrial autophagy.

Thank you for your probing question to help us better clarify the conclusions that can be drawn from our studies using the Parkin-HeLa cell model. This model has been validated by the Youle

laboratory and others as a reliable tool to assess PINK1/Parkin-dependent mitophagy, and using GRAF1-specific siRNAs we provide strong evidence that GRAF1 is necessary for OA-mediated mitochondrial clearance (as assessed by a significant increase in mitochondrial IMM and matrix proteins in OA-treated GRAF1-deficient versus sufficient cells). However, as pointed out by this reviewer, depletion of the canonical macro-autophagy regulator, Atg7 led to a more modest attenuation of OA-mediated mitophagy. This is likely due to the fact that while Atg7 is critical for the formation of autophagosomes during general autophagy (i.e. in response to nutrient depletion), there are other membrane sources that can be utilized to drive autophagosome formation during mitophagy. Indeed, Hirota et al showed that mitophagy-associated autophagosomes can be produced in Hela cells an ATG7-independent but Rab9-dependent manner in which vesicles derived from the trans-Golgi and late endosomes mediate elongation of isolation membranes. Our data are consistent with their data and others showing that ATG7 contributes to, but is not essential for mitophagy (Harper JW., et al., Nat Rev Mol Cell Biol. 2018 Jan 23;19(2):93-108; Nishida Y., et al., Nature. 2009 Oct 1;461(7264):654-8; Hirota Y., et al., Autophagy 2015;11(2):332-43). We have modified our results to better clarify this point.

Additional technical concerns.

1. In Figures 2H and 2I, Manders' coefficients are used to quantify the level of co-localization of Tomm20 and GFP-LC3. First, in OA-treated cells, LC3 expression seems enhanced compared to the OA-treated GRAF1 knockdown cells. Therefore, the high background of LC3 signal in OA-treated cells can skew the Manders' coefficient. In figure 2I, the figure legend did not indicate whether this quantification is for vehicle-treated or OA-treated cells, which should be specified (lines 1084-1085). In material and method (M&M), the co-localization quantification is summarized in one sentence in Line 728, which is not sufficient for readers to replicate the results. Moreover, Manders' coefficients measures the co-occurrences or signal overlaps and not signal correlation, which has its own pitfall when quantifying co-localization (1,2). It would be more convincing if authors can provide co-localization coefficients using different methods (e.g. Pearson's correlation coefficient or Spearman's coefficient).

Thank you for your valuable feedback. To clarify, Figure 2I quantifies the fraction of GFP-LC3 puncta that overlaps with Tomm20 labeled mitochondria in OA-treated cells, and we have updated the figure legend accordingly. We have also expanded the Materials and Methods section to provide more detailed information as requested. Specifically, we included information on the software used for image analysis (e.g., Fiji) and provided a step-by-step description of how the Manders overlap coefficients were calculated.

We agree that the relatively high background staining in the GFP-LC3 channel in some cells could affect the Manders overlap coefficient analysis. To address this, we applied an identical threshold to all images prior to analysis. We then selected planes that displayed the full cell area (based on the GFP-LC3 signal) to identify labeled cells as individual regions of interest (ROIs). These ROIs were then used to perform the Manders overlap coefficient analysis on all z slices. We chose the Mander's method since it is commonly used in the literature for co-occurrence analysis to determine the proportion of a molecule present within a particular area,

compartment, or organelle. Thus, we believe this method is the most appropriate way to answer our question since our goal was to determine the proportion of GFP-LC3 present within mitochondria (as assessed by Tomm20 staining). We did not expect a linear correlation of intensity distribution between the two channels, as is usually observed for protein-protein interactions (typically assessed using Pearson's or Spearman's correlation coefficient).

2. In Figures 2C, 3J and 4I, mitochondrial clustering is quantified by measuring ATP5B signal. However, the quantification method is not described in either the M&M or figure legend, which is confusing. In addition, it seems that this assay has only been done once with 20 cells/condition/cell line.

Thank you for bringing to our attention that our methods section lacked appropriate detail for the quantification methods used herein to assess mitochondrial clustering and clearance. In Figure 2C, we utilized ImageJ to highlight the ATP5B-stained mitochondria as the region of interest (ROI) using the same threshold value in all samples, and then measured the size of the ROI by ImageJ. N=50 cells from three independent experiments. In Figure 3J, we followed the same method used in Figure 2C to measure the area of ATP5B-marked mitochondria, subsequently, the mitochondria volume was calculated by multiplying the cell height, which was determined from the z-slice distances of the entire cell. In figure 4I, we quantified mitochondrial clustering. This was performed manually in a blinded fashion using a previously reported method. Further details are provided in the revised methods section. In each case, data were acquired from three independent experiments and the total numbers of cells evaluated were reported. For the rescue experiment, a low dose of lentivirus was used to induce siRNA-resistant Flag-GRAF1 re-expression at low levels in order to limit any potential confounding effects that could arise due to GRAF1 over-expression. This was important to limit F-actin dissolution-induced changes in cell morphology. As such, our method led to a low transfection efficiency and fewer cells per independent experiment.

3. In Figure 4I, the authors examine the mitochondrial clustering phenotype in Lat-B treated cells. No micrographs were included in the main figure or in the supplemental figures. There are also no images to illustrate mitochondrial clustering phenotypes (none, diffuse and focal). In addition, there are unexplained symbols in figure 4I (▲ and †.)

We apologize for the oversight. We now provide examples of images used to evaluate the extent of mitochondrial clustering (none, diffuse, and focal) for Fig 4I (see new Fig S3N and Fig 4J) and all symbols are now appropriately defined in the legend.

4. In lines 333-335, the authors described that GRAF1 deficient cells fail to reorganize F-actin and cluster mitochondria in OA-treated cells. The mitochondrial clustering phenotype is evident, but, in figure 4H, the differences in actin and mitochondria morphology between GRAF1 deficient cells and WT cells are very subtle (there are still actin filaments on the mitochondria in GRAF1 deficient cells), which cannot support the argument in lines 333-335.

We see a very consistent difference in OA-dependent actin remodeling in GRAF1-containing and GRAF1-deficient cells, although we agree the particular high powered image shown in Fig

4H was not as convincing as it could be. We added additional images of cells at lower power magnification that provide better visualization of the interconnection between GRAF1, actin-remodeling, and mitochondrial clustering in OA-treated cells (Figure S3I). With this combination of low and high powered images, it is clear that OA treatment facilitates the dissolution of linear F-actin filaments and promotes formation of a mitochondrial-associated branched actin network. In GRAF1-deficient cells, long, linear F-actin filaments remain intact following OA treatment and mitochondrial-associated branched actin formation is reduced.

5. In Figure S3I, there is no description of the method used to measure the percentage of cells with branched actin filaments.

We apologize for not providing more specific detail regarding the methodology for quantifying actin filament structures. To this end, we randomly acquired images from cells dual labeled with phalloidin and mitochondria marker and manually scored in a blinded fashion at least 450 cells per condition for the presence of branched actin.

Reviewer #2 Comments

The authors found that GRAF1 was the direct substrate of PINK1 that controls mitophagy. Furthermore, they found that GRAF1 could promote mitochondria release from F-actin anchors, facilitated mitochondria capture by autophagosomes by enhancing Parkin-LC3 interactions, and mitochondria-associated Arp2/3-dependent actin remodeling. This work is of great meticulous and comprehensive. However, I have some questions as follows:

1. The authors showed GRAF1 might be a possible target in heart disease and Parkinson's disease, but did not verify in any real disease model.

We appreciate the reviewers concern, but the current studies were based upon our previous demonstration that GRAF1 plays an important cardioprotective role in *Xenopus* and in dystrophin-deficient (*mdx*) mice (Doherty, J.T., et al., J Biol Chem, 2011. **286**(29): p. 25903-21. Lenhart, K.C., et al., Skelet Muscle, 2015. **5**: p. 27) and that GRAF1 deficiency in these models was associated with a significant increase in mitochondrial defects. In brief, depletion in *Xenopus* led to cardiac dilation, pronounced pericardial edema, and lethality by tadpole stages. In mice, despite modest pathology in individual knock-outs, hearts and skeletal muscles from male and female GRAF1/dystrophin double-deficient mice exhibited severe pathologies, including significantly increased cardiomyocyte hypertrophy, reduced cardiac performance (Ejection fraction, Fractional shortening) and ultrastructural mitochondrial defects (grossly enlarged mitochondria with OMM separation and cristae disorganization characteristic of mitochondrial myopathies). Furthermore, mounting evidence suggests that mitochondrial respiratory dysfunction is among the earliest cellular defects observed in *mdx* mice and dystrophic patients and that increased autophagy is an important compensatory mechanism under these conditions (Kang, C., et al., Cardiovasc Res, 2018. **114**(1): p. 90-102.). Thus, when coupled with our robust and reproducible findings that cardiac-restricted depletion of GRAF1 inhibited PINK1/Parkin-dependent mitophagy in cardiomyocytes (see below and Fig 5K and L) and decreased cardiac function (FS and ES) and metabolic flexibility following ISO treatment (a

vigorous model of HCM and metabolic stress), we feel that our results strongly implicate GRAF1 in the progression of cardiac disease.

2. The ISO treatment did not result in a decrease in ejection fraction, that is, it was not studied in a disease model. TAC can be adopted as a classic model of heart failure.

We chose our treatment paradigm based on prior studies indicating that ISO stimulation at 15mg/kg/day for 2 weeks can induce mitophagy-associated mitochondrial biogenesis and promote fuel flexibility in hearts in order to maintain mitochondrial homeostasis and cardiac function. Using this model, we provide strong evidence that GRAF1 is necessary for maintaining both mitochondrial homeostasis and cardiac physiology (as assessed by a significant reduction in FS and EF in ISO-treated GRAF1^{CKO} mice). As noted above, we agree that it will be of future interest to explore other disease models, but these are beyond the scope of the current manuscript. However, in response to this reviewer's concern regarding the physiological relevance of our findings, we added new data that lend further strong support for our conclusion that GRAF1 plays a crucial role in promoting cardiomyocyte mitophagy in intact hearts. To this end, as noted above, we performed additional experiments using the powerful mt-Keima mouse model. Mt-Keima is a pH sensitive fluorescent protein that is targeted to the mitochondrial matrix by the N-terminal targeting signal of COX8A and has been validated as a rigorous and reproducible reporter to track PINK1/Parkin-dependent mitophagy in vivo (Sun N., et al., Mol Cell. 2015 Nov 19;60(4):685-96. Liu YT., et al., Autophagy. 2021 Nov;17(11):3753-3762). Using this model system, we now show that hearts from ISO-treated mt-Keima/GRAF1^{CKO} mice have significantly lower levels of mitophagy than hearts from similarly treated mt-Keima/GRAF1⁺⁺ mice (Figure 5K and L). These new results provide strong support for our hypothesis that GRAF1 facilitates cardiomyocyte mitophagy and, when coupled with the significant functional and metabolic changes in cardiac-restricted GRAF1 knockout (GRAF1^{CKO}) mice following ISO treatment, further highlight the physiological relevance of our study.

3. The experiments in vitro was of cardiomyocytes, and the part of the experiment did not cover Parkinson's disease at all. It is recommended not to mention Parkinson's disease in the text.

We removed the previous figure from the manuscript that used human disease-related Parkin variants and softened our speculation on a putative role for GRAF1 in Parkinson's disease pathogenesis as suggested.

4. The mechanism was explored in great detail, however, whether a key target or small molecule compound can be identified to have a therapeutic effect on related diseases

Indeed, one of the reasons we have been pursuing the GRAF family of Rho-GAPs is because of the possibility that these may be worthy druggable targets to treat a variety of diseases. Indeed our prior studies on a highly related family member, GRAF3 (Bai, X., et al., Nat Commun, 2013. 4: p. 2910; Bai, X., et al., J Clin Invest, 2017. 127(2): p. 670-680) revealed that GRAF3 is a major regulator of blood pressure in mice and confirmed the importance of GRAF3 in human hypertension. Moreover, consistent with our studies a recent meta-analysis of all BP-related GWAS studies showed that this eQTL was within the top 10 single variant association signals for systolic and diastolic BP and hypertension and we have even identified a few patients with

GRAF3 haploinsufficiency who have early onset HTN (Li, Q., et al., PLoS Genet, 2021. **17**(7): p. e1009639).

To provide the theoretical framework for our long-term goal of targeting GRAFs as a cardiovascular disease therapeutic, we have been using several state-of-the-art approaches to characterize the activation mechanism for GRAF family member allosteric regulation. We initially reported that the GAP activity of GRAF3 was regulated by a steric interaction between the N-terminal BAR/PH domain and the C-terminal GAP domain (Dee, R.A., et al., *Cells*, 2020. **9**(4)) and other investigators found that GRAF1 is controlled in a similar fashion (Eberth, A., et al., *Biochem J*, 2009. **417**(1): p. 371-7), suggesting that autoinhibition might be relieved by a small molecule. In the absence of high resolution structures for any BAR-PH-GAP protein, we used PyMol and the molecular docking search tool ClusPro to model the BAR/PH-GAP interaction (based on crystal structure of the GRAF1 GAP domain) and recovered 2 structural clusters; a "closed" conformation that likely represents inactive GRAF since conserved residues required for GTP hydrolysis and RhoA binding are concealed and an "open" active conformation in which these residues are exposed. Notably the change in the orientation of the GAP domain between the two dockings is a simple rotation about the horizontal axis by 90 degrees. Using this structural information, we developed and validated novel GRAF3 fluorescent (FRET) and bioluminescent (BRET) biosensors to monitor and quantify GRAF3 activation in sub-cellular locales or whole cell populations, respectively (Dee, R.A., et al., *Cells*, 2020. **9**(4)). Two major conclusions can be gleaned from our biophysical studies. 1) GRAF proteins undergo dynamic allosteric regulation in cells in response to physiologically relevant stimuli. And 2) GRAF conformations (and catalytic activity) can be modulated on a minute to minute timescale by post-translational modifications. The finding that a single phosphomimetic mutation attenuated the inhibitory BAR-PH-GAP interaction strongly supports our idea that GRAF family members could be activated by small molecules that target a region surrounding this residue. We are currently leveraging these data to identify activation-state stabilizing compounds with the long term goal of developing GRAF activators as potential therapeutics. Based on our new findings herein that GRAF SH3 binding is also regulated by phosphorylation-dependent relief of an allosteric inhibition between the SH3 and Serine-Proline rich domain, we also intend to explore the possibility that small molecules could target this interface to facilitate GRAF1-ABI2 interactions.

Reviewer #3 (Remarks to the Author):

The authors study the mechanisms of clearance of dysfunctional mitochondria, in particular how the clearance processes are spatially controlled by cytoskeletal remodeling. The authors postulate that the RhoGAP, GRAF1, is a PINK1 substrate that controls mitophagy. Cardiomyocyte-restricted depletion of GRAF1 resulted in accumulation of dysfunctional mitochondria and heart failure.

The paper is very well written, elegantly combines a range of different techniques and the methods are very detailed. Altogether, this manuscript is nicely put together and I enjoyed reading it. I have the following suggestions:

1) Many experiments are performed in cell lines (Fig 2-4). Can you reproduce more of your key findings in the relevant cell type, i.e. adult cardiomyocytes?

We appreciate the importance of reproducing key findings in relevant cell types and have broadened our approach accordingly. As noted above, we have performed additional experiments to confirm that GRAF1 phosphorylation in NRVCs is PINK1-dependent (Fig 5C and D). We have also shown that following OA treatment, GRAF1 is phosphorylated on identical PINK1-dependent AA in HeLa cells and primary cardiomyocytes (Fig 3G, Fig 5A and B), is translocated to the OMM in both HeLa-Parkin and primary cardiomyocytes (Fig S1N and O, Fig S2C), is required for OA-mediated actin remodeling in both cell types (Fig 4G and H, Fig S3I and J, Fig 5F and G), is required for mitophagosome formation in both cell types (Fig 2H and I, Fig 1K and M), and is required for mitochondrial clearance in both cell types.

With respect to repeating key findings in adult cardiomyocytes, we utilized GRAF1 genetrapped mice (GRAF1^{gt/gt}) and inducible cardiac-restricted GRAF1 knockout mice (GRAF1^{CKO}) to examine the impact of GRAF1 depletion on cardiac mitochondria homeostasis (Fig 1 and Fig S1) and metabolism (Fig 5M and Fig S3O). We also validated key findings including phosphorylation of GRAF1 on previously identified PINK1 phosphosites in adult hearts from ISO-treated mice and in cardiac samples from adult DCM and HCM patients. Moreover, we found that hearts from ISO-treated GRAF1^{CKO} mice exhibited reduced mitophagy as assessed by LC3 turnover and accumulation of damaged mitochondria. Importantly, as noted above, we also now demonstrate that GRAF1 facilitates mitophagy in adult cardiomyocytes in the intact heart using the powerful mt-Keima mouse model (Fig 5K and L). Mt-Keima is a pH sensitive fluorescent protein that is targeted to the mitochondrial matrix by the N-terminal targeting signal of COX8A and has been validated as a rigorous and reproducible reporter to track PINK1/Parkin-dependent mitophagy in vivo (Sun N., et al., Mol Cell. 2015 Nov 19;60(4):685-96. Liu YT., et al., Autophagy. 2021 Nov;17(11):3753-3762). Using this model system, we now show that hearts from ISO-treated mt-Keima/GRAF1^{CKO} mice have significantly lower levels of mitophagy than hearts from similarly treated mt-Keima/GRAF1⁺⁺ mice (Figure 5K and L). This finding is in keeping with additional new data provided that reveals fewer mitophagosomes in ISO-treated GRAF1^{CKO} hearts as assessed by TEM (see new Fig 1M and Fig S1J and below). Thus we have successfully reproduced many key findings in cardiomyocytes which increase the robustness and generalizability of our study.

Collectively our results provide strong support for our hypothesis that GRAF1 facilitates cardiomyocyte mitophagy and, when coupled with the significant functional and metabolic changes observed in cardiac-restricted GRAF1 knockout (GRAF1^{CKO}) mice following ISO treatment, further highlight the physiological relevance of our study.

2) Can you visualize mitochondrial remnant material in the autophagosomes to complement your assessment of the ratio of LC3I/II by Western blotting?

We sincerely appreciate your valuable comment, which has prompted us to expand our analyses and incorporate visualization techniques to further support our findings. As suggested,

we quantified the presence of autophagosomes attached to mitochondria in hearts from ISO-treated wild-type (wt) and GRAF1^{CKO} mice using TEM. Indeed, we observed significantly fewer autophagosomes per unit area in GRAF1^{CKO} hearts. Additionally, we observed increased mitochondria-associated glycogen accumulation (Fig 1M, Fig S1J), which is consistent with our observed defects in autophagy and inefficient utilization of glycogen as a substrate. We believe that these observations enhance the comprehensiveness and reliability of our study.

3) Figure H-J: The n-numbers in the in vivo experiment using ISO are small. Experiments should be performed in male and female mice.

Our sample size determination was based on power analysis using previous data, statistical considerations, and ethical guidelines, ensuring that we achieved statistical power while maintaining the highest standards of animal welfare. Notably, cardiac responses to ISO treatment via osmotic minipumps lead to very consistent outcomes. In our study, we utilized a sample size of 5 mice per group, which allowed us to observe distinct and significant differences between the wild-type (WT) and GRAF1^{CKO} mice. Meanwhile, from an ethical perspective, our study adhered to the principles of the 3Rs (Replacement, Reduction, and Refinement) to minimize the use of animals and ensure their welfare. Our findings highlight the robustness of the observed effects and the statistical power achieved with the chosen sample size.

With respect to consideration of sex in our study design- our mechanistic studies performed in the HeLa cell line (from a female patient) revealed similar results as observed in Cos7 cells (derived from male CV-1 cells) and our primary cultured cardiomyocytes which were mixed cultures from male and female mice. We acknowledge the importance of investigating sex differences in pre-clinical studies. Importantly, our prior published studies that separately analyzed cardiac function in male and female GRAF1^{gt/gt} mice revealed no statistical differences due to GRAF1 depletion. However, we were required to dramatically reduce our mouse colony at the start of the pandemic and to keep numbers low for nearly two years' time as fewer animal handlers were on-site to care to provide care. To achieve this without losing valuable strains, we culled all non-breeding females from litters. In this case, given our previously published studies, and reports from others showing a lack of sexual dimorphism in isoproterenol-induced cardiac outcomes including heart rate, hypertrophy, dysfunction, and fibrosis, we would not expect a difference between males and female mice in this particular stress model.

4) Can you perform a further computational validation how likely it is that GRAF1 and Parkin interact via the proposed mechanism?

As noted above, the only high resolution structural data that are currently available for GRAF1 is of the isolated GRAF1 GAP domain. We have used computational modeling for the BAR-PH- and GAP domains using existing structures from related proteins, but at this juncture, we currently have no structural information that includes the serine-proline and SH3 domains which are necessary for modeling GRAF1 and Parkin interactions.

5) With regards to identifying mitophagy-dependent GRAF1 modifications, did you consider alternative proteases other than trypsin?

We originally chose trypsin because it is typically preferred for mass spec analysis as it has high specificity, generates peptides with C-terminal arginine and lysines (charges that are detectable by MS), and cleaves proteins into peptides with an average size of 700-1500 daltons (the ideal range for MS analysis). However, as noted in our manuscript digestion with trypsin only yielded 44% coverage of GRAF1. Since this technique failed to uncover any peptides differentially phosphorylated by PINK1/Parkin, and because peptides derived from the serine-proline domain were not captured, we explored other options. Specifically, we performed in silico digestion of GRAF1 using multiple proteases such as Asp-N, Chymotrypsin, and Lys-C, both individually and in combination. Unfortunately, none of these methods proved capable of generating appropriate peptide fragments (6-30 amino acids) to sufficiently cover this region. Thus, we proceeded with a combination of deletion and single amino acid mutation studies to identify the three PINK1-dependent phosphorylation sites described herein and subsequently validated that these were the major OA-dependent phosphorylation sites in GRAF1.

6) The validation in human heart tissue is important. How comparable are the patients that were used as non-failing controls vs HCM and DCM patients (age, sex, etc)? Could medication affect your read outs, i.e. beta blockers?

We appreciate your insightful comment regarding the comparability of the patient groups and the potential impact of medication on our study outcomes. The human heart samples used in our study were procured from the Duke Human Heart Repository (DHHR). These myocardial tissue samples were obtained from male patients aged 30-60 years with hypertrophic cardiomyopathy (HCM) (mean age 49.83 ± 4.75), dilated cardiomyopathy (DCM) (mean age 45.83 ± 9.7), and male non-failing (NF) controls (6 samples/group) (mean age 46.67 ± 8.01). While we aimed to select patient groups that were age-matched as closely as possible, we acknowledge that there might be inherent differences in age and other demographic factors between the HCM, DCM, and NF groups. Patient data at the time of tissue procurement was retrospectively collected by DHHR personnel from an electronic medical record database or from organ donor records. We have added this information to the manuscript to ensure transparency in our sample characteristics. Remarkably, not all DCM patients were treated with beta blockers, but given the wide range of treatments we cannot comment on the impact of these various treatments on the outcomes measured.

7) Also, the NF hearts may have been stored in a transplant solution? This could affect your protein phosphorylation levels.

Left ventricular (LV) cardiac tissue from HCM and DCM patients were obtained at the time of cardiac transplantation. LV cardiac tissue from NF subjects were obtained from deceased donors whose hearts were not used for transplantation. At the time of collection, the LV tissue was sharply dissected into 1-2 mm thick samples then flash frozen in a cryovial using liquid nitrogen. The tissue was stored at -80°C until it was utilized. As a result, we anticipate that the storage method employed in our study has had limited influence on the phosphorylation levels of GRAF1.

8) Figure 5K: It seems GRAF1 is reduced in HF patients? Did you also assess mitochondrial protein content in failing versus non-failing hearts?

Thank you for this suggestion. We assessed mitochondrial protein levels and indeed found a significant reduction in mitochondria in failing hearts (new Fig 5O, FigS3P). When taken together with the increased levels of GRAF1 phosphorylation and the reduced levels of total GRAF1, these findings are in keeping with disruptions in mitochondrial homeostasis in HF.

9) Figure 5I. The metabolite profiles look very variable between ISO-treated WT and GRAF1 conditional KO hearts? Did you control the fasting state of the mice? How do the changes relate to mitochondrial content/ morphology in WT and conditional KO hearts? How similar are there metabolite profiles at baseline without ISO treatment?

We did observe more variability in the metabolic profiles of ISO treated WT and GRAF1^{CKO} hearts than observed in analyses of mitochondrial content or mitophagy (as assessed by LC3 turnover or mt-Keima), and as pointed out by the reviewer this could be due, at least in part, to the fact that the mice were fed ad libitum, and we did not control for the fasting state of the mice. Nonetheless, metabolites from ISO treated WT and GRAF1^{CKO} hearts did stratify into two distinct clusters and subsequent KEGG pathway analysis identified 10 metabolic pathways that differed significantly between genotypes. Interestingly, most of the pathways elevated in WT hearts were associated with carbohydrate metabolism and energy production, whereas most of the pathways elevated in GRAF1^{CKO} hearts involved amino acid metabolism/catabolism. In further support of these data, less variable, though similar (albeit more modest) differences were observed in serum metabolites isolated from the ISO-treated mice (e.g. ISO-treated WT mouse serum exhibited significantly higher levels of metabolites derived from fructose/mannose metabolism, the pentose phosphate pathway and glycolysis/gluconeogenesis), whereas none of these serum metabolites were significantly different when comparing saline-treated WT and GRAF1^{CKO} mice (data not shown). Indeed, only 5 of the 224 metabolites detected in serum samples differed between saline treated animals, consistent with the notion that GRAF1-dependent processes are employed under conditions of high energy demand. The high levels of glycolytic intermediates in WT animals is consistent with prior studies showing that cardiac stress promotes a mitophagy-dependent change in fuel utilization from fatty acid oxidation to glycolysis and that the variability observed in GRAF1^{CKO} hearts likely resulted from differences in the compensatory mechanisms employed to maintain ATP levels.

10) It would strengthen the manuscript if also immunostainings could be performed on histological sections from human non-failing and failing hearts.

This is an excellent idea that we agree would further enhance the strength of our manuscript. Unfortunately, the human heart samples we obtained for this study were cryopreserved at -80 °C, rendering them unsuitable for histological sectioning and staining.

REVIEWER COMMENTS

Reviewer #1 (Remarks to the Author):

Overall, although the authors have made several revisions in the manuscript, fundamental concerns raised in the review of the previous submission have not been addressed.

1. The data supporting a decrease in autophagic flux and autophagosomes in response to depleting or silencing of GRAF1 in cardiomyocytes are sound. However, it is not clear that the decrease is due to effects on mitophagy as opposed to general macroautophagy. The new data in Fig. 1M, that mitochondria are encapsulated in and associated with autophagosomes in ISO-treated hearts, is not compelling.

a. The green arrow points to a structure that appears to be an autophagosome since its boundary membrane is a double membrane. However, based on visual inspection, it is not clear that the structure within the autophagosome is a mitochondrion. Thus, there may be changes in autophagosome number, as documented in SFig. J, but it is not clear that the structures scored are mitochondria-associated autophagosomes (mitophagosomes) as opposed to autophagosomes with diverse cargoes. The statement that there are fewer mitophagosomes in GRAF1CKO hearts (lines 159-162) should be modified to state that there are fewer autophagosomes.

b. Contrary to the statement in the figure legend, the structure associated with the yellow arrow does not appear to be an autophagosome: it is surrounded by a single membrane. In addition, that structure is not associated with the mitochondrion in the field. Rather, it appears to be associated with a thin tubular membrane-bound structure (ER?) and that may be associated with a mitochondrion.

c. Previous studies from this group revealed that a role for GRAF1 in limiting RhoA-dependent F-actin formation during maintenance of striated muscle tissues. While it is clear that 1) the steady state levels of mitochondrial marker proteins increased in response to OA treatment (SFig. 1K-L) and 2) autophagic flux is decreased by silencing of GRAF1 Si cardiomyocytes (SFig. 1H-I), it is not clear whether this is due to elevated mitophagy or general macroautophagy. Similarly, while it is clear that mitochondrial function is compromised by silencing of GRAF1 in cardiomyocytes (Fig. 1F-G), overall heart function is also compromised when GRAF1 is depleted (e.g. Fig. 1I-J and SFig. 1F-G). Is the decrease in autophagy in cardiomyocytes due to effects on mitochondria or to mitochondria-independent functions of GRAF1 in maintenance of heart muscle?

2. The evidence that silencing of GRAF results in a reduction in OA-induced mitophagy in Hela cells (Fig. 2A-B) is not compelling.

a. While it is clear that the steady state levels of mitochondrial marker proteins increase after 16 and 20 hrs of OA treatment of Hela cells in which GRAF1 is silenced, the levels of each of those markers is also elevated at 0 hrs of OA treatment in GRAF Si cells. Thus, while silencing GRAF may result in an increase in

the steady state level of mitochondrial marker proteins, OA-induced decreases in those markers relative to t=0 time of treatment, may not be affected by silencing of GRAF. Here, the relevant analysis is not total level of mitochondrial markers, but the change in the level of those markers relative to 0 time of treatment. It is not clear why quantitation was performed for t=16 and 20 hrs of OA treatment but not of t=0 hrs of OA treatment in Fig. 2A-B, while quantitation of mitochondrial marker proteins at all time points studied was carried out for SFig. 1K-L.

b. The same concern exists for Fig. 2C. Here too, quantitation was carried out for ATP5B staining at 16 hrs but not at 0 (or 20 hrs). Therefore, it is not clear whether silencing of GRAF has effects on ATP5B levels in untreated HeLa cells, or whether silencing of GRAF results in OA-associated changes in ATP5B levels.

c. The evidence that macroautophagy is critical for control of mitochondrial steady levels in response to OA-induced mitochondrial stress in HeLa cells (Fig. 2A-B) is not compelling. Deletion of ATG7, a key mediator of macroautophagy, has no significant effect on the levels of Tim50 and PHB2 in OA treated cells after 16 - 20 hours. There was an increase in Hsp60 levels at those timepoints in ATG7 Si vs WT cells. However, there was also an increase in Hsp60 at t=0 hrs of OA treatment. So, the increase observed is not due to defects in mitophagy produced by silencing of ATG7.

This concern was raised in the review of the previous submission. In the rebuttal to that critique, the authors claimed that deletion of ATG7 stated is not a good approach to inhibit macroautophagy. Specifically, they stated that “Our data are consistent with their data and others showing that ATG7 contributes to, but is not essential for mitophagy.” One of the take home messages from this paper is that GRAF1 regulates mitochondrial quality control through effects on mitophagic degradation of damaged mitochondria. If they want to make this claim, they need to provide compelling evidence that mitochondria are indeed degraded mitophagy under the conditions studied. If deletion of ATG7 is not a sound approach to determine whether mitophagy is being regulated by GRAF1, then they should identify another approach (e.g. deletion other core ATG genes). For example, the authors could identify an ATG gene, which when deleted inhibits OA-induced mitophagy in cells of interest (e.g. detected using mtKeima), and determine the effect of this deletion on OA-induced, GRAF1-dependent changes in mitochondria.

Overall, while silencing of GRAF1 may impact on mitochondrial abundance in HeLa cells, it is not clear that the observed effects on mitochondrial abundance are due to GRAF1 function in regulating mitophagy. This fundamental concern was raised in the review of the previous submission that has not been addressed in the revised manuscript.

3. One fundamental concern that was raised in the review of the previous submission is whether HeLa cells are a good model for cardiomyocytes.

a. In the previous critique, this reviewer stated that localization of GRAF1 in NRVM cells in response to OA treatment is distinct from GRAF1 localization in HeLa cells exposed to OA treatment. In response to this concern, the authors added images of GRAF1 and Parkin, and GRAF1 and Tomm20 in OA-treated NRVM cells to provide better documentation GRAF1 localizes to mitochondria in cardiomyocytes as in HeLa cells (SFig. 2N-O). However, the new data is not compelling. First, it is not supported by quantitative

analysis. Second, the localization of GRAF1 and mitochondrial surface makers (Parkin or Tomm20) in the new data are not consistent with the claim that GRAF1 is recruited to the surface of damaged mitochondria. In the images shown, GRAF1-containing structures are a fundamentally different in shape compared Parkin- or Tomm20-containing structures in OA-treated NRVM cells. This suggests that GRAF1 is not localized to the same structures that contain Parkin or Tomm20. Thus, the new data do not support recruitment of GRAF1 to the surface of mitochondria in response to OA treatment in NRVM cells. Thus, the fundamental concern that HeLa cells are not a good model for heart has not been addressed.

b. Further evaluation of the data revealed another case in which HeLa cells exhibited a phenotype that is distinct from that of cardiomyocytes. Specifically, comparison of mitochondrial marker protein levels at $t=0$ for OA treatment in cardiomyocytes (SFig. 2K-L) and HeLa cells (Fig. 2A-C) indicates that silencing of GRAF1 results in an increase in the mitochondria in HeLa cells but not in cardiomyocytes. This supports a possible role for GRAF1 in regulating mitophagy in heart under the conditions used. However, as described above, the evidence for mitophagy in control of mitochondrial abundance in response to OA treatment in HeLa cells is not compelling in the original submission or in the revision under consideration. Thus, the fundamental concern that HeLa cells may not be a good model for cardiomyocytes remains unaddressed.

4. As described above, there are concerns regarding the using HeLa as a model for heart. The authors also used COS7 cells for some studies or varied the conditions for stress exposure (variable times of treatment with OA or use of CCCP as a stressor instead of OA). Moreover, there were no controls provided to document confirm that COS7 cells are good models for heart, or that the variable stress conditions used in these studies are comparable. As a result of these issues with experimental design, it is difficult to interpret the findings obtained.

a. The evidence that Parkin co-IPs with GRAF1 was carried out in COS7 cells, and not heart cells. In addition, the conditions for treatment of COS7 cells with OA or for silencing GRAF1 are not described. If the authors want to switch to COS7 cells for IP analysis, they need to provide evidence that silencing of GRAF1 and OA treatment of GRAF1 silenced and non-silenced COS7 cells produce similar effects on mitochondrial function and localization, macroautophagy, mitophagy or GRAF1 function in control of actin organization in heart. In the absence of that, the evidence that OA treatment induces interactions between GRAF1 and Parkin is not compelling.

b. Fig. 3A show results from experiments carried out +/- 2 hrs of CCCP treatment. Fig. 3A is a western of GRAF1, YFP-Parkin, OPTN, tubulin and VDAC in subcellular fractions (cytosol and mitochondria) isolated from HeLa cells +/- 2 hrs of CCCP treatment. GRAF1 appears to be recovered with mitochondria in response to 2 hrs CCCP treatment in control cells. However, with 2 hrs of CCCP treatment, there is no obvious change in the amount of Parkin that is recovered with mitochondria; however, there appears to be an increase in recovery of optineurin with mitochondria. Thus, it is not clear whether 2 hrs of CCCP treatment induces mitophagy or the relationship between 2 hrs of CCCP treatment and 16-20 hrs of OA treatment. In light of this, it is not clear how to interpret the changes in GRAF1 recovery with mitochondria or the electrophoretic mobility of GRAF1 under experimental conditions used.

c. Fig. 2G shows that 2 hrs of CCCP treatment in HeLa cells results in co-IP of Parkin with LC3, and silencing of GRAF1 reduces the amount of Parkin that is recovered with immunoprecipitated LC3. As described in the critique of the previous submission, Parkin is overexpressed in these studies. LC3 is also ectopically expressed. So, it is not clear whether the observed co-immunoprecipitation of Parkin with LC3 is physiologically relevant. In addition, as described above, it is not clear whether the CCCP conditions used induce mitophagy. Therefore, there are fundamental concerns regarding the relevance of the findings obtained after 2 hrs of CCCP treatment to the stress response associated with 16-20 hrs of treatment with OA.

5. The evidence that GRAF1 is phosphorylated in response to OA treatment, and a role for Pink1 in that process and Pink1-dependent clustering of mitochondria in the perinuclear region of HeLa cells are largely sound. However, there are concerns (in addition to concerns regarding the use of HeLa cells as a model for heart).

a. In Fig. 3I-J, the authors expressed non-phosphorylatable GRAF1 (GRAF1 AAA) in HeLa cells in which endogenous GRAF1 has been silenced and studied mitochondrial clustering by quantitation of ATP5B (Fig. 3I-J). There is no description of how clustering of mitochondria was quantitated in Fig. 3J. However, the Y axis label (ATP5B stain remaining) and graph layout resemble that shown for quantitation of mitochondrial abundance (e.g. Fig. 2C) and are clearly distinct from the quantitation of mitochondrial clustering in Fig. 4I where mitochondrial distribution was assessed by score the percentage of total mitochondria that exhibit no clustering, diffuse clustering or focal clustering. Does expression of AAA GRAF1 in GRAF1 silenced cells restore mitochondrial abundance or distribution?

b. Regardless of the nature of the assay shown in Fig. 3J, it was carried out after 16 hrs of OA treatment. In contrast, phosphorylation studies were carried out with 0-6 hrs of OA treatment. Indeed, in Fig. 2D, the authors show localization of GRAF1 and Parkin after 4 hrs of OA treatment in HeLa cells. Parkin localizes to punctate structures which are presumably mitochondria. However, there is little or no perinuclear clustering of mitochondria at 4 hrs of OA treatment (Fig. 2D). In contrast, clustering is evident after 16 hrs of OA treatment (Fig. 3I). Similarly, treatment with CK-666, an inhibitor of the Arp2/3 complex, had no effect on OA-induced changes in mitochondrial abundance after 16 hrs of CK-666 treatment but did show a subtle but significant effect after 20 hrs of treatment (SFig. 3L-M). Studies on GRAF1 phosphorylation and the effect of GRAF1 phosphorylation on OA-induced stress responses need to be carried out under the same experimental conditions.

6. The data that GRAF1 interacts (and co-IPs) with ABI2 and is critical for recruitment of ABI2, WAVE2 and CYFIP1 to mitochondria seem sound. However, since GRAF1 has been implicated in regulation of actin organization at sites other than mitochondria, the manuscript would benefit from more than one line of evidence that the Arp2/3 complex and its modulator affects the organization of actin associated with mitochondria and do so in a GRAF1-dependent manner.

7. The studies carried out in cardiomyocytes provide more support for a role for GRAF1 in actin organization and mitochondrial localization. However, the changes in actin organization under the conditions studied occur throughout the cell. Therefore, it is not clear whether effects on mitochondria are due to general effects on actin organization or GRAF1 at the mitochondrial surface.

Reviewer #2 (Remarks to the Author):

This is a much improved version of the manuscript and there are no further comments.

Reviewer #4 (Remarks to the Author):

I still have some minor points which are listed below:

- Although the authors have performed in vitro experiments in cells obtained from both males and females, this can't recapitulate the sex-related differences occurring in vivo which are mainly due to different hormonal profiles. As only male mice were employed for in vivo experiments, this should be stated as a limitation of the study in the Discussion.
- Regarding the human data included in Fig. 5, an additional Table should be added to the Supplementary material providing the relevant characteristics of each patient separately (i.e., sex, age, known medications etc.).
- As the metabolite profiles in Fig. 5 M look very variable also between mice from the same group, it would be interesting to know if they would still group separately after an unsupervised clustering analysis.
- For more clarity, the mouse model GRAF1gt/gt should be better explained in the main text (i.e., in line 134 when it is mentioned for the first time).
- A quantification of total LC3 levels as mentioned in the main text (lines 165-167) should be provided in Figure 1 K, L.
- In the first Results section, the authors only observe impaired heart functionality in GRAFCKO mice upon isoproterenol administration. However, in Fig. 1 K and L LC3II/LC3I ratios in saline treated mice are reduced to the same extent than in ISO treated mice. If an impaired mitophagy is thought to be the cause of the impaired heart functionality, how do the authors justify this discrepancy?
- In line 169, the text should be amended "as shown in REPRESENTATIVE figure 1M", since no quantification is provided for this analysis.
- At the end of page 9, a short explanation should be added about the role of Atg7 in the proposed model. Also, full names for acronyms should always be provided the first time they are used in the manuscript.

General Response to Reviewers:

We thank the reviewers for their positive comments and further suggestions for the improvement of our manuscript. Reviewer 2 previously remarked that our study was comprehensive and meticulous and had no further concerns. Reviewer 3 previously remarked that our manuscript was “very well written, elegantly combines a range of different techniques and the methods are very detailed” and had a few additional minor concerns which we address below and discuss in our revised manuscript. Reviewer 1 appreciated prior revisions, but still had some fundamental concerns that we now address below and in our revised manuscript using a combination of new experiments, additional quantification, and additional discussion points. Collectively our new findings provide further rigorous support for our contention that GRAF1 plays a major role in promoting Pink1/Parkin-dependent mitophagy in both HeLa cells and cardiomyocytes.

In the next section we will summarize major findings. Additional detail can be found in our subsequent point-by-point response.

1. ***Validation of YFP-Parkin/HeLa cell model to assess selective mitophagy versus general macroautophagy.*** In response to concerns addressed by Reviewer 1, we performed side-by-side experiments in stable YFP/Parkin-expressing HeLa cells and in the parent sHeLa (Parkin-deficient) cell line that was used to generate the former (both lines were obtained by the Youle laboratory). As now shown in new Figure S2C and D, treatment of the Parkin-deficient sHeLa cells with oligomycin-A and antimycin-A (OA) to induce respiratory-failure mediated mitochondrial damage did not lead to significant degradation of the mitochondrial inner membrane protein, Tim50 or the mitochondrial matrix protein, Hsp60. In contrast, OA treatment of YFP/Parkin-expressing HeLa cells induced marked degradation of both Tim50 and Hsp60, confirming that this response was Parkin-dependent. As further confirmation that this model involves autophagosome-dependent clearance, we now show that depletion of the core autophagosomal elongation protein, ATG12 fully attenuated OA-mediated degradation of inner membrane and matrix proteins in YFP/Parkin HeLa cells (Figure S2A,B). Notably, while ATG12 is known to get covalently linked to ATG5 to drive the expansion and formation of the autophagosome, previous studies identified another phagophore-elongation regulator, ATG3 as an additional receptor for ATG12 and further showed that lack of the covalent ATG12/3 complex formation selectively impaired mitophagy (Cell 2010 Aug 20;142(4):590-600), a finding consistent with our new data. With respect to induction of general autophagy versus mitophagy in this model, we now show that while OA treatment of YFP/Parkin- HeLa cells leads to significant degradation of mitochondrial inner membrane and matrix proteins, it led to a slight but significant increase in levels of p62 (also known as Sequestosome 1, or SQSTM1) a substrate primarily degraded by general autophagy (new figure S2E). This finding suggests that under the conditions of OA-induced mitophagy, the general autophagy machinery undergoes a shift from the conventional autophagy pathway to a selective clearance pathway, specifically aimed at degrading damaged mitochondria. Taken together, these

data provide strong support for use of the YFP/Parkin-Hela cells to study selective autophagic removal of mitochondria (mitophagy).

2. **Validation of selective role for GRAF1 in promoting mitophagy.** We also performed additional GRAF1 knock-down studies in Parkin-deficient sHela and YFP/Parkin Hela cells. As shown in new Figure S2C and D, GRAF1 depletion had no significant impact on the levels of the mitochondrial inner membrane protein, Tim50 or the matrix protein, Hsp60 in vehicle or OA-treated sHela cells. In contrast, GRAF1 depletion (like ATG12 depletion) attenuated OA-induced degradation of Tim50 and Hsp60 in Parkin-expressing Hela cells, indicating that GRAF1 is involved in Parkin mediated mitochondria clearance (mitophagy). When taken together with our prior findings that GRAF1 phosphorylation is induced to a much greater extent following OA treatment than following induction of general autophagy in YFP-Parkin/Hela cells (Fig S3B), that GRAF1 is phosphorylated in a PINK1/Parkin-dependent fashion in both YFP-Parkin/Hela cells and in cardiomyocytes (Fig3G, FigS3C, Fig 5C and D); that mitochondrial depolarization promotes GRAF1 localization to the OMM in both Parkin/Hela cells and in cardiomyocytes (Fig S1P and Q, Fig S2H,), that GRAF1 is required for targeting of mitochondria to lysosomes in both YFP-Parkin/Hela cells, and that GRAF1 is required for mitochondrial inner membrane protein and matrix protein degradation (but not for MFN1 degradation, a mitochondria outer membrane protein reported to be degraded by proteasome when mitochondria were damaged) in both cell types, (FigS1K, FigS2A), these data strongly support our contention that GRAF1 selectively promotes PINK1-Parkin dependent mitophagy in settings in which mitochondrial membrane depolarization or dysfunction is induced. That said, it is formally possible that GRAF1 plays additional roles in autophagosome formation (as previously suggested by others) and as such may also impact general autophagy in conditions that favor this response and we now discuss this possibility in the discussion of the revised manuscript.
3. **Limitations of our study-** While our temporal approaches provide strong evidence for GRAF1 acting down-stream of Parkin to regulate autophagosome recruitment by altering localized actin remodeling and promoting Parkin-LC3 association, we acknowledge that the future application of high powered, real time imaging to track the fates of individual GRAF1-labeled mitochondria will be necessary to provide further support for this conclusion and we have added additional discussion of this limitation to our revised manuscript. We also address other potential limitations of our in vivo studies including the potential for additional RhoA-dependent effects that could contribute to cardiac dysfunction in our model and the fact that we did not include females in our study (see below for more detail).

Specific Responses: In the following section we highlight specific comments/queries from reviewers in yellow and our responses are in blue text.

1. The data supporting a decrease in autophagic flux and autophagosomes in response to depleting or silencing of GRAF1 in cardiomyocytes are sound. However, it is not clear that the decrease is due to effects on mitophagy as opposed to general macroautophagy. The new data in Fig. 1M, that

mitochondria are encapsulated in and associated with autophagosomes in ISO-treated hearts, is not compelling.

a. The green arrow points to a structure that appears to be an autophagosome since its boundary membrane is a double membrane. However, based on visual inspection, it is not clear that the structure within the autophagosome is a mitochondrion. Thus, there may be changes in autophagosome number, as documented in SFig. J, but it is not clear that the structures scored are mitochondria-associated autophagosomes (mitophagosomes) as opposed to autophagosomes with diverse cargoes. The statement that there are fewer mitophagosomes in GRAF1CKO hearts (lines 159-162) should be modified to state that there are fewer autophagosomes. b. Contrary to the statement in the figure legend, the structure associated with the yellow arrow does not appear to be an autophagosome: it is surrounded by a single membrane. In addition, that structure is not associated with the mitochondrion in the field. Rather, it appears to be associated with a thin tubular membrane-bound structure (ER?) and that may be associated with a mitochondrion.

We agree with this reviewer and with a former publication that we co-authored entitled “Guidelines for the use and interpretation of assays for monitoring autophagy- 4th edition” that it can be difficult to detect the difference between general and selective autophagy by TEM. We used spatial information to quantify the autophagosomes that were associated with mitochondria in cardiac TEM images and now more appropriately refer to these structures as “autophagic vacuoles” which are typical of the structures reported by others found in cardiomyocytes in patients with dilated cardiomyopathy (Kanamori H, et al. *Autophagy*. 2015, 11(7):1146-60). As described above, we now provide several additional pieces of evidence to support our thesis that the major impact of GRAF1 is to promote selective mitophagy in conditions wherein the PINK1/Parkin pathway is active.

c. Previous studies from this group revealed that a role for GRAF1 in limiting RhoA-dependent F-actin formation during maintenance of striated muscle tissues. While it is clear that 1) the steady state levels of mitochondrial marker proteins increased in response to OA treatment (SFig. 1K-L) and 2) autophagic flux is decreased by silencing of GRAF1 Si cardiomyocytes (SFig. 1H-I), it is not clear whether this is due to elevated mitophagy or general macroautophagy. Similarly, while it is clear that mitochondrial function is compromised by silencing of GRAF1 in cardiomyocytes (Fig. 1F-G), overall heart function is also compromised when GRAF1 is depleted (e.g. Fig. 1I-J and SFig. 1F-G). Is the decrease in autophagy in cardiomyocytes due to effects on mitochondria or to mitochondria-independent functions of GRAF1 in maintenance of heart muscle?

i. As noted above, we have performed additional experiments to confirm and extend our hypothesis that GRAF1 plays a major role in promoting mitophagy rather than general macroautophagy in the HeLa cell and cardiomyocyte model systems used herein.

ii. We want to highlight that in all model systems tested, depletion of GRAF1 proteins led to a physiologically relevant (~2 fold) and transient enhancement of endogenous RhoA activity. As these protein facilitate RhoA GTP hydrolysis, they enhance and prolong RhoA in response to agonist treatment but do not lead to constitutive (un-regulated increase in activity). While ~20 fold over expression of constitutively active RhoA in cardiomyocytes can, over time, lead to heart failure and development of dilated cardiomyopathy, moderate (~2-5 fold) overexpression of constitutively active RhoA does not have a negative impact on cardiac function; nor does depletion of GRAF1 alone. We chose an isoproterenol challenge experiment to determine the extent to which GRAF1 might be necessary for autophagy/mitophagy in vivo, since prior studies from others showed that cardiac-specific Atg5 knockout mice exhibited no cardiac phenotypes

under baseline conditions, but developed cardiac dysfunction following treatment with isoproterenol (see Ref 36 in manuscript). Similarly, we found that cardiac-restricted GRAF1 knock-out mice developed heart rate-independent cardiac dysfunction following isoproterenol (ISO) treatment. Moreover, we showed that ISO induced cardiac mitophagy within 3 days and that this outcome was attenuated in GRAF1 null mice. When combined with further evidence that damaged mitochondria accumulated in GRAF1 knock-out mice and that GRAF1-deficient ISO-treated hearts exhibited accompanying defects in metabolic fuel flexibility, we reason that the decrease in heart function is due, at least in part, to a decrease in mitochondrial autophagy. However, it is formally possible that additional RhoA-dependent pathways are involved and this possibility is now noted in a new limitations section in the discussion of our revised manuscript.

2. The evidence that silencing of GRAF results in a reduction in OA-induced mitophagy in HeLa cells (Fig. 2A-B) is not compelling.

a. While it is clear that the steady state levels of mitochondrial marker proteins increase after 16 and 20 hrs of OA treatment of HeLa cells in which GRAF1 is silenced, the levels of each of those markers is also elevated at 0 hrs of OA treatment in GRAF Si cells. Thus, while silencing GRAF may result in an increase in the steady state level of mitochondrial marker proteins, OA-induced decreases in those markers relative to t=0 time of treatment, may not be affected by silencing of GRAF. Here, the relevant analysis is not total level of mitochondrial markers, but the change in the level of those markers relative to 0 time of treatment. It is not clear why quantitation was performed for t=16 and 20 hrs of OA treatment but not of t=0 hrs of OA treatment in Fig. 2A-B, while quantitation of mitochondrial marker proteins at all time points studied was carried out for SFig. 1K-L. b. The same concern exists for Fig. 2C. Here too, quantitation was carried out for ATP5B staining at 16 hrs but not at 0 (or 20 hrs). Therefore, it is not clear whether silencing of GRAF has effects on ATP5B levels in untreated HeLa cells, or whether silencing of GRAF results in OA-associated changes in ATP5B levels.

We agree that the rate of marker degradation is important and apologize for not previously providing quantification of all time points. As is now more clear from these additional analyses, GRAF1 depletion does not significantly alter levels of mitochondrial inner membrane or matrix proteins in vehicle-treated cells at t=0 as assessed by Western analysis (Fig2 A and B) or as assessed by ATP5B staining (Fig 2C).

c. The evidence that macroautophagy is critical for control of mitochondrial steady levels in response to OA-induced mitochondrial stress in HeLa cells (Fig. 2A-B) is not compelling. Deletion of ATG7, a key mediator of macroautophagy, has no significant effect on the levels of Tim50 and PHB2 in OA treated cells after 16 - 20 hours. There was an increase in Hsp60 levels at those timepoints in ATG7 Si vs WT cells. However, there was also an increase in Hsp60 at t=0 hrs of OA treatment. So, the increase observed is not due to defects in mitophagy produced by silencing of ATG7.

This concern was raised in the review of the previous submission. In the rebuttal to that critique, the authors claimed that deletion of ATG7 stated is not a good approach to inhibit macroautophagy. Specifically, they stated that "Our data are consistent with their data and others showing that ATG7 contributes to, but is not essential for mitophagy." One of the take home messages from this paper is that GRAF1 regulates mitochondrial quality control through effects on

mitophagic degradation of damaged mitochondria. If they want to make this claim, they need to provide compelling evidence that mitochondria are indeed degraded mitophagy under the conditions studied. If deletion of ATG7 is not a sound approach to determine whether mitophagy is being regulated by GRAF1, then they should identify another approach (e.g. deletion other core ATG genes). For example, the authors could identify an ATG gene, which when deleted inhibits OA-induced mitophagy in cells of interest (e.g. detected using mtKeima), and determine the effect of this deletion on OA-induced, GRAF1-dependent changes in mitochondria.

Overall, while silencing of GRAF1 may impact on mitochondrial abundance in Hela cells, it is not clear that the observed effects on mitochondrial abundance are due to GRAF1 function in regulating mitophagy. This fundamental concern was raised in the review of the previous submission that has not been addressed in the revised manuscript.

Please see general response points 1 and 2 above.

3. One fundamental concern that was raised in the review of the previous submission is whether Hela cells are a good model for cardiomyocytes.

a. In the previous critique, this reviewer stated that localization of GRAF1 in NRVCN cells in response to OA treatment is distinct from GRAF1 localization in Hela cells exposed to OA treatment. In response to this concern, the authors added images of GRAF1 and Parkin, and GRAF1 and Tomm20 in OA-treated NRVCN cells to provide better documentation GRAF1 localizes to mitochondria in cardiomyocytes as in Hela cells (SFig. 2N-O). However, the new data is not compelling. First, it is not supported by quantitative analysis. Second, the localization of GRAF1 and mitochondrial surface makers (Parkin or Tomm20) in the new data are not consistent with the claim that GRAF1 is recruited to the surface of damaged mitochondria. In the images shown, GRAF1-containing structures are a fundamentally different in shape compared Parkin- or Tomm20-containing structures in OA-treated NRVCN cells. This suggests that GRAF1 is not localized to the same structures that contain Parkin or Tomm20. Thus, the new data do not support recruitment of GRAF1 to the surface of mitochondria in response to OA treatment in NRVCN cells. Thus, the fundamental concern that Hela cells are not a good model for heart has not been addressed. b. Further evaluation of the data revealed another case in which Hela cells exhibited a phenotype that is distinct from that of cardiomyocytes. Specifically, comparison of mitochondrial marker protein levels at t=0 for OA treatment in cardiomyocytes (SFig. 2K-L) and Hela cells (Fig. 2A-C) indicates that silencing of GRAF1 results in an increase in the mitochondria in Hela cells but not in cardiomyocytes. This supports a possible role for GRAF1 in regulating mitophagy in heart under the conditions used. However, as described above, the evidence for mitophagy in control of mitochondrial abundance in response to OA treatment in Hela cells is not compelling in the original submission or in the revision under consideration. Thus, the fundamental concern that Hela cells may not be a good model for cardiomyocytes remains unaddressed.

Please see points 1 and 2 in the general response above. Also, we acknowledge that there are differences between the localization, size, shape and dynamics of mitochondria in cardiomyocytes and Hela cells before and after treatment with mitochondrial poisons and that the kinetics and spatial-temporal locale of OA-dependent mitophagy differs between these cell types. Nonetheless, despite these differences our data strongly support a major role for GRAF1

in facilitating Parkin-dependent mitochondrial clearance in both cell types- a fact that broadens the applicability of our study. With respect to OA-induced outer mitochondrial membrane (OMM translocation of GRAF1), in addition to showing that GRAF1 is phosphorylated by PINK1 following OA treatment (a kinase known to associate with the OMM of damaged mitochondria), and that phosphorylated GRAF1 selectively accumulates in mitochondrial fractions, we show that following OA-treatment, a portion of endogenous GRAF1 and TOMM20 co-localize on the mitochondrial surface in both cell types (FigS1 P, Q and FigS2H). Co-localization was further confirmed using single-plane line scan intensity profiles (see new FigS1O and Q).

4. As described above, there are concerns regarding the using HeLa as a model for heart. The authors also used COS7 cells for some studies or varied the conditions for stress exposure (variable times of treatment with OA or use of CCCP as a stressor instead of OA). Moreover, there were no controls provided to document confirm that COS7 cells are good models for heart, or that the variable stress conditions used in these studies are comparable. As a result of these issues with experimental design, it is difficult to interpret the findings obtained. a. The evidence that Parkin co-IPs with GRAF1 was carried out in COS7 cells, and not heart cells. In addition, the conditions for treatment of COS7 cells with OA or for silencing GRAF1 are not described. If the authors want to switch to COS7 cells for IP analysis, they need to provide evidence that silencing of GRAF1 and OA treatment of GRAF1 silenced and non-silenced COS7 cells produce similar effects on mitochondrial function and localization, macroautophagy, mitophagy or GRAF1 function in control of actin organization in heart. In the absence of that, the evidence that OA treatment induces interactions between GRAF1 and Parkin is not compelling. b. Fig. 3A show results from experiments carried out +/- 2 hrs of CCCP treatment. Fig. 3A is a western of GRAF1, YFP-Parkin, OPTN, tubulin and VDAC in subcellular fractions (cytosol and mitochondria) isolated from HeLa cells +/- 2 hrs of CCCP treatment. GRAF1 appears to be recovered with mitochondria in response to 2 hrs CCCP treatment in control cells. However, with 2 hrs of CCCP treatment, there is no obvious change in the amount of Parkin that is recovered with mitochondria; however, there appears to be an increase in recovery of optineurin with mitochondria. Thus, it is not clear whether 2 hrs of CCCP treatment induces mitophagy or the relationship between 2 hrs of CCCP treatment and 16-20 hrs of OA treatment. In light of this, it is not clear how to interpret the changes in GRAF1 recovery with mitochondria or the electrophoretic mobility of GRAF1 under experimental conditions used. c. Fig. 2G shows that 2 hrs of CCCP treatment in HeLa cells results in co-IP of Parkin with LC3, and silencing of GRAF1 reduces the amount of Parkin that is recovered with immunoprecipitated LC3. As described in the critique of the previous submission, Parkin is overexpressed these studies. LC3 is also ectopically expressed. So, it is not clear whether the observed co-immunoprecipitation of Parkin with LC3 is physiologically relevant. In addition, as described above, it is not clear whether the CCCP conditions used induce mitophagy. Therefore, there are fundamental concerns regarding the relevance of the findings obtained after 2 hrs of CCCP treatment to the stress response associated with 16-20 hrs of treatment with OA.

Cos7 cells are used in our manuscript, and in the field in general, to explore putative protein-protein interactions and to further define protein interaction sites. Cos7 cells are readily transfected with cDNAs and were used herein to show that GRAF1 is necessary to bridge active Parkin with LC3; a finding validated by co-localization studies in isolated cardiomyocytes and further confirmed in vivo using the mitoKeima mouse line. Cos7 cells were also used for mapping studies which revealed that GRAF1 associates with the linker region of Parkin that is exposed upon Parkin activation. These studies suggested that GRAF1 acts down-stream of

Parkin activation and this finding was further validated by our mitochondrial fractionation assays in HeLa cells. We confirmed that Cos7 cells were a reliable model since we showed that mitochondrial depolarization by CCCP or mitochondrial poisoning by OA led to Parkin activation at the time points assessed as indicated by the slower migrating forms of ubiquitinated Parkin (now highlighted by arrowheads in relevant figures, Fig2 E and F, FigS2J and K) and/or a later timepoints by conversion of LC3I to LC3II. Moreover, these same treatments in Cos7 cells promoted GRAF1 phosphorylation on the identical sites as found in OA- and CCCP-treated Parkin/HeLa cells; OA-treated NRVCs; ISO-treated hearts and in hearts from patients with human heart failure.

With respect to the concerns raised by the reviewer in point 4 and below in point 5 regarding use of various time points when assessing the outcomes of transient signals, I would like to note that we have many years of experience in the signal transduction field. It is well known from our studies and others that phosphorylation can initiate a cascade of events that culminate in a response many hours later. Take for example phosphorylation of ERK following treatment of cells with EGF. Maximal ERK phosphorylation is typically observed 5 to 10 minutes following EGF treatment- a signal that leads over time to ERK accumulation in the nucleus and at even later times to ERK-dependent increases in gene expression. If one were to examine phospho-ERK 24 hours after EGF treatment, there would be no significant elevation at this time point when a significant change in ERK-dependent gene expression is maximal. That does not mean that phosphorylation is not important. In fact, if the initial burst in phosphorylation is blocked then the impact on gene expression is blocked. The key to studying signal transduction in a rigorous fashion is to identify the window within which the particular outcome you are investigating is occurring in a linear fashion and then to test your variable numerous times at that same time point. We knew from prior rigorous studies from the Youle lab that Parkin is recruited to mitochondria as early as 15-30 minutes following OA or CCCP treatment in the HeLa/Parkin model and continues to increase over a ~2-4 hr period during which time other adapter proteins and then LC3 are recruited. Thus a 2 hour time point was chosen to see if GRAF1 accumulated on the mitochondria during this relevant time frame and if in doing so it also impacted Parkin recruitment. As noted in the manuscript, the former occurred but the latter did not. We further showed that this was a valid time point, since increased levels of ubiquitinated Parkin were observed in the mitochondrial fractions of CCCP-treated cells (note arrowhead in Fig3A). In our hands and in other reports, CCCP and OA both promote dose-dependent increases in mitophagy in Parkin/HeLa cells, but the OA response is slightly delayed. We used both drugs in our studies to show that either membrane depolarization (by CCCP) or ETC uncoupling (by OA) can promote GRAF1 phosphorylation and recruitment to mitochondria.

The finding that GRAF1 did not impact Parkin recruitment was consistent with our next findings that GRAF1 interacts with active Parkin and acts downstream to control the next notable step of mitophagy which is mitochondrial clustering. We performed additional time-dependent analyses to identify appropriate time points to analyze mitochondrial clustering in HeLa/Parkin cells and cardiomyocytes (4-6 hr) and found that GRAF1 depletion significantly impacted this step. We next wanted to be sure that there was a functional consequence, so we next followed mitochondrial clearance and/or mitochondrial-lysosome fusion (via mitoKeima). Mitochondrial clearance occurs over a prolonged time frame from 8-24 hours, however our studies indicated that this response was particularly robust in the 16-20 hr window. These later time points were also a bit more amenable than a 12 hr time point when considering work-life balance for the investigators performing these complicated studies. For the mitochondrial clearance assays we used both populational studies (i.e. Western blots to quantify IMM and matrix proteins) and validated cell imaging approaches to quantify mitochondrial abundance.

5. The evidence that GRAF1 is phosphorylated in response to OA treatment, and a role for Pink1 in that process and Pink1-dependent clustering of mitochondria in the perinuclear region of HeLa cells are largely sound. However, there are concerns (in addition to concerns regarding the use of HeLa cells as a model for heart). a. In Fig. 3I-J, the authors expressed non-phosphorylatable GRAF1 (GRAF1 AAA) in HeLa cells in which endogenous GRAF1 has been silenced and studied mitochondrial clustering by quantitation of ATP5B (Fig. 3I-J). There is no description of how clustering of mitochondria was quantitated in Fig. 3J. However, the Y axis label (ATP5B stain remaining) and graph layout resemble that shown for quantitation of mitochondrial abundance (e.g. Fig. 2C) and are clearly distinct from the quantitation of mitochondrial clustering in Fig. 4I where mitochondrial distribution was assessed by score the percentage of total mitochondria that exhibit no clustering, diffuse clustering or focal clustering. Does expression of AAA GRAF1 in GRAF1 silenced cells restore mitochondrial abundance or distribution?

For these particular experiments, we had previously focused on the final functional outcome of mitochondrial clearance by assessing mitochondrial abundance, but now show that, as expected GRAF1 AAA mutant also impacts mitochondrial clustering (see new Fig3K)

b. Regardless of the nature of the assay shown in Fig. 3J, it was carried out after 16 hrs of OA treatment. In contrast, phosphorylation studies were carried out with 0-6 hrs of OA treatment. Indeed, in Fig. 2D, the authors show localization of GRAF1 and Parkin after 4 hrs of OA treatment in HeLa cells. Parkin localizes to punctate structures which are presumably mitochondria. However, there is little or no perinuclear clustering of mitochondria at 4 hrs of OA treatment (Fig. 2D). In contrast, clustering is evident after 16 hrs of OA treatment (Fig. 3I). Similarly, treatment with CK-666, an inhibitor of the Arp2/3 complex, had no effect on OA-induced changes in mitochondrial abundance after 16 hrs of CK-66 treatment but did show a subtle but significant effect after 20 hrs of treatment (SFig. 3L-M). Studies on GRAF1 phosphorylation and the effect of GRAF1 phosphorylation on OA-induced stress responses need to be carried out under the same experimental conditions.

See response to point 4 above for our valid and rigorous approach to studying signal transduction cascades.

6. The data that GRAF1 interacts (and co-IPs) with ABI2 and is critical for recruitment of ABI2, WAVE2 and CYFIP1 to mitochondria seem sound. However, since GRAF1 has been implicated in regulation of actin organization at sites other than mitochondria, the manuscript would benefit from more than one line of evidence that the Arp2/3 complex and its modulator affects the organization of actin associated with mitochondria and do so in a GRAF1-dependent manner.

We provide a few lines of evidence to show that the Arp2/3-GRAF1 connection is robust, regulated, and important. We identified this interaction in an unbiased screen for phosphorylation-dependent binding partners and validated this in 2 different cell types. As described, GRAF1 phosphorylation is essential for mitochondrial clustering and clearance. We show that Arp2/3 inhibitors mirror the impact of GRAF1 depletion on mitochondrial clustering and clearance. We also show that GRAF1 depletion impairs OA-dependent recruitment of Arp2/3 activators (ABI2, Wave2, CYFIP1) to mitochondria and leads to altered mitochondrial associated branched actin formation (an Arp2/3-dependent process) and reduced mitochondrial

clustering that is rescued by actin modifying drugs. We realize that it is formally possible that additional mechanisms are involved and have described this possible limitation in the discussion.

7. The studies carried out in cardiomyocytes provide more support for a role for GRAF1 in actin organization and mitochondrial localization. However, the changes in actin organization under the conditions studied occur throughout the cell. Therefore, it is not clear whether effects on mitochondria are due to general effects on actin organization or GRAF1 at the mitochondrial surface.

Our studies support a mechanism wherein GRAF1 modulates mitophagy by both inducing general F-actin depolymerization and by promoting mitochondrial-localized actin formation. However, we appreciate that future use of specialized real-time 3D imaging approaches will be necessary to simultaneously track actin polymerization and mitochondrial dynamics during mitophagy. Indeed, this is a general limitation in the field and such studies will be important moving forward to answer many questions that remain with respect to the spatial and temporal control of this process, particularly in highly specialized cells like cardiomyocytes and neurons.

Reviewer #2 (Remarks to the Author):

This is a much improved version of the manuscript and there are no further comments.

Reviewer #4 (Remarks to the Author):

I still have some minor points which are listed below:

- Although the authors have performed in vitro experiments in cells obtained from both males and females, this can't recapitulate the sex-related differences occurring in vivo which are mainly due to different hormonal profiles. As only male mice were employed for in vivo experiments, this should be stated as a limitation of the study in the Discussion.

We agree and as mentioned above, we now address this limitation.

- Regarding the human data included in Fig. 5, an additional Table should be added to the Supplementary material providing the relevant characteristics of each patient separately (i.e., sex, age, known medications etc.).

We have added a detailed Table (new Table S2) as requested.

- As the metabolite profiles in Fig. 5 M look very variable also between mice from the same group, it would be interesting to know if they would still group separately after an unsupervised clustering analysis.

We performed an unsupervised clustering analysis and results from 4/5 animals from each genotype clustered together (see below). Thus, while as noted there is some variability in this analysis, there is a significant gene effect.

- For more clarity, the mouse model GRAF1^{gt/gt} should be better explained in the main text (i.e., in line 134 when it is mentioned for the first time).

Thank you for pointing this out, we added further description as is appropriate.

- A quantification of total LC3 levels as mentioned in the main text (lines 165-167) should be provided in Figure 1 K, L.

The notation in the text was corrected. We relied on the ratio of LC3I/LC3II to assess autophagy as described in “Guidelines for the use and interpretation of assays for monitoring autophagy- 4th edition”

- In the first Results section, the authors only observe impaired heart functionality in GRAFCKO mice upon isoproterenol administration. However, in Fig. 1 K and L LC3II/LC3I ratios in saline treated mice are reduced to the same extent than in ISO treated mice. If an impaired mitophagy is thought to be the cause of the impaired heart functionality, how do the authors justify this discrepancy?

As noted above, cardiac GRAF1-deficient mice phenocopied cardiac ATG5-deficient mice in that they had no significant differences at baseline, however both exhibited functional decline following isoproterenol treatment. This suggests that autophagy/mitophagy is not essential for basal cardiac function, but is necessary for situations in which there is a mismatch in energy supply and demand. Recent studies by the Muoio lab shed some light on why this might be the case as they have shown that mitochondrial flexibility is necessary to maintain energy stability and muscle function in conditions with high demand/fuel ratios (Cell Reports 2023). We now address this point in the discussion of our revised manuscript.

- In line 169, the text should be amended “as shown in REPRESENTATIVE figure 1M”, since no quantification is provided for this analysis.

Completed, thank you for pointing out this discrepancy.

- At the end of page 9, a short explanation should be added about the role of Atg7 in the proposed model. Also, full names for acronyms should always be provided the first time they are used in the manuscript.

Given our new results with Atg12, we discuss its role in the model. We also provide full names for acronyms as suggested.

REVIEWERS' COMMENTS

Reviewer #1 (Remarks to the Author):

The authors have addressed concerns raised by further experimentation (e.g. addition of compelling evidence that the phenotypes observed upon silencing of GRAF1 are due to mitophagy as opposed macroautophagy) and added text on the limitation of the study.

Reviewer #4 (Remarks to the Author):

No further comments.